# A presynaptic source drives differing levels of surround suppression in two mouse retinal ganglion cell types

David Swygart [1], Wan-Qing Yu [2], Shunsuke Takeuchi[3], Rachel O. L. Wong [2] & Gregory W. Schwartz [1,4,5] ✉

In early sensory systems, cell-type diversity generally increases from the periphery into the brain, resulting in a greater heterogeneity of responses to the same stimuli. Surround suppression is a canonical visual computation that begins within the retina and is found at varying levels across retinal ganglion cell types. Our results show that heterogeneity in the level of surround suppression occurs subcellularly at bipolar cell synapses. Using single-cell electrophysiology and serial block-face scanning electron microscopy, we show that two retinal ganglion cell types exhibit very different levels of surround suppression even though they receive input from the same bipolar cell types. This divergence of the bipolar cell signal occurs through synapse-specific regulation by amacrine cells at the scale of tens of microns. These findings indicate that each synapse of a single bipolar cell can carry a unique visual signal, expanding the number of possible functional channels at the earliest stages of visual processing.

Visual processing is already well underway in the retina. The analog luminance, contrast, and wavelength representations that begin in photoreceptors are transformed into >40 unique, behaviorally relevant channels of digital information that exit the retina via spikes in retinal ganglion cell (RGC) axons. Stratification of RGC dendrites with presynaptic bipolar cell (BC) and amacrine cell (AC) interneurons in the inner plexiform layer (IPL) is an established organizing principle by which retinal circuits build feature selectivity[1–4]. Nonetheless, the number of functionally distinct RGC types exceeds their stratification diversity[1,5,6]. What circuit motifs enable RGC types with nearly identical stratification patterns to have different light responses?

Previous studies have identified contributions to functional divergence from precise wiring specificity even within the same IPL stratum[7] or from differences in intrinsic properties of the RGCs[8,9]. Here, we examine such an example where two RGC types receive the same set of excitatory inputs but exhibit functionally distinct output signals. We isolate the circuit location at which their functions diverge.

Surprisingly, it is at the level of BC output synapses, speaking to whether BCs constitute single information channels or whether each BC can represent multiple channels[4,10–14].

We compared two RGC types in the mouse ($Pix_{ON}$ and ON alpha) that share very similar patterns of IPL stratification but show a striking difference in feature selectivity. The visual feature we investigated is surround suppression: one of the best-studied visual computations[15]. The first recordings of the receptive fields (RFs) of mammalian RGCs showed a center region that was antagonized by the surrounding region, resulting in strong responses to small stimuli that only activated the receptive field center and weaker responses to large stimuli that additionally activated the receptive field surround[16]. Over the many decades of work that followed, it has become clear that surround suppression is not computed by a single mechanism. Instead, it differs by species and cell types and can arise at multiple locations in the retina[15]. We sought to identify the circuit locations at which surround suppression is computed in $Pix_{ON}$ RGCs,

[1]Northwestern University Interdepartmental Neuroscience Program, Chicago, IL, USA. [2]Department of Biological Structure, University of Washington, Seattle, WA, USA. [3]Department of Biological Sciences, Graduate School of Science, The University of Tokyo, Tokyo, Japan. [4]Departments of Ophthalmology and Neuroscience, Feinberg School of Medicine, Northwestern University, Chicago, IL, USA. [5]Department of Neurobiology, Weinberg College of Arts and Sciences, Northwestern University, Chicago, IL, USA. ✉e-mail: greg.schwartz@northwestern.edu

where it is particularly prominent[17] compared to ON alpha RGCs, where it is much weaker[18].

Previous publications have shown that surround suppression can be driven by wiring patterns between specific cell types[19–23]. However, we show that Pix_ON and ON alpha RGCs have very similar circuit connectivity, particularly in their excitation, but show different levels of surround suppression in their spiking responses. We find that these differences in surround suppression are inherited from differences in the RGC presynaptic excitatory drive, suggesting that this computation occurs at the subcellular level. These findings reveal a surprising location for computing a classical receptive field property. More generally, they suggest that subcellular computation imparts neural circuits with even more capacity for functional

divergence than can be inferred from their synaptic wiring diagrams alone.

## Results

### The Pix_ON RGC has stronger surround suppression than the ON alpha RGC

We identified Pix_ON and ON alpha RGCs by their unique morphology and light responses[6,17,18]. These two RGC types have large dendritic arbors that primarily stratify in stratum 5 of the IPL and exhibit ON-sustained light responses (Fig. 1a–d). Despite their many similarities, the Pix_ON and ON alpha have been shown to correspond to two unique cell types[6,17,24,25]. Morphological characteristics, such as soma size and arbor complexity, do differ between the two cell types, and ON alpha

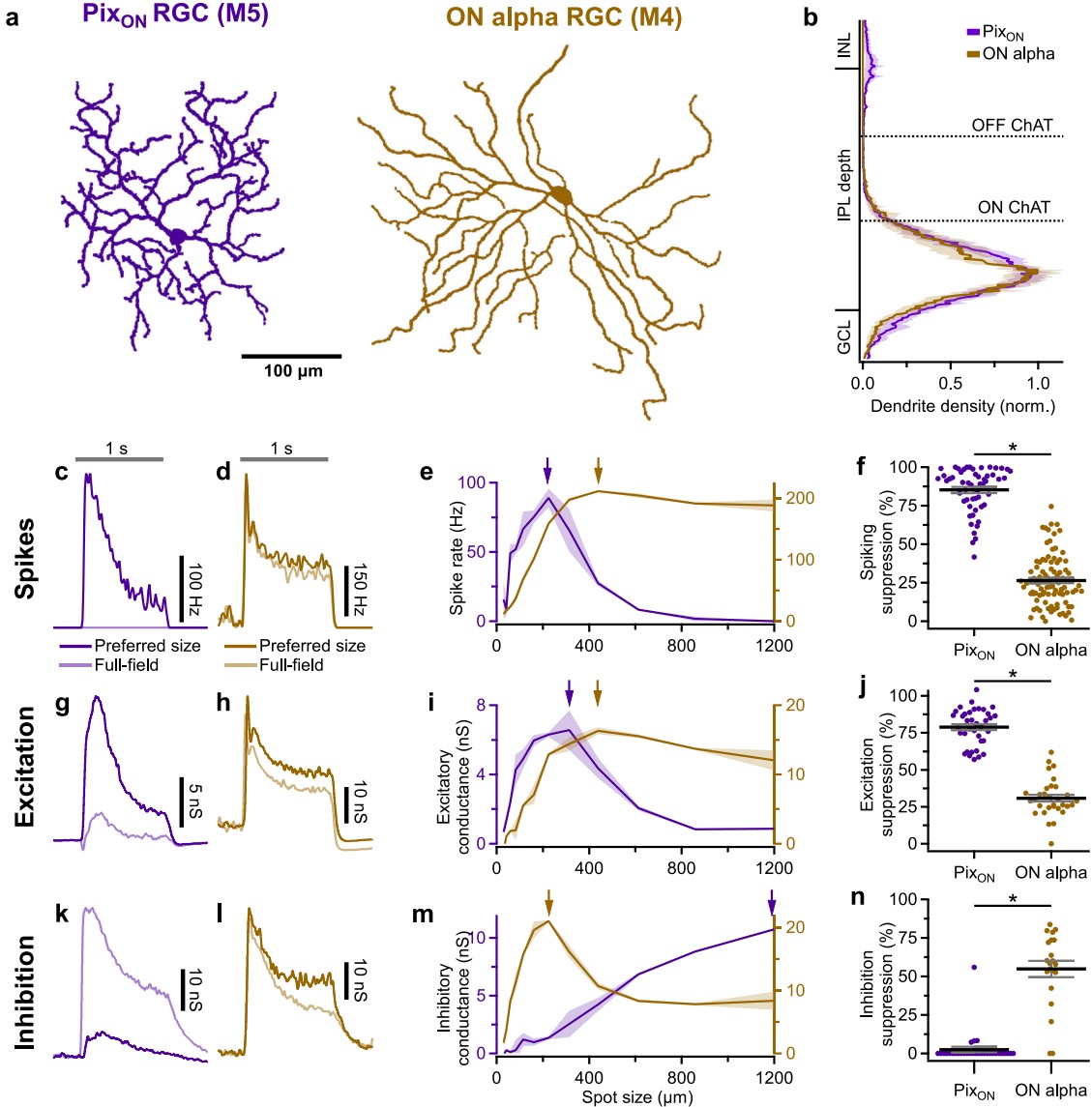

**Fig. 1 | Surround suppression is stronger in Pix_ON RGCs than in ON alpha RGCs.** **a** En-face view of a Pix_ON (purple) and an ON alpha (brown) dendritic arbor (maximum intensity z-projection after manual tracing). **b** Average dendritic stratification of Pix_ON (*n* = 19) and ON alpha (*n* = 10) RGCs within the inner nuclear layer (INL), inner plexiform layer (IPL), and ganglion cell layer (GCL). Dotted lines refer to the ON and OFF choline acetyltransferase (ChAT) bands used to determine stratification. Shaded region indicates standard error of the mean. **c** Example peristimulus time histograms recorded from a Pix_ON RGC in response to preferred size and full-field light spot stimuli. The gray horizontal bar indicates the 1-second presentation of the 250 R*/rod/s spot stimulus from a background luminance of -0.3 R*/rod/s.

**d** Same as (**c**), but recorded from an ON alpha RGC. **e** Mean spike rates recorded from a Pix_ON RGC (purple) and an ON alpha RGC (brown) in response to a range of spot sizes. Shaded region indicates the standard error of the mean. Arrows indicate the preferred spot size for each RGC. **f** Surround suppression of spiking response for Pix_ON (*n* = 55) and ON alpha (*n* = 90) RGCs. Dots indicate data from individual cells. Bar plots indicate average ± s.e.m., *\**p* < 0.05, two-sided Welch's *t* test. **g–j** Same as (**c–f**) but measuring excitatory conductances via whole-cell voltage clamp configuration. **j** Pix_ON (*n* = 37) and ON alpha (*n* = 31). **k–n** Same as (**c–e**) but measuring inhibitory conductances via whole-cell voltage clamp configuration. **n** Pix_ON (*n* = 32) and ON alpha (*n* = 21). Source data are provided as a Source Data file.

but not $Pix_{ON}$ RGCs are SMI-32 immunoreactive (Supplementary Fig. 1a–i and [ref. 26]). The $Pix_{ON}$ and ON alpha RGC types both exhibit weak intrinsic light responses and correspond to the M5 and M4 intrinsically photosensitive RGC types, respectively (Supplementary Fig. 1j, k and [refs. 24,27]). Functionally, these RGC types exhibit differing excitatory, inhibitory, and spiking receptive fields (Fig. 1c–n). Independent of receptive field properties, we distinguished $Pix_{ON}$ RGCs from ON alpha RGCs by their characteristic spike amplitude adaptation for spot sizes near the peak firing rate (Supplementary Fig. 2).

The most obvious way the $Pix_{ON}$ and ON alpha RGCs' receptive fields differ is in their magnitude of surround suppression. Both RGC types exhibited ON sustained spiking responses when presented with small preferred size spots of light (224 µm for $Pix_{ON}$ in Fig. 1c and 440 µm for ON alpha in Fig. 1d). However, when presented with full-field stimuli (1200 µm diameter spot), the $Pix_{ON}$ RGC's spike response was strongly suppressed, while the ON alpha's spike response was only weakly suppressed (Fig. 1c–f; $Pix_{ON}$ suppressed 89% ± 1.8%, $n = 46$; ON alpha suppressed 26% ± 1.8%, $n = 90$; $p < 10^{-47}$). This difference in surround suppression of the spiking responses between the two RGC types was present in both scotopic and photopic conditions (Supplementary Fig. 3) and across retinal locations (Supplementary Fig. 4).

To investigate if synaptic conductances could lead to the differing levels of surround suppression of the spiking response in these two RGC types, we voltage-clamped both cell types and recorded excitatory and inhibitory synaptic conductances across a range of stimulus sizes. Previous work demonstrated that $Pix_{ON}$ RGCs have spatially distinct regions of their receptive fields in which they receive excitation and inhibition[17]. We took advantage of this property to confirm that voltage-clamp effectively isolated excitation and inhibition (Supplementary Fig. 5). The excitatory conductances of both RGC types mirrored their spike responses; the $Pix_{ON}$ excitatory conductances showed strong surround suppression, and the ON alpha excitatory conductances showed weak surround suppression (Fig. 1g–j; $Pix_{ON}$ suppressed 79% ± 2.0%, $n = 37$; ON alpha suppressed 30% ± 2.1%, $n = 30$; $p < 10^{-24}$). As previously reported[17], the $Pix_{ON}$ inhibitory conductances were small for small spot sizes but continually increased for larger spot sizes. In contrast, the ON alpha inhibitory conductances were large for small spot sizes and moderately suppressed for larger spot sizes (Fig. 1k–n; $Pix_{ON}$ suppressed 2.6% ± 2.8%, $n = 31$; ON alpha suppressed 55% ± 5.3%, $n = 21$; $p < 10^{-8}$).

### Excitatory synaptic conductances drive surround suppression of $Pix_{ON}$ spiking responses

The differing levels of surround suppression between the $Pix_{ON}$ and ON alpha RGC spiking responses could be driven by differences in synaptic conductances (e.g., excitation and inhibition, see Fig. 1g–n) or by differences in cell-intrinsic factors (e.g., voltage-gated channels). To independently test the contribution of synaptic conductances and cell-intrinsic factors, we used dynamic clamp to simulate previously recorded $Pix_{ON}$ and ON alpha excitatory and inhibitory conductances in a new set of $Pix_{ON}$ and ON alpha RGCs (Fig. 2a). Figure 2b–g shows that strong surround suppression of the spiking responses occurred when simulating $Pix_{ON}$ conductances in either $Pix_{ON}$ RGCs (suppressed 99% ± 0.4%, $n = 4$) or ON alpha RGCs (suppressed 99% ± 1%, $n = 3$). In contrast, the simulation of ON alpha conductances induced weak surround suppression of the spiking responses in both $Pix_{ON}$ RGCs (18% ± 0.9%, $n = 4$) and ON alpha RGCs (23% ± 5%, $n = 3$). These results show that the differing levels of surround suppression in the $Pix_{ON}$ and ON alpha spiking responses are driven by their differing conductances (99% of total variance, $p < 10^{-12}$), not by cell-intrinsic factors (0.1% of total variance, $p = 0.22$, two-way ANOVA).

To test the relative role of excitation versus inhibition in driving surround suppression of $Pix_{ON}$ spiking responses, we again utilized dynamic clamp. First, we simulated excitatory and inhibitory conductances for the preferred spot size. To test the role of excitation, we then measured how much the preferred spot spiking response was suppressed when switching to full-field excitatory conductances while maintaining the same preferred size inhibition. Likewise, to test the role of inhibition, we measured how much the preferred spot spiking response was suppressed when simulating full-field inhibition while maintaining preferred size excitation (Fig. 2h).

We found that both inhibition and excitation induced some level of surround suppression of spiking responses. However, full-field excitation induced significantly more surround suppression of the spiking responses (96% ± 3%) than full-field inhibition (29% ± 3%; $n = 4$, $p = 0.0002$; Fig. 2i). These results suggest that suppression of the $Pix_{ON}$ excitatory conductances by full-field stimuli is an important driver of surround suppression in the $Pix_{ON}$ spiking output. Conversely, the absence of strong surround suppression of the ON alpha excitatory conductances allows the ON alpha RGC to exhibit very little surround suppression in its spiking output. While we acknowledge that dynamic clamp at the soma fails to simulate possible interactions between excitation and inhibition in the RGC dendrites, we found qualitatively similar results when blocking direct GABAeric inhibition onto the RGCs (Supplementary Fig. 6). Together, these data demonstrate that excitatory conductances play a prominent role in dictating surround suppression of the spiking output of $Pix_{ON}$ RGCs. We next investigated sources that could cause the excitatory conductances of the $Pix_{ON}$ and ON alpha RGCs to experience differing levels of surround suppression.

### Postsynaptic saturation or desensitization does not alter surround suppression of ON alpha excitatory conductances

Differing levels of surround suppression between the $Pix_{ON}$ and ON alpha excitatory conductances could result from differing expression of postsynaptic excitatory receptors. $Pix_{ON}$ RGC EPSCs did tend to be more transient than ON alpha EPSCs (Supplementary Fig. 7a, b). However, a cocktail of bath-applied drugs to block inhibition, nicotinic acetylcholine receptors, and NMDA receptors (strychnine, gabazine, saclofen, TPMPA, Hexamethonium, TTX, D-AP5), did not alter $Pix_{ON}$ EPSC kinetics (Supplementary Fig. 7c, d). Measuring IV curves in the presence of inhibitory blockers suggests that the excitatory conductances of both the $Pix_{ON}$ and ON alpha are driven by AMPA glutamate receptors (Supplementary Fig. 8). Additionally, blocking NMDA and nicotinic acetylcholine receptors did not significantly decrease the amplitude of excitatory conductances or the strength of surround suppression in $Pix_{ON}$ or ON alpha RGCs (Supplementary Fig. 9).

But perhaps the $Pix_{ON}$ and ON alpha express different types of AMPA receptors, with differing saturation or desensitization properties. If so, this could drive the differing levels of surround suppression of the $Pix_{ON}$ and ON alpha excitatory conductances. If the presynaptic BCs do experience strong surround suppression of their glutamate release, and the $Pix_{ON}$'s glutamate receptors do not saturate or desensitize, then the $Pix_{ON}$ excitatory responses would also have strong surround suppression inherited from the presynaptic BC. Whereas, if the ON alpha's glutamate receptors saturate or desensitize to the BCs' preferred size responses but not to the BCs' full-field responses, then surround suppression of the ON alpha's excitatory responses would be reduced (Fig. 3a).

To test if glutamate receptor saturation or desensitization is necessary for the weak surround suppression of ON alpha excitatory conductances, we measured surround suppression of ON alpha excitatory conductances during bath application of subsaturating concentrations of either a low-affinity glutamate receptor antagonist (700 nM kynurenic acid) or a high-affinity glutamate receptor antagonist (300 nM NBQX). While both kynurenic acid (KYN) and NBQX are expected to decrease the magnitude of the excitatory conductances, only the rapidly dissociating KYN is expected to prevent glutamate receptor desensitization and saturation[28]. KYN's rapid binding ($k_{on}$ = 1 to 50 µM⁻¹ s⁻¹) and dissociation ($k_{off}$ = 170–8600 s⁻¹)

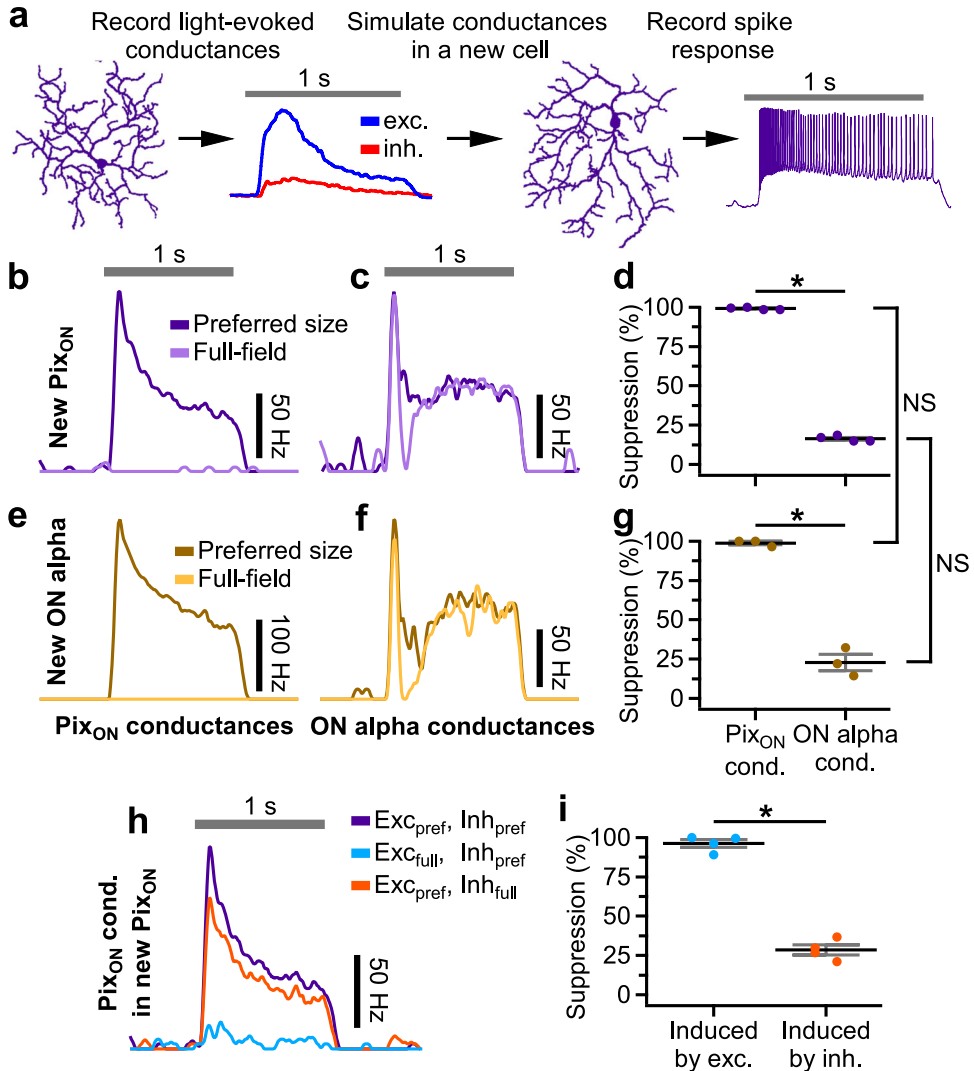

**Fig. 2 | Excitatory conductances drive differing levels of surround suppression in Pix_ON and ON alpha RGC spiking responses. a** Schematic illustrating dynamic clamp protocol in which previously recorded excitatory (blue) and inhibitory (red) conductances are simulated in a new RGC via current injections. Example peristimulus time histograms recorded from a Pix_ON RGC when simulating excitatory and inhibitory conductances recorded from a different Pix_ON RGC (**b**) or an ON alpha RGC (**c**). "Preferred-size" (dark purple) indicates the maximal spiking response when simulating conductances recorded during 200, 600, and 1200 μm diameter spot stimuli. "Full-field" (light purple) indicates simulation of conductances recorded during 1200 μm spot stimulus. **d** Surround suppression of Pix_ON spiking responses when simulating conductances recorded from a different Pix_ON (left) or an ON alpha (right) ($n = 8$). **e**–**g** Same as (**b**–**d**) but simulating conductances within

an ON alpha RGC ($n = 3$). **h** Example peristimulus time histograms recorded from a Pix_ON RGC when simulating Pix_ON conductances to isolate the effect of full-field excitation or full-field inhibition. Purple indicates simultaneous simulation of preferred size excitation and preferred size inhibition (same as "preferred size" in (**b**)). Blue indicates the simulation of full-field excitation and preferred size inhibition. Orange indicates simulation of preferred size excitation and full-field inhibition. **i** Suppression of spiking responses induced when switching from preferred size excitation to full field excitation (blue dots, $n = 4$) or switching from preferred size inhibition to full field inhibition (orange dots, $n = 4$). **d**, **g**, **i**, Dots indicate data from individual cells. Bar plots indicate average ± s.e.m., *$p < 0.05$, Significance was determined by two-way ANOVA for (**d**, **g**) and paired, two-sided, two-sample, Student's $t$ test for (**i**). Source data are provided as a Source Data file.

means it will bind to and dissociate from the AMPA receptor many times during the 1 s light stimulus[28]. While bound, KYN will protect the receptor from desensitization. When it then unbinds, it will allow glutamate to activate the AMPA receptor, thus decreasing the effective glutamate affinity of the receptors. This competition with KYN for binding sites additionally prevents glutamate from saturating AMPA receptors. Conversely, NBQX's slower dissociation ($k_{off} = 0.024$–$1.2\,s^{-1}$) means that AMPA receptors bound to NBQX will typically be rendered inoperable during the length of the 1 s light stimulus[28]. Thus, the AMPA receptors that are not bound to NBQX are still exposed to glutamate during the length of the stimulus and still undergo saturation and desensitization.

Excitatory conductances were significantly smaller in the presence of either KYN (20% ± 1% of control, $p < 10^{-3}$, $n = 3$) or NBQX

(18% ± 7% of control, $p < 10^{-2}$; $n = 3$; Fig. 3b–d). However, surround suppression of the ON alpha excitatory conductances was not stronger in the presence of KYN (12% ± 6%) compared to NBQX (17% ± 1.9%; $n = 3$, $p = 0.8$; Fig. 3e–g). These results suggest that neither glutamate receptor saturation nor desensitization is responsible for the weak surround suppression observed in ON alpha excitatory conductances.

Some caveats should be noted when interpreting these results. While KYN has been successfully used in the retina to detect effects driven by saturation and desensitization[28], we could not perform such a positive control for ON alpha excitatory conductances. Additionally, bath application of AMPA receptor antagonists could alter surround inhibition. So, we conducted an additional non-pharmacological experiment to test for differential saturation or desensitization of Pix_ON and ON alpha excitatory conductances. We stimulated both RGC

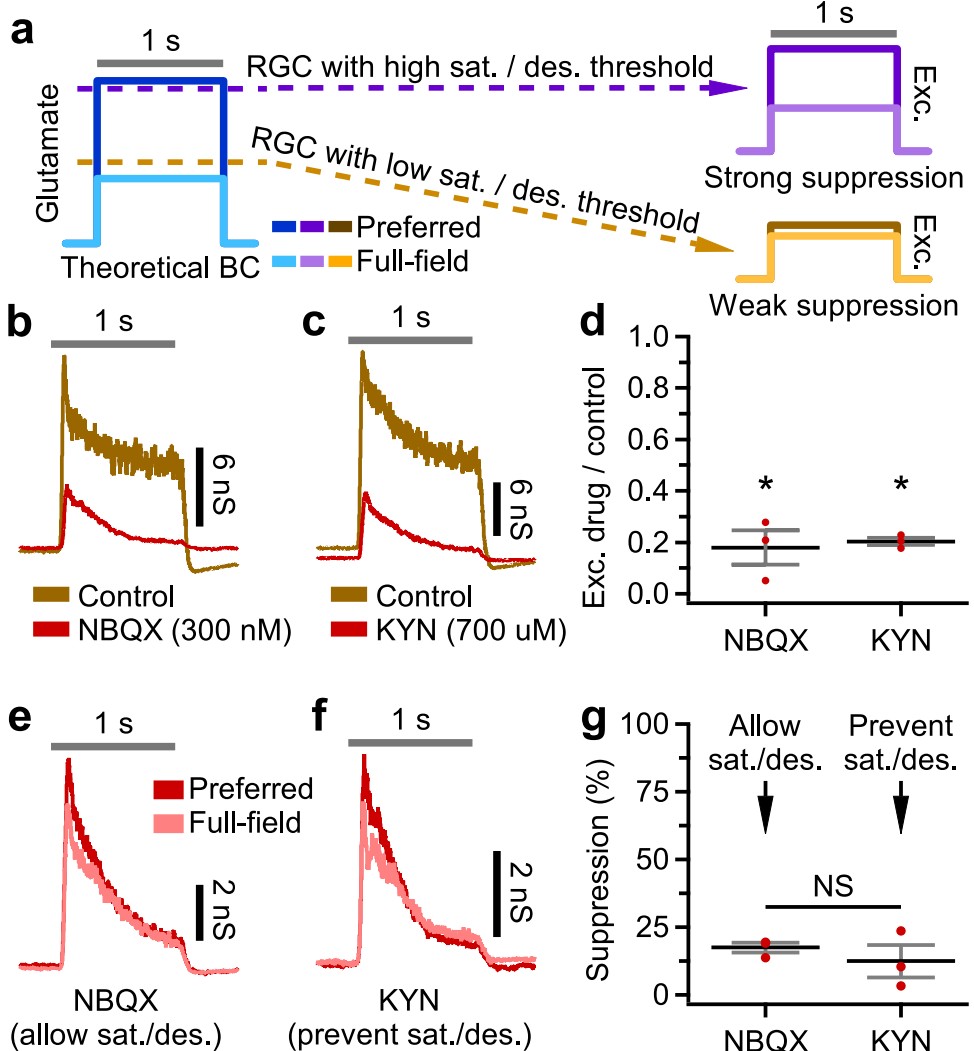

**Fig. 3 | Weak surround suppression of ON alpha excitatory conductances does not depend on glutamate receptor saturation or desensitization. a** Theoretical model hypothesizing how the saturation or desensitization of glutamate receptors could lead to decreased surround suppression of postsynaptic RGC excitatory conductances. Blue indicates a BC whose glutamate release has strong surround suppression. Purple indicates an RGC with glutamate receptors that do not undergo saturation or desensitization and thus responds with excitatory conductances that exhibit strong surround suppression inherited from the BC glutamate response. Brown indicates an RGC with glutamate receptors that do undergo saturation or desensitization; thus, the preferred size excitatory response is decreased relative to the full-field excitatory response. **b** Example ON alpha excitatory conductances evoked by a preferred spot size in control conditions (brown) or during subsaturating bath application of NBQX (red). **c** Same as (**b**), but red indicates bath application of kynurenic acid (KYN). **d** Proportion of ON alpha excitatory response (averaged across 1 s stimulus) evoked in NBQX ($n = 3$) or KYN ($n = 3$) compared to control conditions. **e** ON alpha excitatory conductances evoked by a preferred (red) or full-field (pink) spot size during bath application of NBQX. **f** Same as (**e**), but during bath application of KYN. **g** Surround suppression of ON alpha excitatory conductances in the presence of NBQX ($n = 3$) or KYN ($n = 3$). **a**, **b**, **c**, **f** Gray horizontal bar indicates a 1-second presentation of the stimulus. **d**, **g** Dots indicate data from individual cells. Bar plots indicate average ± s.e.m., *$p < 0.05$, paired, two-sided, two-sample Student's $t$ test. Source data are provided as a Source Data file.

types with a range of contrast steps. The two RGC types showed similar excitatory contrast response functions, and neither cell type experienced saturation at 100% contrast (Supplementary Fig. 10). Together, these results suggest that differential saturation or desensitization of postsynaptic glutamate receptors does not drive the differing levels of surround suppression between the Pix_ON and ON alpha RGCs.

**Surround suppression of Pix_ON and ON alpha excitatory conductances is accurately predicted from differing BC receptive fields but not differing RGC dendritic fields**
Having demonstrated that glutamate receptor saturation or desensitization is not the source of functionally distinct excitation in Pix_ON vs. ON alpha RGCs, we shifted our investigation upstream to the presynaptic BC subunits that drive excitation. An RGC's excitatory

receptive field is composed of BC subunits sampled across its dendritic arbor, with each of these BC subunits activated according to its own receptive field. The differing levels of surround suppression between the Pix_ON and ON alpha excitatory conductances could occur if their BC subunits had differing receptive fields, such as Pix_ON BC subunits exhibiting stronger surrounds. Alternatively, the differing levels of surround suppression between the Pix_ON and ON alpha excitatory conductances could be driven by differences in the Pix_ON and ON alpha dendritic arbors, resulting in a different spatial sampling of their BC subunits.

To investigate how BC receptive fields and RGC dendritic arbors might influence surround suppression of Pix_ON and ON alpha excitation, we modeled an RGC's light-evoked excitatory conductances as the summation of BC subunits sampled across its dendritic arbor

(Fig. 4a). We estimated the receptive field properties of these BC subunits by supplying the model with $Pix_{ON}$ or ON alpha dendritic skeletons and then optimizing the BC's center-to-surround ratio (CSR) and receptive field surround size ($\sigma_s$) so that the model output best replicated the cell's experimentally measured excitatory response across spot sizes.

Fitting the BC receptive fields to $Pix_{ON}$ RGCs resulted in a smaller CSR ($1.1 \pm 0.01$, median $\pm$ median absolute deviation) and a smaller $\sigma_s$ ($75 \pm 5\,\mu m$) than when fitting to ON alpha RGCs (CSR $= 1.8 \pm 0.2$, $\sigma_s = 100 \pm 20\,\mu m$; Fig. 4c, f). Encouragingly, these receptive field properties enabled the model to approximately reproduce the experimentally measured $Pix_{ON}$ and ON alpha excitatory responses (Fig. 4b, e) and are within the range of those estimated from ON BC glutamate signals[10,29].

When cross-validating these BC receptive fields, the model more accurately predicted surround suppression of excitatory responses when testing against RGCs of the same type to which the receptive field parameters were fit but failed to accurately predict surround suppression of the opposite cell type's excitatory responses (Fig. 4d, g). When fit to $Pix_{ON}$ RGCs, the model underestimated surround suppression of excitatory responses for new $Pix_{ON}$ RGCs by only $5\% \pm 3\%$ ($n = 14$) but overestimated surround suppression of ON alpha excitatory responses by $42\% \pm 3\%$ ($n = 8$). Conversely, when fit to ON alpha RGCs, the model underestimated surround suppression of excitatory responses for new ON alpha RGCs by only $0.8\% \pm 4\%$ ($n = 8$) but

underestimated surround suppression of $Pix_{ON}$ excitatory responses by $49\% \pm 3\%$ ($n = 14$).

To directly test if any BC RF could enable the RGC dendritic arbors to predict both the PixON and ON alpha excitatory responses, we simultaneously fit a single BC RF to both RGC types (Fig. 4h). This resulted in a BC RF that provided a poorer fit to both RGC types and whose surround size was much smaller than previously reported for BC glutamate release (CSR $= 1.1 \pm 0.04$, $\sigma_s = 44 \pm 7\,\mu m$, Fig. 4i[10,29]). When cross-validating against new $Pix_{ON}$ and ON alpha RGCs, this BC RF underestimated surround suppression of $Pix_{ON}$ excitatory responses by $29\% \pm 4\%$ and overestimated surround suppression of ON alpha excitatory responses by $14\% \pm 3\%$ (Fig. 4j).

Together, these results suggest that BC subunits with different receptive field properties are capable of producing the surround suppression observed in the $Pix_{ON}$ and ON alpha excitatory conductances but differences in the dendritic arbors of $Pix_{ON}$ and ON alphas do not appear capable of producing their differing levels of surround suppression.

### $Pix_{ON}$ and ON alpha RGCs receive input from the same BC types

If the functionally distinct excitation in $Pix_{ON}$ and ON alpha RGCs is driven by functionally distinct BC input, how might this difference arise? Although $Pix_{ON}$ and ON alpha RGCs have very similar stratification profiles in the IPL where they form synapses with BCs (Fig. 1b), perhaps they selectively form synapses with different BC types. To determine which BC types synapse onto the $Pix_{ON}$ and ON alpha RGCs,

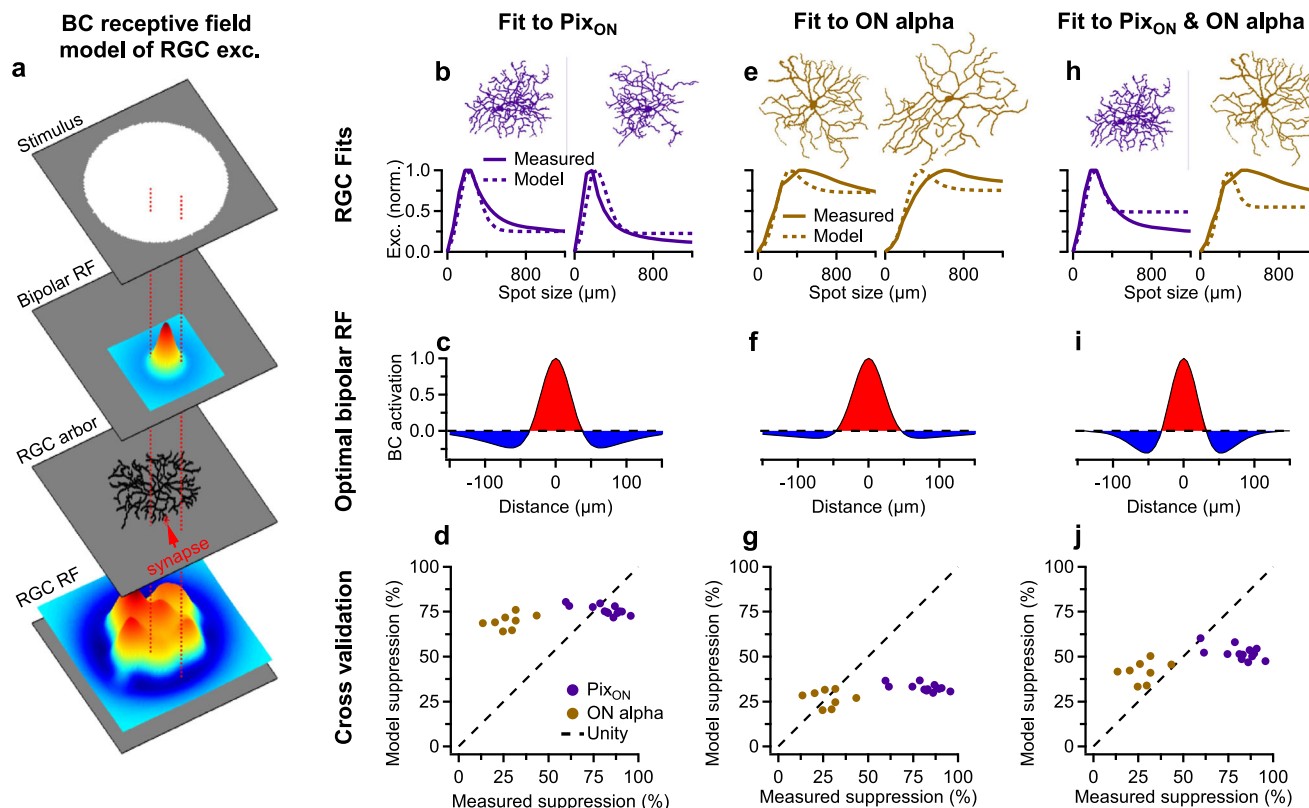

**Fig. 4 | A BC receptive field model of RGC excitation suggests differing BC receptive fields are necessary to evoke the differing level of surround suppression observed. a** Schematic illustrating the BC receptive field model of RGC excitation. The RGC receptive field (RGC RF) is constructed from BC receptive fields (Bipolar RF) randomly sampled across its dendritic arbor. RGC excitation is modeled as the summation of the RGC receptive field within a virtual stimulus. **b** Two example $Pix_{ON}$ dendritic arbors (*top*) and their corresponding excitatory conductances (*bottom*, solid line). Dotted lines indicate the model-predicted excitatory responses when using the BC RF in (**c**). **c** The BC RF that minimized the absolute

error between measured and model-predicted excitatory responses from (**a**) (see "Methods" for details). Fitting was performed simultaneously on 6 $Pix_{ON}$ RGCs. **d** Experimentally measured surround suppression from $Pix_{ON}$ ($n = 14$) and ON alpha ($n = 8$) RGCs plotted against the average surround suppression predicted by the model when cross-validating against a new set of $Pix_{ON}$ and ON alpha RGCs. Note: Alignment to unity indicates perfectly accurate model prediction. **e**–**g** Same as (**b**–**d**), but fitting to 6 ON alpha RGCs. **h**–**j** Same as (**b**–**d**), but simultaneously fitting to 3 $Pix_{ON}$ RGCs and 3 ON alpha RGCs. Source data are provided as a Source Data file.

we carried out serial block-face scanning electron microscopy (SBFSEM) on retinal sections that contained overlapping dendritic arbors of functionally identified $Pix_{ON}$ and ON alpha RGCs. We identified ribbon synapses onto the dendrites of both RGCs ($Pix_{ON}$ $n = 86$, ON alpha $n = 50$) and reconstructed their presynaptic BCs (Fig. 5). SBFSEM revealed that the $Pix_{ON}$ and ON alpha RGCs synapsed with the same BC types in similar proportions. Type 6 BCs (T6 BCs) provided the majority of excitatory synapses to both the $Pix_{ON}$ RGC (60% of excitatory input synapses) and the ON alpha RGC (52% of the excitatory input synapses). The remaining synapses were provided by type 7 (T7; $Pix_{ON} = 31\%$, ON alpha = 46%), type 8 (T8; $Pix_{ON} = 1\%$, ON alpha = 2%), and type 9 (T9; $Pix_{ON} = 7\%$, ON alpha = 0%) BCs. The proportion of input from each BC

type was not significantly different between the $Pix_{ON}$ and the ON alpha RGCs (T6 $p = 0.7$, T7 $p = 0.3$, T8 $p = 0.7$, T9 $p = 0.2$, two-proportions z-test with Holm-Bonferroni correction). This is also consistent with results from a different EM volume[30]. While the SBFSEM data allows us to count synapses, it does not offer a reliable measure of synaptic strength. However, previous work has successfully estimated functional BC input from anatomical synapse counts[31].

Although both RGC types received input from a similar complement of BC types, perhaps the $Pix_{ON}$ and ON alpha RGCs form synapses with distinct subpopulations of cells within the same BC type. To investigate this possibility, we analyzed the T6 and T7 BCs in the area of overlapping $Pix_{ON}$ and ON alpha dendrites with more than one

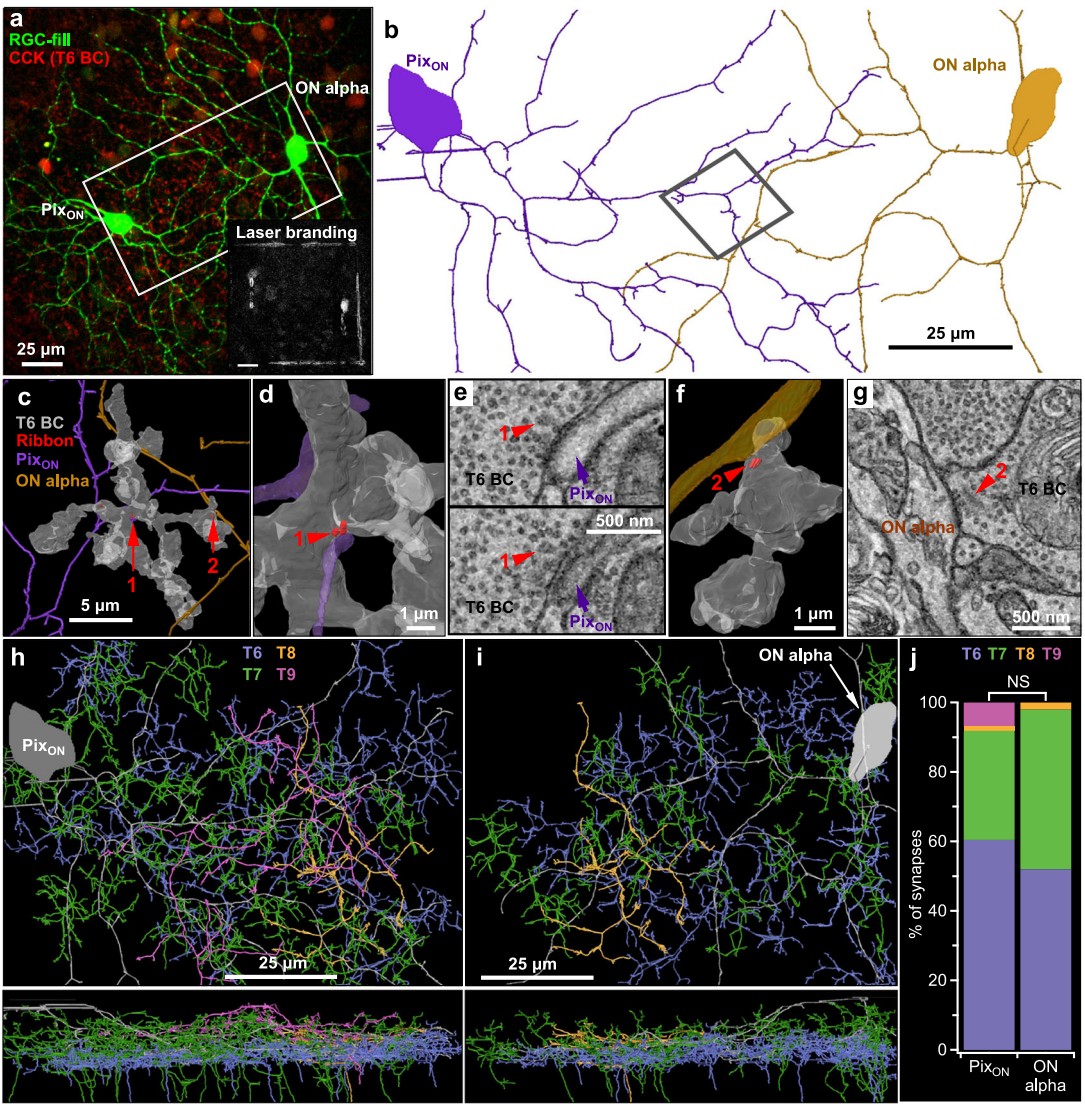

**Fig. 5 | $Pix_{ON}$ and ON alpha RGCs receive excitatory input from the same BCs.** **a** En-face view of filled $Pix_{ON}$ and ON alpha RGCs (green) imaged with 2-photon microscopy in the CCK-ires-Cre/Ai14 mouse line, which labels T6 BCs (red). Inset shows laser burn marks used as fiducial markers during SBFSEM alignment (see "Methods"). **b** $Pix_{ON}$ and ON alpha SBFSEM reconstructions of the tissue volume indicated by the white rectangle in (**a**). **c** Example reconstruction showing a T6 BC (semi-transparent gray mesh) forming ribbon synapses (red) onto a $Pix_{ON}$ dendrite (purple) and an ON alpha dendrite (brown). Reconstruction is taken from the approximate location indicated by the gray rectangle in (**b**) and rotated for better visibility of synapses. **d** Reconstruction of a T6 BC ribbon synapse onto a $Pix_{ON}$ dendrite (synapse #1 from (**c**)). **e** SBFSEM slices used to identify the ribbon synapse

from (**d**) (red arrow). Top and bottom slices are situated on the same XY location, but the bottom slice is 50 nm deeper in Z. **f, g**, same as (**d, e**) but showing a T6 BC ribbon synapse onto an ON alpha dendrite (synapse #2 from (**c**)). **h** En-face (top) and orthogonal view (bottom) of BC types (T6-T9) presynaptic to the $Pix_{ON}$ RGC. **i** Same as (**h**) but for BCs presynaptic to the ON alpha RGC. **j** The proportion of synapses formed by each BC type onto the $Pix_{ON}$ ($n = 86$ synapses) and the ON alpha ($n = 50$ synapses) RGCs. Differences in the proportion of BC type between $Pix_{ON}$ and ON alpha were not significant. $p > 0.05$, two-sided two-proportions z-test with Holm-Bonferroni correction for multiple comparisons. Source data are provided as a Source Data file. Data come from one reconstruction.

identified ribbon output synapse and determined if the BCs synapsed onto a single RGC type or onto both RGC types. We found that 5 of the 10 T6 BCs and 5 of the 7 T7 BCs formed synapses with both the $Pix_{ON}$ and the ON alpha RGC (Fig. 5c–g). However, we assume this to be a gross underestimate of homogenous connectivity given that our SBFSEM reconstruction only identified a single $Pix_{ON}$ and a single ON alpha, while we expect around three $Pix_{ON}$ and three ON alpha RGCs to be present given the high degree of dendritic overlap within the mosaics of each RGC type[1].

To investigate BC preference for $Pix_{ON}$ and ON alpha RGCs while taking into account the real RGC coverage factor, we analyzed $Pix_{ON}$ and ON alpha dendritic overlap with T6 and T7 BC arbors within a previously published EM reconstruction which identified a more complete mosaic of $Pix_{ON}$ and ON alpha RGCs (Supplementary Fig. 11a–f[1]). While we could not identify synapses in this dataset, previous studies have found a strong relationship between BC axon · RGC dendrite overlap and the number of BC · RGC synapses[31]. We found that each of the T6 BCs ($n = 47$) had axons that overlapped with both $Pix_{ON}$ and ON alpha RGCs (Supplementary Fig. 11c). Likewise, 97% of the T7 BCs ($n = 29$) had axons that overlapped with both $Pix_{ON}$ and ON alpha RGCs (Supplementary Fig. 11g). We analyzed the preference each of the T6 and T7 BCs had for $Pix_{ON}$ and ON alpha RGCs based on the proportion of overlap area with each RGC type. We found that most T6 and T7 BCs had similar preferences for both RGC types and were not different from a control in which BC location was randomly shifted (Supplementary Fig. 11d, h).

Since our SBFSEM reconstruction only covered a small area ($80 \times 150\,\mu m$), we sought an additional method to investigate BC input across the entire dendritic arbor of $Pix_{ON}$ and ON alpha RGCs. To do this, we filled $Pix_{ON}$ and ON alpha RGCs with Neurobiotin in a mouse line that fluorescently labels T6 BCs (CCK-ires-Cre/Ai14 [refs. 32,33]). We then used antibodies to fluorescently label an excitatory post-synaptic scaffolding protein present at excitatory synapses (PSD95; ref. 34, Supplementary Fig. 12a). After confocal imaging the entire dendritic volume, we identified which PSD95-labeled synapses within the RGC dendrite were apposed to T6 BCs (Supplementary Fig. 12b). In agreement with our SBFSEM results, we found that a majority of PSD95-labeled synapses were apposed to T6 BCs for both $Pix_{ON}$ RGCs ($61\% \pm 2\%$, $n = 3$) and ON alpha RGCs ($72\% \pm 3\%$, $n = 2$; Supplementary Fig. 12c) and these proportions did not significantly differ across dendritic eccentricity for either RGC type (Supplementary Fig. 12d, $Pix_{ON}$ p = 0.25, ON alpha $p = 0.17$, Kolmogorov-Smirnov test).

While the above experiments use anatomy to show common BC input to the $Pix_{ON}$ and ON alpha RGCs, they do not directly provide evidence of shared functional input. Previous work has demonstrated that RGCs which receive common input from upstream photoreceptors and BCs have correlated excitatory noise. However, we did not find significant correlations of the excitatory noise between $Pix_{ON}$ and ON alpha RGCs with overlapping dendrites (Supplementary Fig. 13). It should be noted, however, that the previous study was performed in primate retina, which might have different levels of localized regulation of its bipolar cells. Synapse-specific regulation of BC output could serve to decorrelate the excitatory conductances recorded in the $Pix_{ON}$ and ON alpha RGCs.

## Amacrine cells regulate the BC ribbon synapse

If the same BCs drive excitatory conductances in both the $Pix_{ON}$ and the ON alpha RGC types, then why is surround suppression different between the $Pix_{ON}$ and ON alpha excitatory conductances? Perhaps surround suppression is generated at a subcellular level within the axons of these BCs, allowing different output synapses of the same BC to convey either strong or weak surround suppression of their glutamate release. Wide-field ACs are a promising candidate for generating surround suppression in BCs because wide-field spiking ACs have been shown to provide surround suppression of BC depolarization and

glutamate release via $GABA_C$ receptors clustered at cone BC output synapses[29,35,36].

To examine the role of presynaptic inhibition by ACs in generating surround suppression of $Pix_{ON}$ excitatory responses, we measured $Pix_{ON}$ excitatory conductances in the presence of GABA or glycine receptor antagonists (Fig. 6a, b). Surround suppression of the excitatory responses was significantly decreased in the presence of a $GABA_C$ receptor antagonist (control $71\% \pm 4\%$; TPMPA $34\% \pm 4\%$; $n = 6$, $p < 10^{-3}$) but was not significantly altered by the application of a $GABA_A$ antagonist (control $76\% \pm 9\%$; gabazine $80\% \pm 10\%$; $n = 3$, $p = 0.8$), a $GABA_B$ antagonist (control $77\% \pm 5\%$; saclofen $74\% \pm 5\%$; $n = 4$, $p = 0.06$), or a glycine receptor antagonist (control $79\% \pm 9\%$; strychnine $78\% \pm 9\%$; $n = 3$, $p = 0.7$). Additionally, we found that surround suppression of the excitatory responses was significantly decreased in the presence of a voltage-gated sodium channel blocker (control $85\% \pm 5\%$; TTX $43\% \pm 2\%$; $n = 5$, $p = 0.003$), which is expected to block spike propagation along the neurites of spiking wide-field ACs. While TPMPA and TTX each significantly reduced surround suppression of the $Pix_{ON}$ excitatory conductances, some surround suppression remained. However, the simultaneous application of TPMPA and TTX completely abolished surround suppression of the excitatory responses (control $71\% \pm 4\%$, TPMPA + TTX $0.3\% \pm 0.3\%$, $n = 5$) and had a greater effect than TPMPA alone ($p = 0.0002$) or TTX alone ($p = 0.0002$). If the inhibition to the BC axon is carried completely by a spiking AC releasing GABA onto $GABA_C$ receptors, we would expect either TPMPA or TTX alone to abolish surround suppression rather than the result we obtained where both drugs were required. While interactions in bath-applied pharmacology experiments can be difficult to interpret, we speculate about possible explanations in the "Discussion".

Experiments in ON alpha RGCs showed qualitatively similar effects but decreases in surround suppression were more difficult to measure since surround suppression of the ON alpha excitatory conductances was already weak in control conditions (Supplementary Fig. 14). These results suggest that the strong surround suppression observed in the $Pix_{ON}$ excitatory conductances is driven by spiking wide-field ACs via $GABA_C$ receptors on the BC axon. While these same cells may drive what little surround suppression is present in the ON alpha excitatory conductances, they appear unable to induce the same level of surround suppression as seen in $Pix_{ON}$ excitatory responses.

While our pharmacology results suggested a role for presynaptic inhibition by spiking GABAergic ACs in generating surround suppression in $Pix_{ON}$ excitatory responses, they did not offer direct evidence of differential inhibition at synapses to $Pix_{ON}$ vs. ON alpha RGCs. To investigate presynaptic AC inhibition at a subcellular level, we reconstructed the ACs that formed output synapses onto the presynaptic BCs identified in our SBFSEM volume (from Fig. 5). For each T6 and T7 BC ribbon synapse onto the $Pix_{ON}$ and ON alpha RGC, we identified the nearest presynaptic inhibitory site (Fig. 6c). Although a presynaptic inhibitory site was always found within a few microns of each BC ribbon synapse, this distance tended to be shorter for T6 BC synapses onto the $Pix_{ON}$ RGC ($0.74 \pm 0.05\,\mu m$, $n = 51$) compared to the ON alpha RGC ($1.04 \pm 0.1\,\mu m$, $n = 26$; $p = 0.009$; Fig. 6f). Although this data was statistically significant, the difference in distance was very small, and its functional implications are unclear. Voltage signals are expected to be nearly identical at this scale (Fig. 7), but these different distances could suggest the presence of other functionally important differences, such as input from differing amacrine cell types or differing synapse structures. Additionally, this difference in presynaptic inhibitory distance was not significant for T7 BCs ($Pix_{ON} = 0.86 \pm 0.07\,\mu m$, $n = 14$; ON alpha $= 0.73 \pm 0.05\,\mu m$, $n = 17$; $p = 0.14$).

We traced the presynaptic ACs nearest to each T6 BC ribbon to determine if they were a likely candidate to carry inhibition from the surround. Due to the limited size of the SBFSEM reconstruction, only 60% of these ACs could be classified by field size, but all of these were

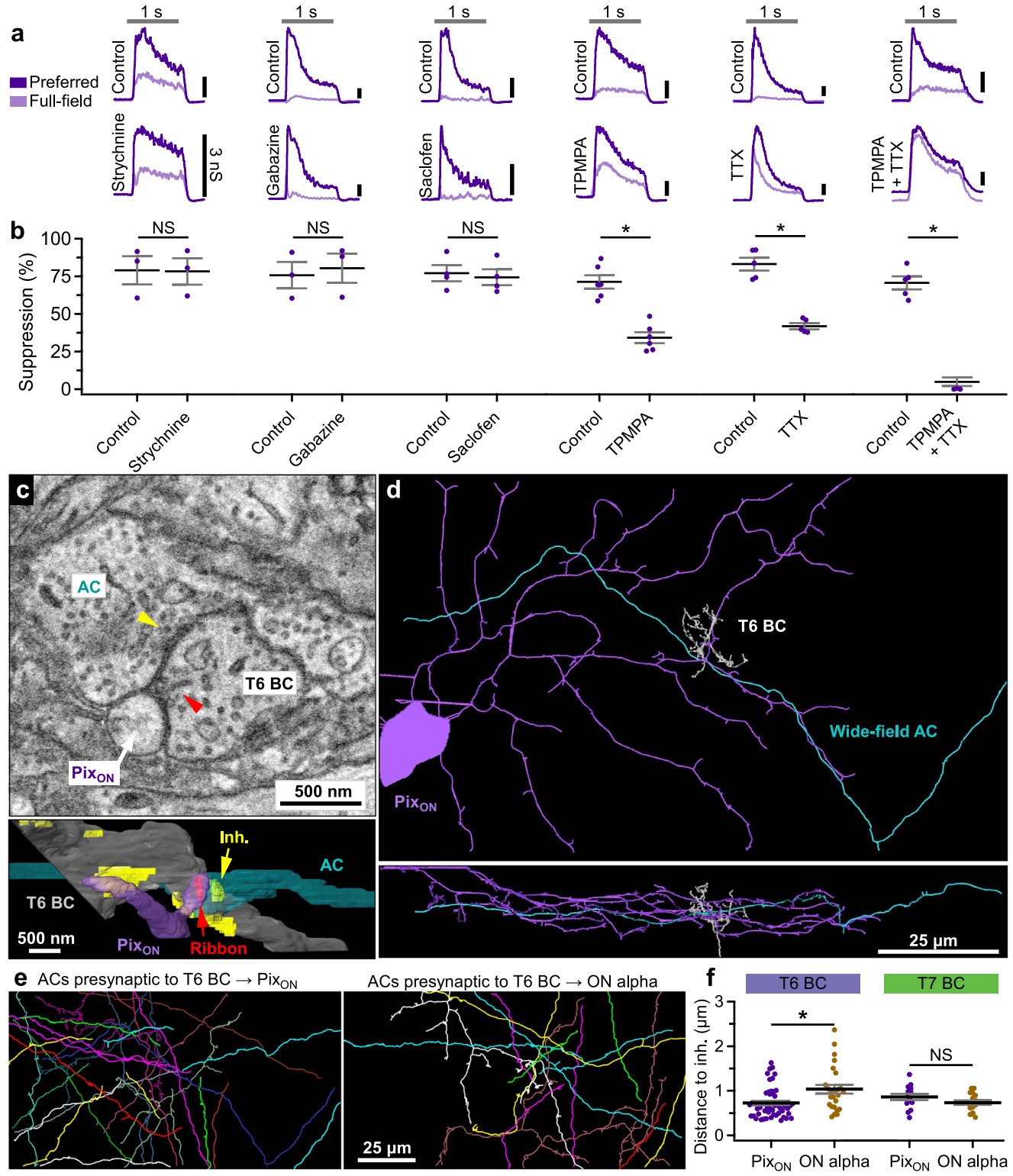

identified as medium to large-field ACs (spanning >40 µm), with none of their somas contained within the reconstructed volume (Fig. 6d, e). Additionally, of the nine ACs for which we observed multiple inhibitory feedback synapses onto the T6 BCs within the field of view, only one constituted the nearest neighbor for ribbon synapses onto both a $Pix_{ON}$ and an ON alpha, suggesting the possibility of synapse preference based on the postsynaptic ganglion cell identity. While we could not determine the specific cell type of these wide-field ACs, these results show that wide-field AC inhibition is present near each BC output synapse, but is more tightly localized at T6 BC-$Pix_{ON}$ synapses.

This suggests that synapse-specific regulation could occur within the same BC axonal arbor dependent upon the identity of the postsynaptic RGC type.

### Electrical compartmentalization in BC axons

Since we found that surround suppression of $Pix_{ON}$ excitatory responses was dependent on ionotropic $GABA_C$ receptors (Fig. 6a, b), one might hypothesize that subcellular surround suppression is achieved by subcellular hyperpolarization localized to BC-$Pix_{ON}$ output synapses. But BCs are small, and so is the distance between their

**Fig. 6 | Wide-field amacrine cell regulation near BC output synapses contributes to stronger surround suppression of $Pix_{ON}$ RGC excitatory responses.** **a** $Pix_{ON}$ excitatory conductances evoked before (top) and after (bottom) bath application of a glycine receptor antagonist (strychnine), a $GABA_A$ receptor antagonist (gabazine), a $GABA_B$ receptor antagonist (saclofen), a $GABA_C$ receptor antagonist (TPMPA), or $Na_V$ channel blocker (TTX). The gray horizontal bar indicates a 1-second presentation of the stimulus. Note: The response to full field stimuli in the TPMPA + TTX conditions was shifted down 2 nS to improve visibility. **b** Surround suppression of excitatory conductances in control and antagonists conditions. Dots indicate data from individual cells strychnine ($n = 3$), gabazine ($n = 3$), saclofen ($n = 4$), TPMPA ($n = 6$), TTX ($n = 5$), TPMPA + TTX ($n = 5$). Bar plots indicate average ± s.e.m., $*p < 0.05$, paired, two-sided, two-sample Student's $t$ test. **c** SBFSEM slice (top) and reconstruction (bottom) showing an AC neurite (cyan) forming an inhibitory synapse (yellow) onto a BC (gray), which then forms a ribbon synapse (red arrow) onto a $Pix_{ON}$ RGC dendrite (purple). **d** A zoomed-out En-face (top) and orthogonal (bottom) view of the AC from (**c**). **e** Reconstruction of nearest presynaptic ACs to T6 BC-to-$Pix_{ON}$ (left) and T6-to-ON alpha (right) ribbon synapses. **f** Distance to nearest inhibitory from T6 BC output synapses ($Pix_{ON}$ $n = 51$, ON alpha $n = 26$) and T7 BC output synapses ($Pix_{ON}$ $n = 14$, ON alpha $n = 17$). Dots indicate data from each BC-to-RGC synapse. Bar plots indicate average ± s.e.m., $*p < 0.05$, two-sided Welch's $t$ test. Source data are provided as a Source Data file. Data come from one reconstruction.

output synapses, bringing into doubt whether voltage could differ enough between output synapses to cause differing glutamate release. To investigate whether electrical compartmentalization can support functionally divergent signals from a BC, we generated a morphologically detailed NEURON compartmental cable model[37] from an SBFSEM reconstruction of a T6 BC, including the locations of 84 ribbon output synapses and 120 presynaptic inhibitory synapses (Fig. 7a). Although BCs are often modeled using only passive membrane properties[13,38,39], multiple studies have measured voltage-gated ion conductances from BCs which could lead to greater electrical compartmentalization[40–45]. Thus we performed all experiments in both a passive model and an active model containing L-type $Ca^{2+}$ channels[41,42], $K_V^+$ channels[43], and HCN2 channels[44,45] (see "Methods" for model details, Supplementary Table 3 for parameter values, and Supplementary Fig. 15 for robustness tests).

To estimate the ability of the inhibitory sites to differentially suppress ribbon output synapses, we measured the ratio of the excitatory center versus the inhibitory surround for each ribbon output synapse (Fig. 7b, c). To measure the excitatory centers, we activated excitatory synapses (10.1 mV reversal potential) on the dendrites and measured the resulting depolarization at each ribbon output synapse. To estimate the inhibitory surrounds, we repeated the simulations, activating the dendritic excitatory synapses but additionally activated subsets of inhibitory synapses (−50.4 mV reversal potential) on the BC axonal arbor. The inhibitory surround was taken as the hyperpolarization induced by activating the inhibitory synapses.

When activating a single inhibitory synapse, we found that the range of CSR values measured across the 84 output ribbon synapses tended to be much larger for the active BC model than for the passive BC model (Fig. 7d). To quantify this range, we split the ribbons into quartiles based on their CSR values. We then calculated the difference between the average CSR value of the fourth quartile (21 ribbons with the highest CSR values) and the average CSR value of the first quartile (21 ribbons with the lowest CSR values). This analysis aimed to determine whether sets of ribbons from the same BC could feasibly support both the strong surround suppression of excitation we observed in $Pix_{ON}$ RGCs and the weak surround suppression of excitation we observed in ON alpha RGCs. Of course, selective subcellular wiring of the BCs to the two RGC types would also be required for such a circuit to match our data.

As it seems unlikely that the BC's inhibitory surround is conveyed by a single inhibitory synapse, we repeated the sequential activation of each of the 120 inhibitory synapses but also included the simultaneous activation of the N-nearest neighbors to that synapse. We do not know if real inhibition onto the BC axonal arbor follows the specificity of the N-nearest synapse activation that we tested, but we chose this activation method since it allows us to estimate the upper bounds of electrical compartmentalization. We found that as more inhibitory synapses were simultaneously activated, the range of CSR values measured at the ribbons decreased (Fig. 7e–g).

We ran the model while varying a range of parameter values to test the robustness of the active model (Supplementary Fig. 15). We found that varying L-type $Ca^{2+}$ channel conductance, HCN2 channel conductance, and leak channel conductance had little impact on ribbon CSR values. However, increasing $K_V^+$ channel conductance, cytoplasmic resistivity, or ribbon depolarization did increase the range of CSR values measured at the ribbons.

To test if the active model's greater range of CSR values was caused by simply increasing total membrane conductance through the active channels, we measured conductance through each of the channels during simulations in which sets of 60 N-nearest inhibitory synapses were activated (Supplementary Fig. 16). We found that total membrane conductance was 5.4 ± 0.15 (mean ± std) times greater in the active model compared to the passive model, with $K_V^+$ channels being the greatest contributor to this increased membrane conductance. To determine if the dynamic properties of the active channels played a role beyond increasing total membrane conductance, we increased the leak conductance in the passive model to match the total membrane conductance in the active model and recorded 4.4 times greater CSR range compared to the original passive model. However, this CSR range was still about half of that recorded in the active model with similar total membrane conductance. These results suggest the voltage-gated channels greatly increase total membrane conductance and thus decrease the effective membrane resistance. This increase in membrane conductance increases the range of CSR values measured at the ribbons, but the dynamic properties appear to increase the range of ribbon CSR values even further.

To measure electrical compactness across a range of frequencies more directly, we injected sinusoidal currents at the BC soma and measured the resulting voltage fluctuation at the soma and at all of the ribbon output synapses (Supplementary Fig. 17). By measuring the attenuation of the voltage fluctuations, we could calculate a length constant for each ribbon output synapse. We first injected a range of sinusoidal frequencies in the active BC model without activating inhibitory or excitatory synapses. We found the length constant for very low frequencies (0.25 Hz) was 455 ± 41 μm. The length constant peaked at 10 Hz (1186 ± 103 μm) and decreased for higher frequencies (174 ± 16 μm at 250 Hz). We repeated these experiments while activating the excitatory and inhibitory synapses and found that ribbon length constants were much lower and varied less across stimulation frequencies (71 ± 6.2 μm at 0.25 Hz, 67 ± 6.2 μm at 250 Hz). Thus, the decreased membrane resistance (shunt) caused by the synaptic conductances had a large effect on electrical compartmentalization in our BC model and caused electrical compartmentalization to become largely frequency-independent.

We examined the anatomical features that influenced ribbon CSR values in the model and found that ribbons nearer to the activated inhibitory synapses tended to have lower CSR values, while ribbons further from the activated inhibitory synapses had higher CSR values due to voltage decay of the inhibitory surround (Supplementary Fig. 18a–c). Additionally, we found that ribbon synapses nearer to the soma tended to have higher CSR values because they were closer to the excitatory synapses and further from the inhibitory synapses (Supplementary Fig. 18d–f). This relationship was relatively weak when activating smaller sets of inhibitory synapses and was most robust when activating all 120 inhibitory synapses. Since we do not know how

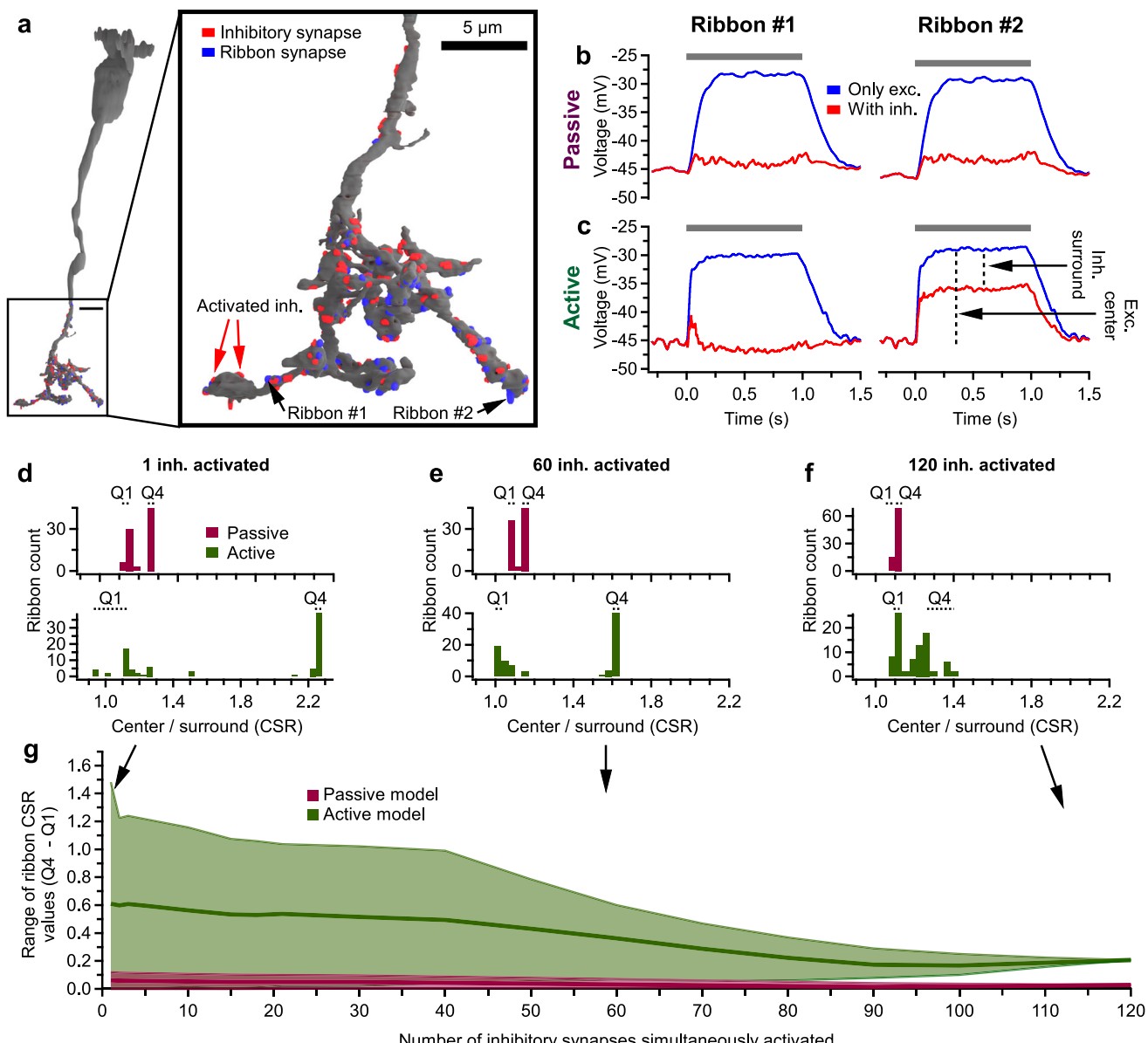

**Fig. 7 | Inhibitory surround strength measured in the axons of a BC compartmental cable model. a** A SBFSEM reconstruction of a T6 BC, including 84 ribbon output synapses (blue) and 120 inhibitory input synapses (red). **b** Voltage of the synaptic ribbons indicated in (**a**) (black arrows) during simulation experiments in a passive model of the T6 BC. Blue lines indicate voltage recorded during simulations in which excitatory synapses located on the BC dendrites were stochastically activated for 1 s (gray bar). Red lines indicate simulations in which the same excitatory dendritic synapses were activated while simultaneously activating the two inhibitory axonal synapses indicated by red arrows in (**a**). **c** Same as (**b**), but simulations were performed in an active model of a T6 BC whose membrane contained voltage-gated channels (L-type Ca²⁺, $K_V^+$, and HCN2). Black arrows illustrate the measurement of the excitatory center as the average depolarization from baseline induced by stimulation of the excitatory synapses and the measurement of the inhibitory surround as the average hyperpolarization when stimulating the inhibitory synapses on the axons. **d** Example histogram of the center-to-surround ratio (CSR) measured at each ribbon synapse in the passive (*top*) and active (*bottom*) BC model when activating a single inhibitory synapse. Q1 indicates the quartile of ribbon synapses (21 ribbons) with the lowest CSR values, and Q4 indicates the quartile of ribbon synapses (21 ribbons) with the highest CSR values. **e**, **f**, Same as (**d**) but when simultaneously activating 60 (**e**) or 120 (**f**) inhibitory synapses. **g** Range of CSR values resulting when stimulating different numbers of N-nearest inhibitory synapses. CSR range is calculated as the difference between the average CSR value of the top quartile of ribbon synapses (Q4) and the average CSR value of the bottom quartile of ribbon synapses (Q1). Number of inhibitory synapses indicates activation of a subset of N-nearest inhibitory synapses, which was repeated for each of the 120 inhibitory synapse locations. Thick lines indicate the median range of CSR values measured, and thin lines indicate the maximum and minimum range of CSR values measured across all 120 sets of inhibitory synapses. Note: As inhibitory synapse number increases, maximum and minimum range values converge on the median as there are 120 range values obtained when activating one inhibitory synapse but only one CSR range value obtained when activating all 120 inhibitory synapses. Green lines indicate the active model of the T6 BC, and red lines indicate the passive model of the T6 BC. Source data are provided as a Source Data file.

many and which of the 120 inhibitory synapses truly carried surround suppressive signals, it is unclear how much of an effect this mechanism could have. We analyzed the SBFSEM reconstruction, for evidence of such a mechanism, but did not find that the T6 BC–ON alpha synapses occurred closer to the BC primary axonal stalk (Supplementary

Fig. 18g), suggesting that this anatomical mechanism does not contribute to decreased surround suppression in the ON alpha.

A prerequisite for the Pix_ON and ON alpha excitatory responses obtaining their differing levels of surround suppression through functional divergence within the BC axon is that the Pix_ON RGCs

selectively synapse with BC ribbons with low CSR values while ON alpha RGCs selectively synapse with BC ribbons with high CSR values. It is unknown if this kind of functionally selective wiring occurs. But even if functionally selective wiring occurs, could the range of CSR values predicted by the BC model in Fig. 7 enable the differing levels of surround suppression measured in the $Pix_{ON}$ and ON alpha excitatory conductances? Answering this question requires moving beyond a model of a single BC, as an RGC receives input from many BCs across its dendritic arbor. Thus, we combined the results of the BC compartmental cable model with the previously described BC receptive field model that predicts an RGC's excitatory response as the summation of BC receptive field subunits sampled across its dendritic arbor (Fig. 4).

To predict $Pix_{ON}$ excitatory responses, we provided the BC receptive field model with the $Pix_{ON}$ dendritic arbors and a BC receptive field with a CSR value equal to the average first quartile of ribbons. Likewise, to predict ON alpha excitatory responses, we provided the BC receptive field model with the ON alpha dendritic arbors and a BC receptive field with a CSR value equal to the average fourth quartile of ribbons (Fig. 8a–c). The model then predicted the excitatory conductances for a range of spot sizes for each of the RGC dendritic arbors (Fig. 8d). We chose to assign the $Pix_{ON}$ RGCs with the lowest CSR quartile of ribbons and the ON alpha RGCs with the highest CSR quartile of ribbons to investigate if sufficient voltage compartmentalization could occur given highly selective synapse formation. However, we do not know if this kind of functionally specific synapse formation occurs in the T6 BC, and our SBFSEM volume does not suggest increased path distance between ribbons with output to differing RGC types (Supplementary Fig. 18h).

Figure 8e shows that in the passive BC model, none of the individual inhibitory synapses provided inhibition with enough voltage decay along the length of the neurite to support the decreased surround suppression measured from the ON alpha excitatory conductances. However, in the active BC model, most inhibitory synapses predicted strong surround suppression of $Pix_{ON}$ excitatory responses and weak surround suppression of ON alpha excitatory responses. These differing levels of surround suppression were similar to experimentally measured surround suppression of excitatory responses in these cells (Fig. 8f). When simultaneously activating 60 N-nearest inhibitory synapses, some sets of inhibitory synapses induced a range of CSR values that accurately predicted $Pix_{ON}$ and ON alpha surround suppression. However, when activating all 120 inhibitory synapses, the model did not match experimentally measured responses and overestimated ON alpha surround suppression.

These modeling results suggest that the BC axonal arbor is not completely isopotential and that voltage gradients could feasibly contribute to localized glutamate release. However, the model suggests that this could only occur if specific subsets of inhibitory neurons provide surround suppression (<60 of 120 inhibitory synapses) and that the $Pix_{ON}$ and ON alpha RGCs selectively synapse with the presynaptic ribbons with strong (Q1) and weak (Q4) surround suppression, respectively. However, a voltage gradient within the BC arbor is not absolutely required for functional divergence. We speculate in the Discussion about chemical sources of subcellular functional divergence at small spatial scales within the BC axonal arbor that could work in concert with or independent from voltage gradients.

## Discussion

Our study identified the site of functional divergence between an RGC type with strong surround suppression of its spiking responses ($Pix_{ON}$) and a type with weak surround suppression of its spiking responses (ON alpha). We found this signal divergence occurred within the output of a shared set of presynaptic cells. This is contrary to the prevailing view of the central nervous system, where functional divergence occurs via differing neuronal cell types. The capacity for

subcellular functional compartmentalization requires a new framework for information processing in the excitatory pathways of the retina. These results and an increasing set of similar observations throughout the brain suggest that more detailed biophysical work on the pre-synapse is required to appreciate the computational complexity of neuronal output.

### Subcellular output divergence in the retina and the brain

In spiking neurons, action potentials measured at or near the soma are typically considered all-or-none signals that invade the full axon to drive synaptic release. Compact, non-spiking neurons, like BCs, are typically modeled as being isopotential[13]. Thus, somatic voltage is assumed to describe synaptic release. There is precedent, however, for both spiking and non-spiking neurons transmitting different signals at different output locations. The degree to which functional divergence occurs in axons, albeit generally axons with much larger arbors than those of BCs, has been highlighted as one of the most important questions in neuroscience[46].

In the retina, subcellular functional divergence has been demonstrated in several types of non-spiking ACs, including the A17[47], VGluT3[48], and starburst[49]. These ACs, however, have substantially larger neuritic arbors than BC axons, so electrical compartmentalization is greater and can more easily support functional divergence. Some types of BCs in the zebrafish retina have distinct lobular output boutons in different layers of the IPL, which can display different light-driven calcium signals. Differential bouton volume has been suggested as a mechanism for functional divergence in these BCs[50]. In the mouse retina, rod BCs have been shown to vary the level of synchrony between their output synapses[51].

Subcellular functional divergence has also been observed in other parts of the central and peripheral nervous systems. Leech mechanoreceptors can propagate spikes to different postsynaptic neurons and fail to propagate to others depending on which part of their receptive field is stimulated[52–57]. Auditory afferents in bush cricket can display different frequency tuning in nearby parts of the axonal arbor through a mechanism involving presynaptic inhibition[58]. Mechanisms for the functional compartmentalization of the axonal arbors of spiking neurons have included both intrinsic electrical properties[59–64] and the external influence of GABAergic interneurons[65–67]. Motor neurons in both rat[59–61] and spiny lobster[68–70] propagate spikes down some parts of the axonal arbor but not others. Motor neurons in locusts contain two axonal branches, each with its own axon initial segment that can initiate spikes independently. These spikes often propagate to the opposite branch but can fail to propagate in some conditions[71].

### Possible mechanisms of synapse-specific surround suppression in BCs

Functionally distinct synaptic release from a single BC is difficult to reconcile with the canonical view that transmitter release within a neuron is controlled exclusively by presynaptic voltage, especially for a small cell type typically assumed to be isopotential. Indeed, our modeling suggests that a passive model of the BC would not support sufficient electrical compartmentalization for much subcellular functional divergence. However, the inclusion of active conductances decreased the effective membrane resistance to the point that voltage gradients could feasibly contribute to functionally divergent signals within a single BC.

This model contained many assumptions, such as the degree of functionally specific synapse formation and the linearity of the voltage-to-glutamate relationship. If these assumptions are shown to be incorrect, the model may overestimate or underestimate voltage-driven functional divergence. For example, if low CSR BC ribbons do not selectively synapse with $Pix_{ON}$ RGCs and instead indiscriminately synapse with both $Pix_{ON}$ and ON alpha RGCs then the model would overestimate voltage-driven functional divergence. On the other hand,

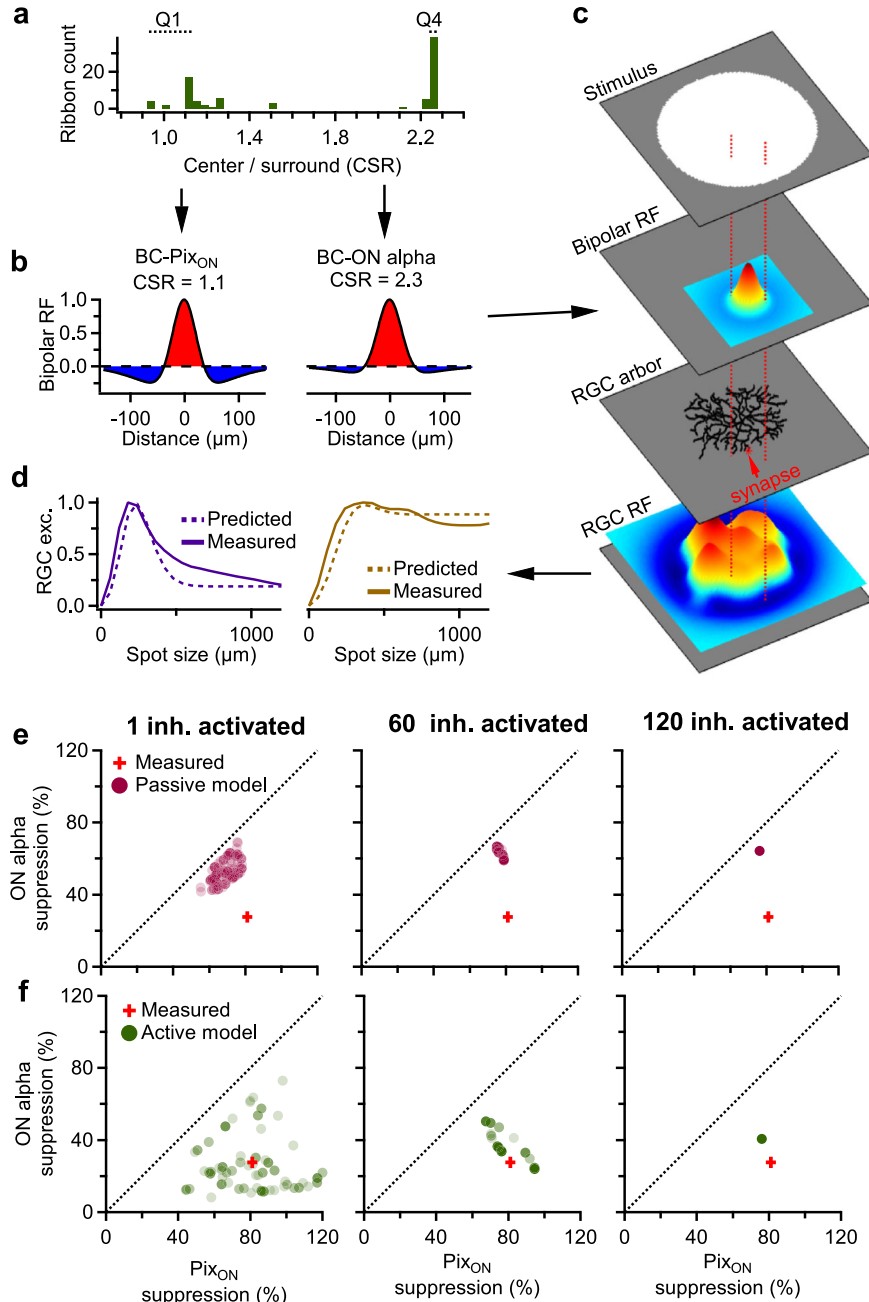

**Fig. 8 | Modeling suppression of RGC excitatory responses through electrical isolation of inhibition in the BC axonal arbor. a** Example histogram of CSR value measured at each of the BC ribbon output synapses when activating a single inhibitory synapse in the active compartmental cable model of the BC (see Fig. 7d). Q1 indicates the quartile of ribbons with the lowest CSR values, and Q4 indicates the quartile of ribbons with the largest CSR values. **b** Difference of Gaussian receptive field when using the average Q1 CSR value (left) or the average Q4 CSR value (right). Note: Center size and surround size were fixed at values obtained when fitting to Pix$_{ON}$ excitatory conductances (see Fig. 4c and "Methods"). **c** BC receptive field model of RGC excitation, which predicts RGC excitatory responses as the sum of BC receptive field subunits (difference of Gaussian receptive fields from (**b**)) sampled across the RGC dendritic arbor (see Fig. 4a and "Methods"). **d** RGC excitatory conductances for a range of spot sizes predicted by the BC receptive field model (dotted line) and experimentally measured (solid line). Left shows the RGC excitatory responses predicted when providing the model with Pix$_{ON}$ RGC dendritic arbors and a BC subunit receptive field with a CSR value of 1.1 (Q1 average from (**a**)). Right shows the RGC excitatory responses predicted when providing the model

with ON alpha RGC dendritic arbors and a BC subunit receptive field with a CSR value of 2.3 (Q4 average from (**a**)). **e, f** Average surround suppression of Pix$_{ON}$ excitatory responses plotted against average surround suppression of ON alpha excitatory responses. Dots indicate values predicted from the BC receptive field model with Pix$_{ON}$ surround suppression predicted using the average Q1 CSR values, and ON alpha surround suppression predicted using the average Q4 CSR values. Note: Each dot represents a separate simulation in which a unique set of inhibitory synapses were simultaneously activated. The red cross indicates the average surround suppression of excitatory conductances experimentally measured in the Pix$_{ON}$ RGCs (81% ± 3.7%, $n = 14$) and the ON alpha RGCs (28% ± 3.1%, $n = 8$). The length of the cross lines indicates the standard error of the mean. Predictions were made with CSR values obtained when activating one inhibitory synapse (left), 60 inhibitory synapses (*middle*), or all 120 inhibitory synapses (right). For (**e**), CSR values were obtained from the passive compartmental cable model of the BC. For (**f**), CSR values were obtained from the active model of the T6 BC. Source data are provided as a Source Data file.

if the relationship between voltage and glutamate release is actually supralinear rather than linear[72,73], our model would underestimate voltage-driven functional divergence. Regardless, our model highlights the importance of considering active conductances and challenges the assumption that all output synapses experience the same voltage signal.

Although our modeling suggests the possibility of voltage compartmentalization, we were unable to find anatomical evidence of wiring specificity in our SBFSEM data (Supplementary Fig. 18). Of course, many functional features of synapses cannot be resolved in ultrastructure. We speculate that chemical compartmentalization within the BC axonal arbor could also contribute to functional divergence. But which molecule(s) could be localized at the micron scale to alter glutamate release? We think the most likely mechanism is that an external, diffusible chemical signal (from ACs to the BC ribbon) causes local regulation of individual synapses. Our pharmacology results showed that GABA$_C$ receptors contribute to surround suppression of Pix$_{ON}$ excitation (Fig. 6a, b), but this does not exclude the involvement of another modulator. Perhaps GABA release from spiking ACs causes a moderate hyperpolarization of the BC, which provides some level of surround suppression of its glutamate release, but differing levels of surround suppression are achieved through the simultaneous release of an additional modulator. Such a modulator could induce a voltage-independent reduction of Ca$^{2+}$ at BC-Pix$_{ON}$ synapses or a voltage-independent enhancement of Ca$^{2+}$ at BC-ON alpha synapses. Calcium channels clustered near ribbon synapses have been shown to locally control vesicle release via calcium nanodomains at the ~20 nm scale[74]. Such a mechanism could explain why bath application of TPMPA decreased but did not completely abolish surround suppression of the Pix$_{ON}$ excitatory conductances.

Collectively, ACs contain at least 20 different small molecule or peptide transmitters and neuromodulators, and the differential expression of these molecules is one of the primary ways to classify them into different types[75], yet the functions of most of these molecules in visual processing remain largely unknown. There are several reports of voltage-independent effects of these substances on calcium levels. Ca$^{2+}$-permeable α7 nicotinic acetylcholine receptors have been found on T7 BCs[76].

D1 dopamine receptors are expressed at BC axon terminals in a type-specific manner in mice and rats[77,78]. Activation of D1 dopamine receptors has been shown to increase Ca$^{2+}$ levels through PKA-dependent enhancement of L-type Ca$^{2+}$ currents or PIP2-dependent Ca$^{2+}$ release from internal stores[79]. Ca$^{2+}$ current in OFF BC axons can be regulated by S-nitrosylation from retrogradely released nitric oxide[80]. While their molecular mechanisms and possible subcellular compartmentalization were not studied, dopamine has been shown to decrease surround suppression in fish BCs[81], and both agonists and inverse agonists of cannabinoid receptors have been shown to alter the surrounds of mouse ON alpha RGCs[82].

### Measuring functional divergence at the micron scale

While our interpretation is that functional divergence in BC axons can occur at the scale of tens of microns, our functional measurements were made at the scale of spikes and synaptic currents in RGCs. Functional imaging of calcium or glutamate with genetically encoded indicators could presumably offer more direct measurements at the micron scale. These techniques have been used for studying functional compartmentalization in retinal ACs[47,83,84], the dendrites of RGCs[85-87], and even in a recent paper that similarly reported divergence of a different function (direction selectivity) in T7 BCs[38].

While calcium and glutamate imaging techniques theoretically offer better spatial resolution, they are indirect measures of synaptic function and suffer from their own technical limitations. Calcium imaging revealed functional compartmentalization in A17 ACs where synaptic boutons are separated by ~20 μm sections of a single,

extremely thin (100 nm) neurite[47]. In contrast, a T6 BC axonal arbor has ~90 ribbon synapses all within a much smaller axonal structure which lacks the large separation of varicosities seen in the A17 AC (Fig. 7a). In addition to the ever-present issue of the nonlinear relationship between calcium changes and neurotransmitter release, the morphology of these axonal arbors make measurements of local calcium at the scale of individual synapses with a diffusible indicator infeasible.

Glutamate imaging enables a more direct measurement of the molecule driving postsynaptic conductance. Still, it suffers from a different kind of spatial localization problem: uncertainty about the origin of the glutamate. The sensor (iGluSnFR) is present throughout the membrane of each cell in which it is expressed. Thus, it lacks synaptic localization. Expressing iGluSnFR in RGCs could reveal postsynaptic compartmentalization, but it would not reveal whether nearby signals arose from the same or different BCs. Alternatively, expressing iGluSnFR sparsely in the BCs themselves, as was achieved for T7 BCs via subretinal viral injections[38], does not guarantee that the measured signals arise from the BCs in which the sensor is expressed given the extremely high density of glutamatergic synapses in the IPL. Of course, any imaging technique in the functioning retina also interferes to some extent with the light responses of the photoreceptors[88]. Laser-induced light exposure is especially problematic when attempting to compare responses to small spots of light within the imaging field (the scale of one or several BCs) to large spots of light that extend beyond the imaging field.

Instead of functional imaging, we used electrophysiology to ascertain the presynaptic origin of the divergence in surround suppression between Pix$_{ON}$ and ON alpha RGCs (Figs. 1-3 and Supplementary Figs. 3-10, 13, 14). We then used SBFSEM and confocal imaging to determine that these RGCs share input from the same set of BCs (Fig. 5 and Supplementary Figs. 11, 12). Thus, while we did not directly measure different functional signals within the axons of a single BC, we showed that the same population of BCs can drive differing excitatory conductances in downstream RGCs.

### Implications for visual processing in BCs

A decade ago, RGC spike recordings during current injections into single salamander BCs suggested that individual BCs could, at least indirectly, transmit different functional signals to different RGCs; however, it remained unclear to what extent postsynaptic mechanisms, ACs, or gap junctions were involved[89]. These authors and others[90] have speculated about the vast computational power of a neural network in which individual connections between neurons could have some degree of functional independence. Our results demonstrate that, indeed, one of the most canonical retinal computations, surround suppression, can manifest within a neuron (a bipolar cell) whose output synapses are, on average, less than 25 μm apart.

We focused on T6 BCs (Figs. 7, 8 and Supplementary Figs. 12, 15-18), but T7 BCs also provide a substantial input to Pix$_{ON}$ and ON alpha RGCs (Fig. 5h-j and Supplementary Fig. 11e-h), and functional divergence of direction selectivity in their axons has been measured by glutamate release[38]. Rather than an exception, functional divergence may be the rule in mouse (and perhaps other mammalian) BCs. Importantly, one cannot necessarily measure functional divergence with a single stimulus paradigm[38]. Since the difference we measured was in the degree of surround suppression, we would not have measured it with spatially uniform stimuli or when analyzing only a single spot size at a time. This could help explain the lack of evidence for subcellular processing in a previous study of mouse BC glutamate release[10], though the same researchers did find evidence for functional divergence in BCs with improved analysis methods[91].

Functional specialization at the subcellular scale is noteworthy in the context of interpretations of ultrastructural (connectomics) datasets, where the mouse retina has been a model for linking circuit structure to function[7,92,93]. SBFSEM reconstructions allowed us to

quantify BC inputs to $Pix_{ON}$ and ON alpha RGCs (Fig. 5) and to measure details of the locations of synapses (Fig. 6f and 7a), but the main conclusion of our study suggests that one should be cautious in interpreting similar patterns of synaptic connectivity as a proxy for function.

## Methods

### Ex vivo retina preparation

Animals were used and cared for in accordance with protocols approved by Northwestern University Institutional Animal Care and Use Committee. Mice aged 6 − 36 weeks were used for recordings and imaging. For experiments requiring labeled T6 BCs (Fig. 5 and Supplementary Fig. 12), CCK-ires-Cre/Ai14 mice were used (Jackson Lab Strain # 012706 / 007914). All other experiments used wild-type mice (C57BL/6, Jackson Lab Strain # 000664). Mice of either sex were used and sex effects were not analyzed. Mice were housed in a 14:10 dark-light cycle (14 h of darkness followed by 10 h of light). Experiments were performed during dark hours. Housing temperature ranged from 21 °C to 23 °C. Housing humidity ranged from 30% to 70%.

Whole mount retinas were prepared similarly to previous publications[8,94–99]. In short, dark-adapted mice were sacrificed, and retinas were dissected under infrared illumination (940 nm). The intact retina was flat-mounted photoreceptor side down on a poly-D-lysine-coated glass coverslip and placed in a recording chamber. Retinas were perfused with bicarbonate buffered oxygenated Ames medium (US Biological A1372-25) at 32 °C at a 10 mL/min rate throughout the experiment.

### Visual stimulation

Visual stimuli were generated with a 912 × 1140 pixel DLP projector (1.3 μm/pixel) at a 60 Hz frame rate using a blue LED (450 nm) focused on the photoreceptor outer segments. Light intensities are reported in rhodopsin isomerizations per rod per second (R*/rod/s). Visual stimuli had intensity values of 200-300 R*/rod/s and background intensity values of ~0.3 R*/rod/s unless otherwise noted (Supplementary Figs. 3, 10). Each cell's receptive field center was determined by flashing horizontal and vertical bars at different locations, and all subsequent stimuli were centered on the location that elicited maximal responses. Surround suppression was probed using a pseudorandom sequence of 12 spot sizes (diameters logarithmically spaced from 30-1200 μm), each presented for 1 second.

### Cell-attached and whole-cell recordings

All recordings were obtained using a 2-channel patch-clamp amplifier (Multiclamp 700B, Molecular Devices) sampling at 10 kHz. Spike trains were recorded using glass pipettes (2–3MΩ) filled with AMES solution in cell-attached configuration. Voltage-clamp recordings were performed using glass pipettes (4–6MΩ) filled with a cesium-based intracellular solution (105 mM Cs methanesulfonate, 10 mM TEA-Cl, 20 mM HEPES, 10 mM EGTA, 2 mM QX-314, 5 mM Mg-ATP, and 0.5 mM Tris-GTP; ~277 mOsm; pH ~7.32 with CsOH). The voltage was clamped to the equilibrium potential of chloride ( − 60 mV) to measure excitatory conductances or the reversal potential of glutamate-induced cation currents (+20 mV) to measure inhibitory conductances. A correction was not made for liquid junction potential (8.6 mV). Current clamp recordings and cell fills of neurobiotin were performed using glass pipettes (4–6MΩ) filled with a potassium-based intracellular solution (125 mM K-aspartate, 10 mM KCl, 1 mM MgCl2, 10 mM HEPES, 1 mM CaCl2, 2 mM EGTA, 4 mM Mg-ATP and 0.5 mM Tris-GTP; ~277 mOsm; pH ~7.15 with KOH).

### Dynamic clamp recordings

Dynamic clamp hardware and software were implemented as described in Desai et al. (2017[100]).

$Pix_{ON}$ and ON alpha excitatory and inhibitory conductances were recorded in response to 200, 600, and 1200 μm diameter spots of light. New RGCs were then patched in whole-cell current-clamp configuration, the previously recorded conductances were simulated via current injections, and the resulting spike train was recorded.

Before the start of each experiment, a scaling parameter was chosen for the conductances. The scaling parameter was multiplied against the recorded conductances to give a scaled version for simulation. The same scaling parameter was used for both excitation and inhibition. This scaling parameter was chosen by testing a range of scaling parameter values (0.4–2.5) while simulating the conductances recorded from a 200 μm spot illumination. We then chose the scaling value that evoked the number of spikes most similar to what was recorded for that same cell type during real visual stimulation.

For Fig. 2b–g, the paired excitatory and inhibitory conductances were always derived from the same size stimuli (e.g., if simulating excitation evoked by a 200 μm spot, inhibition evoked by a 200 μm spot was simultaneously simulated). After simulating 200 μm, 600 μm, and 1200 μm conductances, the "Preferred size" response was taken as whichever of these conductance sets elicited the largest spiking response. "Full-field" responses were taken as the spiking response when simulating excitation and inhibition recorded during 1200 μm spot stimuli.

For Fig. 2h, "$Exc_{pref}$" and "$Inh_{pref}$" refer to the $Pix_{ON}$ excitatory and inhibitory conductances that were found to elicit the maximal spiking response (see Fig. 2b). Conversely, "$Exc_{ff}$" and "$Inh_{ff}$" refer to the $Pix_{ON}$ excitatory and inhibitory conductances recorded while presenting a 1200 μm diameter light spot.

### Pharmacology

Intrinsic light responses were measured in both $Pix_{ON}$ and ON alpha RGCs (Supplementary Fig. 1j) by providing full-field light stimuli while voltage clamping at −60 mV during bath application of L-AP4, DNQX, and D-AP5 to block photoreceptor-driven light responses[101].

See Supplementary Table 1 for a complete listing of pharmacological agents and their targets.

### Physiology analysis

RGC spiking responses were measured as the average spike rate during the 1-second light stimulus. RGC conductance responses were measured as the total charge transfer during the 1-second light stimulus. The preferred size response ($R_{preferred\ size}$) was defined as the maximal response measured during the presentation of all sizes of spot stimuli (30−1200 μm diameter). The full-field response ($R_{full-field}$) was defined as the response recorded during the presentation of the largest stimulus spot (1200 μm diameter). Suppression was calculated as:

$$\text{Suppression} = 1 - (R_{full-field}/R_{preferred\ size}) \qquad (1)$$

### Two-photon imaging

RGCs were filled with AlexaFluor 488 (0.2 mM) via a whole-cell patch pipette. Images were collected through a ×60 water immersion objective (Olympus LUMPLan FLN 60x/1.00 NA) using 980 nm two-photon laser excitation (MaiTai HP, SpectraPhysics). Volume images were captured with 0.5 μm z-steps with enough z-slices to capture the entire dendritic arbor and soma. Red and green channels were split with a 565 nm dichroic mirror and bandpass filtered with red (Omega, 595AF60) and green (Chroma, ET52550 M-2P) notch filters respectively.

### Confocal imaging

For experiments requiring high-resolution images of the RGC dendritic arbors (Supplementary Fig. 1a–h) or immunohistochemical labeling of proteins (Fig. 1b, Supplementary Fig. 1a, i, and Supplementary Fig. 12), RGCs were filled with Neurobiotin tracer (Vector Laboratories, SP-1150, ~3% w/v potassium-based internal solution) and fixed in 3%

paraformaldehyde solution for 15 min. After performing immunohistochemical labeling and incubation with streptavidin (see "Immunohistochemistry" below), tissues were imaged on a Nikon A1R laser scanning confocal microscope through a ×40 or ×100 oil immersion objective (Nikon Plan Apo VC × 40/ × 60/1.4 NA).

## Immunohistochemistry

Retinas were fixed at room temperature for 15 min in 3% paraformaldehyde in 0.1 M phosphate buffer (Electron Microscopy Sciences) and then blocked at room temperature for 2 h in 3% Normal Donkey Serum (Jackson Labs) and 0.5% Triton (Sigma) in 0.1 M phosphate buffer.

Retinas were then incubated with primary antibodies for five days at 4 °C. After washing, retinas were incubated with secondary antibodies for two days at 4 °C. All antibodies were diluted 1:500. Retinas were then mounted on glass coverslips using Vectashield Antifade (Vector Labs). See Supplementary Table 2 for a complete list of antibodies used.

## Quantification of RGC morphology

From both two-photon and confocal images, soma diameter was calculated by tracing an outline of the soma using 'Freehand Selections' in FIJI to calculate soma area and then solving for diameter with the assumption of a circular soma (area = $\pi * r^2$). Similarly, the dendritic field diameter was measured by drawing a convex polygon around the tips of the dendrites in a flattened view of the image (maximum z-projection) and calculating the diameter from the area encompassed by the dendritic arbor. Average branch length (distance between branching nodes), number of branches, and total dendritic length were calculated by tracing the RGC dendrites using the SNT plug-in in FIJI and its built-in analysis tools[102].

Stratification analysis was performed by measuring dendrite depth in the IPL in relation to the immunohistochemically labeled ChAT bands (starburst AC neurites). Custom MATLAB software[94] based on a published algorithm[103] was used to flatten the image prior to analysis.

M5 and M4 RGC morphological data were generously provided by Professor David Berson and were published in Stabio, et al.[101]. and Estevez et al.[27], respectively.

## PSD95 puncta analysis

PSD95 puncta images were thresholded in FIJI using Otsu Auto Local Thresholding (radius 15 μm)[104]. Individual PSD 95 puncta were then identified using FIJI's built-in 3D object counter tool[105]. RGC dendritic arbors were traced and filled with the SNT plug-in in FIJI[102]. PSD95 puncta were ascribed to the RGC if at least 90% of its volume was contained within the filled volume of the RGC dendrite. Each PSD95 puncta was then manually assessed to determine if it was directly apposed in 3D space to a CCK labeled T6 BC axonal process. Control analysis were performed in which the PSD95 puncta image channel was rotated 90° compared to the T6 BC image channel. The identity of experimental vs. control images was obfuscated to provide a double-blind analysis.

## Correlative fluorescence and serial block-face scanning electron microscopy (SBFSEM)

Neighboring Pix$_{ON}$ and ON Alpha RGCs with overlapping dendritic arbors were physiologically identified in a mouse line with fluorescently labeled T6 BCs (CCK-ires-Cre/Ai14[32,33]). After verifying that the two ganglion cells had differing levels of surround suppression in their spiking response, they were filled with Alexa 488. Two-photon volume images of the RGCs overlapping dendrites and the T6 BC axonal arbors were then acquired (see Methods: Two-photon imaging). The retina was fixed with 1.5% glutaraldehyde and 2.5% paraformaldehyde in 0.1 M Na$^+$ Cacodylate buffer for 10 min. The retina was washed with 0.1 M Na$^+$

Cacodylate buffer and transferred to 4% glutaraldehyde for 4 h at 4 °C to further fix the tissue.

We utilized the previously published near-infrared branding technique[106] to burn fiducial markers into the retina with the two-photon laser (860 nm, ~100 mW), allowing for the alignment of two-photon images with electron microscopy volumes. The tissue was prepared for SBFSEM according to the protocol described previously[107]. Image stacks were acquired using a VolumeScope SEM (Apreo, Thermo Fisher Scientific) at a voxel size of $5 \times 5 \times 50\,nm^3$.

## Volume reconstruction and image analysis

SBFSEM image stacks were aligned and registered using ImageJ/TrakEM2[108]. The neuronal processes were traced and segmented using AreaTree or AreaList function, whereas synapses were segmented using AreaList function in TrakEM2. The 3D objects of either traced skeletons or surface segmentations were visualized in either 3D view in ImageJ or exported to and rendered in Amira (Thermo Fisher Scientific).

The somata of both Pix$_{ON}$ and ON alpha RGCs were located according to the fiducial markers. We traced the dendritic arbors of both RGCs within the limit of the SBFSEM volume. BC synapses were identified by the presence of a presynaptic ribbon apposed to the postsynaptic dendrites of both RGCs (Supplementary Fig. 19). Presynaptic AC contacts were identified by the presence of clusters of synaptic vesicles apposed to BC axons. Presynaptic BCs were reconstructed, and their type was determined according to their stereotyped morphology[109–111]. T6 BCs were further confirmed by the presence of the fluorescence marker in the corresponding 2-photon volume.

We identified all the presynaptic inhibitory sites on the BC axon segment where the ribbon synapses resided. For each ribbon, the Euclidean distances between the ribbon and all the presynaptic inhibitory sites were measured, and the inhibitory synapse with the shortest distance was identified.

## BC receptive field model

We modeled RGC excitation across spot sizes as the summation of BC subunits sampled across the RGC's dendritic arbor. To do this, a skeleton of the RGC's dendritic arbor was provided to the model, and excitatory input synapses were randomly assigned along the length of the dendritic skeleton (0.3 μm / synapse[31,112]). A BC was assigned to each synapse, and its receptive field was centered on that synapse.

RGC excitatory responses across spot sizes were predicted by presenting virtual spots of multiple sizes centered at the centroid of the ganglion cell dendritic field and calculating each BC's activation as the overlap of its receptive field with the presented spot. RGC excitatory conductances were then modeled as the linear sum of each BC's activation. Both experimentally measured and model-predicted excitatory responses were normalized across spot sizes by the maximal response.

The BC receptive field was modeled as a circular difference of Gaussians[113] with three parameters; center size ($\sigma_c$), surround size ($\sigma_s$), and center-to-surround ratio (CSR). While $\sigma_c$ was fixed at 22 μm [ref. 31], $\sigma_s$ and CSR were obtained by minimizing the mean absolute error between the model output and the experimentally recorded RGC excitatory responses across all spot sizes. Error was minimized using the Interior-point optimization algorithm, and initial values of 100 μm for $\sigma_s$ and 1 for CSR. 6 RGCs were simultaneously fit for each estimation of $\sigma_s$ and CSR. Cross-validation was performed on the remaining RGCs. When fitting to both Pix$_{ON}$ and ON alpha RGCs, three Pix$_{ON}$ and three ON alpha RGCs were used for fitting. Four hundred random fitting combinations of the 14 Pix$_{ON}$ and 8 ON alpha RGCs were performed to obtain average cross-validation values.

The model was written using MATLAB 2022a. Code and data are available for download[114].

## NEURON compartment model of a T6 BC

Cable modeling was performed using Python 3.8 and NEURON 8.0[37]. SBFSEM reconstructions were imported to NEURON using NEURON's *Import3d* tool. 91 ribbon output synapses were identified in the SBFSEM reconstruction, but some of these ribbons were located on the axon stalk near the center of the IPL. To restrict our analysis to those ribbon synapses that might actually synapse onto Pix$_{ON}$ and ON alpha RGCs, we measured the path distance from the first T6 BC axon branching point to T6-Pix$_{ON}$ and T6-ON alpha ribbon synapses in the SBFSEM reconstruction from Fig. 5. We then restricted our model analysis to ribbons whose path distances fell within the 99% confidence interval of these measurements (84 ribbons).

Excitation was simulated through the stochastic activation of 8 excitatory synapses at the BC dendrites. The excitatory synapses were modeled with the built-in NEURON point process, Exp2Syn, a two-state kinetic scheme synapse described by rise time (10 ms[115]) and decay time (100 ms[115]). The passive resting membrane potential was −60 mV as set by passive leak channels. However, for all simulations, a "Dark current" was provided by activating the excitatory synapses to push the membrane potential of the ribbon synapses to their expected value of −45 mV[72]. To simulate light-evoked activation, the rate of stochastic events was increased onto the excitatory synapses to depolarize the axonal arbor to around −30 mV[72,116].

Inhibition was simulated through the stochastic activation of some or all of the 120 inhibitory synapses located on the BC axonal arbor. The inhibitory synapses were modeled with the NEURON Exp2Syn point process (rise time = 1.8 ms, decay time = 100 ms[117]). Inhibition was only provided during the simulation of light activation and always coincided with excitation, as described in the previous paragraph. The stochastic event rate of the inhibitory synapses was set so that the first quartile of ribbons exhibited CSR values of 1.1 (−44 mV). This value was chosen to match the value fit by the BC receptive field model (Fig. 4b, c).

All simulations were given 500 ms to reach a steady state before taking any measurements. When measuring the effect of a single inhibitory synapse (Fig. 7d), CSR was measured for all 84 ribbon output synapses when activating each of the 120 inhibitory synapses (a new simulation was performed for each inhibitory synapse). When testing the simultaneous activation of inhibitory synapses (Fig. 7e–g), the same 120 inhibitory synapses were sequentially activated, but the additional N-nearest inhibitory synapses (by path distance) were also simultaneously activated.

Key model parameters can be found in Supplementary Table 3. Model stability was tested by measuring the CSR range across a range of key model parameters (Supplementary Fig. 15). Whenever model parameters were altered, excitatory and inhibitory conductances were adjusted to maintain the same membrane potentials (−45 mV resting, −30 mV during excitation, and CSR of 1.1).

Code and data are available at github.com/davidswygart/T6_NEURON_python[118].

## Statistical tests

Statistics and data representation are reported in figure legends. In short, data are reported as mean ± standard error of the mean, unless otherwise noted. Differing means were assessed with Welch's *t* test for unpaired data, paired two-sample Student's *t* test for paired data, and two-way ANOVA for multivariate data. Comparisons of proportions were assessed with a two-proportions *z*-test with Holm-Bonferroni correction. Differing continuous distributions were assessed with Kolmogorov–Smirnov tests.

## Reporting summary

Further information on research design is available in the Nature Portfolio Reporting Summary linked to this article.

## Data availability

Source data are provided with this paper. Raw electrophysiology data have been deposited in the Mendeley database[119].

## Code availability

Physiology data was collected using custom code written in MATLAB 2018b. Code is available at github.com/Schwartz-AlaLaurila-Labs/sa-labs-extension[120]. Physiology data was analyzed using custom code written in MATLAB 2022a. Code is available at github.com/SchwartzNU/SymphonyAnalysis[121]. The BC receptive field model (Figs. 4 and 8) was written using MATLAB 2022a. Code and data are available for at github.com/davidswygart/rgc_bipolar_dog[114]. Cable modeling of the T6 BC (Fig. 7 and Supplementary Figs. 15–18) was performed using Python 3.8 and NEURON 8.0[37]. Code and data are available at github.com/davidswygart/T6_NEURON_python[118].

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

## Acknowledgements
We are thankful to all Schwartz lab members for their feedback and technical assistance throughout the project. We would like to thank Tiffany Schmidt and Anna Vlasits for their feedback and comments on the manuscript and David Berson for sharing M4 and M5 morphological data[27,101]. Funding for this research was provided by National Institutes of Health grants F31EY030344 (D. S.), EY10699 (R.W.) and R01EY031029 (G. W. S.). Additionally, S. Takeuchi was supported by the Graduate Research Abroad in Science Program (GRASP) of The University of Tokyo and the Graduate Program for Leaders in Life Innovation (GPLLI).

## Author contributions
D.S. and G.W.S. designed the experiments. D.S. performed experiments and analyzed results related to electrophysiology, flourescent imaging, and mathematical modeling. W-Q.Y, S.T., R.O.W. performed experiments and analyzed results related to Serial Blockface Electron Microscopy. D.S. and G.W.S. wrote the manuscript with feedback from W-Q.Y., and R.O.W.

## Competing interests
The authors declare no competing interests.
