## [Peer Review File · Nature Communications]

REVIEWER COMMENTS

Reviewer #1 (Remarks to the Author):

D Swygart, W-Q Yu, S Takeuchi, RROL Wong & Schwartz GW. "A presynaptic source drives differing levels of surround suppression in two mouse retinal ganglion cell types". Nature Communications (NCOMMS-22-46416-T).

In this extremely interesting paper, Swygart et al. address the possibility that release of glutamate from the large number of presynaptic ribbon synapses within a single cone bipolar cell axon terminal (in the inner plexiform layer) can be differentially modulated by inhibitory input from GABAergic amacrine cells targeting the same axon terminals. This hypothesis, with local inhibitory control of release, is contrasted with an alternative mechanism where output is modulated in a global way, with little or no local control. Because of the small size of a bipolar cell axon terminal, many workers in the field would presumably have guessed that there would be little chance for the presynaptic voltage to vary sufficiently at different locations of a cone bipolar cell axon terminal. This is a very interesting and important topic for retinal signal processing, potentially with implications for sensory and neural processing in general.

In their study, Swygart et al. used a number of experimental techniques:

whole-cell electrophysiological recording (including dynamic clamp) and visual stimulation of ganglion cells in a whole-mount preparation of rat retina, 2P microscopy, immunocytochemistry and confocal microscopy, ultrastructural imaging in the form of serial block face scanning electron microscopy (SBFSEM), and computational modeling.

In general, the conclusions and claims made in the paper are strongly supported by the experimental work. The methodology is sound in relation to the questions addressed. I do believe, however, that the authors need to address a few important questions concerning the feasibility of the proposed mechanism (see below). There are also some issues concerning the data analysis and presentation that should be addressed. Some of the points I raise may possibly be fixed with a careful revision, potentially with additional simulations. In other places, I believe that the authors should provide additional details for the work to be reproduced (see below).

MAJOR COMMENTS / CRITICISMS:

1. In their discussions of "electrotonic compactness", the authors need to disentangle several related issues. For a passive compartmental model, the degree of electrotonic compactness can be precisely defined, preferably for a series of frequencies, including the DC level which is what the authors seem to address in their illustrated examples. In the Introduction, line 38-39, they refer to the supposedly commonly held view of bipolar cells as "electrically compact neurons". The implicit message here is that perhaps bipolar cells are not electrotonically compact, or at the very least that this (commonly held) idea somehow has to be modified. In the remaining part of the paper, the authors do not bring up this issue again in a clear way, but their "solution" to the problem is that a different behavior (with respect to

synaptic output) is found in their active models of bipolar cells. The authors do indeed acknowledge this (Results, line 327), but as a whole, the treatment is a bit unsatisfactory, as it does not make clear whether this change in behavior is a result of the dynamic properties of the added ion conductances or a change that for all practical purposes appears as a change in passive membrane properties. This needs to be addressed, potentially by additional simulations or analysis.

More specifically, on page 21 (line 440->), the authors state explicitly that the results observed for the simulation with the active compartmental model might be explained by the influence of the added (active) ion conductances on the (effective) membrane resistance. To clarify their arguments, the authors need to state clearly how the conductances were added to the model, i.e., was the density the same in all compartments? Did the introduced channels change the effective resistance “at rest”? Have the authors investigated if similar results can be obtained simply by reducing the membrane resistance (to values that are still realistic)?

2. In Figure 5f (and the accompanying) text in the Results section, the authors illustrate the distance from ribbon synapses to the nearest inhibitory synapse.

- first, why do the authors believe that measuring the Euclidean distance is the correct / relevant measure here? Cf. Methods (line 671-672). Would not path distance be more relevant in the context of functional interaction?

- second, the authors make it a point that the difference between Pix_ON and ON alpha was statistically significant. However, although statistically significant, the difference seems very small and there is considerable overlap. Please discuss the functional importance of the difference (potentially for path distance, see above).

3. The explanation of the analysis performed of the extent of hyperpolarization induced by inhibitory inputs and the extraction of top and bottom quartiles (Q1 and Q4) is very confusing. It seems to me that part of the confusion is due to inconsistent description of what “top” and “bottom” quartiles refer to. Is “top” Q1 and “bottom” Q4? Or is it the reverse? The legend of Figure 6 calls Q1 top and Q4 bottom, but the legend of Figure 7 calls Q4 top and Q1 bottom. Figure 6c-e illustrates that the values for Q1 are smaller than the values for Q4. The legend for Figure 7f refers to the decrement of average hyperpolarization from Q2 to Q1, but the Y axis label indicates “Q4 to Q1”. See also the text on page 16, Figure 6 (compare panels c-e with the text in the legend), and text on page 28 (lines 713-716).

4. The details of the compartmental simulations of the interaction between excitatory and inhibitory inputs in the bipolar cell (axon terminals) are a bit confusing:

- it should be clearly stated if all measurements of inhibitory hyperpolarizations were made with reference to (relative to) a prior excitatory input (incl. Methods, line 711)

- the way I understand it, all inhibitory synapses had their strength adjusted to induce a local hyperpolarization to -45 mV. How can the authors justify this and what are the consequences for the rest of the analysis? Does this not amount to a “correction” of the associated conductance such that synaptic input at a location with low(er) input resistance is automatically increased to compensate for the higher

dissipation of the evoked voltage? Perhaps the simulations should instead vary the inhibitory conductances randomly for the different locations?

- for Figure 6b-e: why does the inhibitory hyperpolarization never go more negative than -45 mV, given the E_{Cl} was set to -50.5 mV (according to Table 3)? Is this a result of the scaling of the inhibitory conductances?

-how was the excitatory input modeled? As a fixed current or as a fixed conductance? As the current/conductance required to depolarize to -35 mV irrespective of other changes to the cell? What are the consequences of the strategy chosen for the results obtained?

5. Discussion, line 495-499: Here the authors discuss the morphology of the reconstructed type 6 bipolar cell and the possibility of measuring local Ca^{2+} at the scale of individual synapses. It is not my intention to argue that the authors should attempt measurement of local Ca^{2+} for the current paper, but I think their discussion should be modified by taking the following into account. They argue (line 496) that the reconstructed axon terminals constitute “a compact 3D structure mostly lacking anatomical compartmentalization”. There are several problems with this statement. First, it seems inconsistent with other reports of the morphology of cone bipolar cell axon terminals that have described discrete varicosities (Golgi, fluorescence, EM). Does this mean that the reconstructed cell is an atypical cone bipolar cell? Second, in a different place in the paper (Methods, line 670), the authors describe the reconstructed axon terminals in different terms, using the words “...all the presynaptic (amacrine) inhibitory sites on the bipolar cell bouton...”, suggesting that anatomical compartmentalization in the form of “boutons” was indeed present. Perhaps the authors need to clarify their use of the term “anatomical compartmentalization”?

6. When an increasing number of inhibitory synapses were activated, the authors stated that they added a number N of nearest inhibitory synapses (by path distance). I am not sure I understand the motivation or necessity of this approach. What would the results be if one instead added a selection of N [1 - 119] extra inhibitory synapses randomly selected from the total population?

7. The authors suggest a differential targeting of ribbon synapses within single cone bipolar axon terminals onto the retinal ganglion cells, depending on their modulation (weak vs/ strong) by inhibitory input from amacrine cells. Can the authors use their SBFSEM data to examine whether the ribbons with the largest / smallest hyperpolarizations differentially (preferentially) target the different types of retinal ganglion cells?

8. From the figures included in the paper, it seems to me that some synapses (ribbon, amacrine) were securely identified, but some of the examples included in the figures are not very convincing. I acknowledge that this can be related to the low resolution in the figures provided for review, but it can also be a problem with the overall lower resolution of SBFSEM images and potentially raises a question of the reliability of the quantitative data generated from these experiments. See specific comments below (FIGURES).

9. In the leftmost part of Figure 5a, we can see that the experiment with combined application of TPMPA and TTX was only performed for two cells. Although the suppression is very strong, it would definitely be preferable to see the result reproduced for a few more cells.

10. It is easy to fix, but there are major problems with lack of consistency between figure panels and legends / text (see below under "FIGURES").

MINOR CRITICISMS:

1. For the morphological results, it is a problem that several figures / panels are very small. For some of these illustrations, larger figures with much higher resolution would have made the evaluation easier. As is, some figures / panels with morphological data (e.g. immunolabeling, reconstructions) mostly have a decorative value. To the extent permitted by the journal, this should be improved.

2. Abstract, lines 17-18: The sentence "Our results show that divergence in the level of surround suppression occurs subcellularly, at bipolar cells synapses" is difficult to understand. Perhaps it would be clearer if "divergence" was replaced with e.g. "variability" or "heterogeneity"?

3. Introduction, line 30-31: The sentence "Nonetheless, the number of functionally distinct RGC types exceeds their stratification diversity" is misleading. Nobody believes that it is the stratification per se, which essentially is just an epiphenomenon, that is the crucial mechanism for differential signal processing. The stratification level provides an opportunity for engaging processes of other cells in synaptic circuits, which directly links to the authors' subsequent arguments. The main result of the current study is that the potential for differential processing goes even further, with mechanism(s) that might permit differential output from within a single bipolar axon terminal.

4. Introduction, line 35: Here and elsewhere (e.g. line 62), the authors need to tighten up their terminology. The terms "sublamina" / "sublaminae" refer to sublamina a (sclerad 2/5 of the inner plexiform layer) and sublamina b (vitread 3/5 of the inner plexiform layer). The mammalian inner plexiform layer is generally divided into five equally thick strata, termed stratum 1 - stratum 5 (S1 - S5). For the example in line 35, "sublamina" should be replaced with "stratum".

5. In some places, e.g. Introduction (lines 38-39, line 51), Discussion (lines 411), the authors contrast their own idea / speculations with what they term e.g. "a commonly held view". This can be a useful "rhetorical strategy", but for it to be more than a straw man argument, the authors need to provide examples of references such that we (as readers) are convinced that the views indeed are (fairly) commonly held. Similarly, if the authors want to argue that "bipolar cells are assumed to be isopotential" (line 411), they should provide a couple of references where this is stated explicitly.

6. The authors use both the term "electrical" and "electrotonic" in relation to compactness. I would suggest that they stick to one expression.

7. Introduction, line 42: To state that "surround suppression" is one of the "oldest" visual computations is unclear. Do the authors mean "oldest" in the sense of "phylogenetically oldest" or in the sense of "classical(ly studied)".

8. Results, line 61-63: The text refers to visual responses ("ON-sustained"), but Fig. 1a,b does not show visual responses.

9. Results, line 75: Should “Fig. 1d,e” be “Fig. 1c-e”?

10. Results, line 84: Here and elsewhere (e.g. line 674, 680), clarity could be improved when the authors use the word “across”, e.g. as in the phrase “across stimulus size”, by re-writing e.g. as “across a range of stimulus sizes”.

11. Results, line 84-87: The authors refer to published results (ref. 12) concerning spatially distinct regions where PixON retinal ganglion cells receive excitation and inhibition, but do they argue that this is a precondition for the ability of voltage-clamp recording to separately record excitatory and inhibitory inputs arriving in the dendritic tree? Please clarify.

12. Results, line 102: Here and in several other places (e.g. line 266), the authors need to increase the level of precision when they use the phrase “surround suppression”. Please be consistent and always make it clear if the phrase refers to spiking or conductance (excitatory / inhibitory).

13. Results, line 110: I do not think that the sentence that starts “To independently test the ability of...” is correctly structured. I suppose that what the authors intend to say is that they wanted to test the ability of an injected excitatory conductance to drive surround suppression of spiking, independently of an inhibitory conductance. Please clarify.

14. Results, line 131: I think it would be clearer to rearrange the sentence to read “Differing levels of surround suppression of excitatory conductances between the...”.

15. Results,

- line 133-135: Here the authors speculate how different size stimuli might impact the state of saturation / desensitization of glutamate receptors on retinal ganglion cells. To make the analysis / arguments easier to follow, please it clear if the authors believe that the excitatory drive (conductance) would be reduced for larger spot sizes. What about invoking a receptor-dependent mechanism for the stronger surround suppression seen for the Pix_ON retinal ganglion cells? Could that work?

- line 142: please restate argument / logic briefly, instead of just providing a reference to earlier work.

- line 145: The way the authors have constructed the text within the parenthesis, “Kyn” and “NBQX” appear as subjects with “suppressed” as verb. I do not think this is the intention. Please re-write / clarify.

16. Results, line 164: Here and elsewhere, the authors use the term “subunit”. As far as I know, this term has its origin as an abstract entity in relation to studies of receptive field properties, but in this paper the authors use it sometimes to refer to an abstract entity and sometimes to a bipolar cell (as a functional term?). As a whole, the text can be a bit confusing and I would encourage the authors to rewrite to make it more stringent and easier to understand.

17. Results, line 169-170: Please re-write to make it clearer what “differences” in “...differences in their dendritic arbor...” might refer to, i.e., what kind of potential differences do the authors have in mind?

18. Results, line 172-173: Please clarify the argument involving the phrase “...which could cancel out BC subunit inhibitory surrounds over a larger area”. Perhaps re-writing with a proposed example could improve the understanding.

19. Results, line 176: It is not (entirely) clear what “These two variables” refer back to. Please clarify / re-write.

20. Results, line 177: I think it would be good if the authors could make it clear when writing that they “modeled ... conductances” if this involved a compartmental model or not. Please think of a way to make this clearer in the main text, i.e., which of the several modeling techniques a specific instance of “modeling” refers to.

21. Results, line 204: The phrase “...did not result in converging...” is unclear. I guess that what the authors intend to say is that the results from fitting remained different, but the use of “converged” is suggestive of some kind of reiterative fitting procedure. Please clarify.

22. Results, line 230 and rest of paragraph: Please clarify what the reported percentages are a percentage of.

23. Results, line 265: I think it is better to write “GABA or glycine receptor antagonists” as the different types of receptors were not blocked simultaneously?

24. Results, line 274-276: Here and elsewhere (Discussion), how can the authors explain that the GABA_C receptor antagonist TPMPA alone was not as efficient as when combined with TTX? Can it be explained by incomplete antagonism with the concentration of TPMPA used?

25. Results, line 279-283: This section is very clearly written. I suggest using some of these phrases to improve the corresponding part of the Abstract, which I found much less clear.

26. Results, line 292-293: Please state the (numerical) ranges for each statistic reported here (synapse distances).

27. Results, line 306-307: Something is wrong with the grammar in “alpha synapses” in line 307. Perhaps it is better to re-write in less “interpretative” terms, e.g. by something like “A synapse made by the amacrine cell constituted the nearest neighbor for ribbon synapses onto both types of RGCs”?

28. Results, line 311: I think it might be better to write “...within the same BC axon terminal...”.

29. Results, line 329: Here and elsewhere, the authors often use the notation “HCN_2” for HCN2 channels. As used in some places in the paper, HCN2 is the standard way to refer to this channel type. Please modify accordingly, here and elsewhere.

30. Results, line 325-328: It is a bit misleading when the authors write that “multiple studies have measured voltage-gated ion conductances from bipolar cells” but provide a series of references to support this statement (ref. 32-35), only one which (ref. 32) is an experimental study. Please re-write and clarify, ideally by citing studies where bipolar cells conductances were measured experimentally.

31. Results, line 348: When referring to recruitment of inhibitory synapses, the authors sometimes refer to the “n-nearest” and sometimes to the “N-nearest”. Please be consistent.

32. Results, line 364: The phrase “...voltage decay of the BC’s inhibitory surround...” seems like lab jargon. Please re-write / clarify.

33. Results, line 393: I guess “independently” should be “independently of it”?

34. Discussion, line 413-414: I am unsure how relevant it is for the context of bipolar cell axon terminals to refer to the paper by Linden (ref. 43) as his discussion referred to much larger axonal trees where the crucial idea is conduction block of action potentials, e.g. at branch points.

35. Discussion, line 430: I suggest changing “electrotonic isolation” to “electrotonic compartmentalization” to make the point more general.

36. Discussion, line 437: It seems misleading to write “...a cell type typically modeled as passive and isopotential”. To model as passive is an explicit choice, but to model as isopotential is a conclusion from modeling, not a choice that can be made before modeling starts.

37. Discussion, line 439: In the context of the terminology appropriate for the compartmental modeling, it is not correct to write that the inclusion of active conductances decreased the membrane resistance. The membrane resistance specifies a passive membrane property. Perhaps it is better to refer to the “input resistance”? Or the “apparent membrane resistance” that would be estimated if an active conductance was active at the membrane voltage at which membrane resistance would be estimated?

38. Discussion, line 458: In “these barriers to Ca²⁺ diffusion”, the phrase “these barriers” refers back to words that are not included in the sentence. Please re-write / clarify.

39. Discussion, line 501: “the molecule driving postsynaptic current” should be changed to “the molecule driving postsynaptic conductance”.

40. Discussion, line 518: The statement that the authors “showed that different glutamate release profiles are experienced by RGCs...” seems a bit too strong. It is an interpretation, not a direct demonstration, as no direct measurements of glutamate release profiles were made. Please improve.

41. Discussion, line 534: It is unclear to me if the citation of the paper in ref. 30 is made to support the statement in the corresponding sentence or not? Please clarify the intended meaning here.

42. When discussing dendritic branches, the authors need to clarify if they mean e.g. “branch segments” (between nodes) or the whole branch from origin at soma to the dendritic tip(s).

FIGURES

1. Figure 1:

- the annotation for panels c, f and i could preferably be moved to make it immediately clear that it pertains to both the left and the right graphs in each panel (row)

- in panels g and j, the Y axis labels could be made easier to understand by changing the unit from “delta nS” (ΔnS) to “nS” and by changing the text from “Excitatory / Inhibitory conductance” to “Delta Excitatory / Inhibitory conductance”

(“ Δ Excitatory / Inhibitory conductance”)

- in the legend for panel c, change “Example peristimulus time histograms to...” to e.g. “Example peristimulus time histograms in response to...”.

2. Figure 2:

- the panel lettering does not match the lettering in the legend and in the main text

- the legend for panels b,c states “...histogram recorded...” but it should be “...histograms recorded...”

- to make panel “h” (erroneously labeled “f” in the figure itself) easier to grasp quickly, consider changing the annotations “Exc_ff” and “Inh_ff” to “Exc_full” and “Inh_full”. This style also better parallels that used for “Exc_pref” and “Inh_pref”
- for panels e-g the legend reads “e-g Same as b-c”. This is awkward, as a consecutive series of three panels does not match two consecutive panels. Consider to re-write as “e-g Same as b-d”
- the legend for panel h states “...conductances as to isolate...” but I believe it should read “...conductances to isolate...”
- the legend for panel i states “...when switching to full-field excitation...” but it is less clear what the switching was from, please clarify

3. Figure 3:

- panel a as labeled in the figure is not included or commented on in the legend or the main text
- panels b-g as labeled in the figure are wrongly labeled in relation to the legend and the main text
- please clarify how the conductances illustrated in panel c (erroneously labeled d in the figure) were measured (peak?)

4. Figure 4:

- panel c: is the region displayed in this panel taken from the region displayed in b? Please clarify and modify panel b accordingly
- panel c: the annotation indicates that ribbons are supposed to be labeled red, but apart from the labels “1” and “2”, it is really hard to see any clearly labeled (red) ribbons
- panel c: what is the significance of the “grayish” and “whitish” regions of the reconstructed bipolar cell axon terminal?
- panel e: is this really a ribbon?
- panel f: is the region displayed in this panel rotated or flipped relative to the corresponding region displayed in panel c? Please clarify
- panel g: where is the ribbon?

5. Figure 5:

- the legend for panel a states “antagonists of” but includes “TTX” at the end of the sentence. However, TTX is a blocker, not an antagonist, of Nav channels (there is no receptor and no [endogenous] agonist). Please re-write

- the legend for panel a states that the conductance trace in TPMPA+TTX was shifted down 2 nS. I assume this only refers to the trace for the full-field stimulus? Please clarify
- panel c: here the indicated ribbon is very convincing, but the (inhibitory) synapse supposedly made by the amacrine cell onto the bipolar axon terminal is much less convincing. Please clarify / comment

6. Figure 6:

- the legend for panel b (and elsewhere) uses the phrase “voltage trace of ribbons”. This is lab jargon, please re-write more precisely, perhaps “voltage response recorded at location of synaptic ribbon” is better?
- in panel b, why does the membrane potential before onset of the excitatory conductance reside at -45 mV, given that the reversal potential of the leak conductance was set to -60 mV (Table 3)?
- in panel c, does “1 inh. activated” refer to the same inhibitory synapse for all simulations, i.e., all ribbons, or does the specific inhibitory synapse activated vary according to the ribbon? Please clarify in figure itself and/or in legend and text

7. Figure 7:

- in the legend title, should “inhibition on” be “inhibition of”?

METHODS:

1. Line 574: the sign of the liquid junction potential does (presumably) not follow the convention of E. Neher. It should probably be +8.6 mV, not -8.6 mV. This is important if the correction involves subtraction or addition of the liquid potential.
2. Line 575: “reversal potential of chloride” should be “equilibrium potential of chloride”.
3. Line 579: “77 mOsm” should presumably be “277 mOsm”, or perhaps another value (a bit odd if exactly identical to the value for the other intracellular solution)?
4. Line 582: Desai et al. is not in the list of references. The hardware described in Desai et al. is by now a bit old and, if relevant, it would be useful if the authors in addition briefly described newer hardware used in their own study.

5. Line 588: What does “best reproduced” mean in terms of the method used? As determined by eye or was a re-iterative fitting procedure used?

6. Line 592: Does “maximal spiking response” refer to the peak firing rate? How was it determined and how many repetitions were averaged?

7. Line 602: I suggest reminding the readers here, using a few words, that the ganglion cells studied are intrinsically photosensitive.

8. Line 618: What was the paraformaldehyde dissolved in?

9. Line 626: It is a bit difficult to figure out what the fills with Neurobiotin were used for. Please also provide a brief description of how Neurobiotin was visualized. Streptavidin is only mentioned as such in the list of compounds.

10. Line 633: The sentence “Similarly, the dendritic field diameter ... around the tips of the dendrites” seems misleading. At first, I thought this had to do with measuring dendritic diameters, but realized that it most likely refers to measuring dendritic field areas, i.e., 2D convex hulls. If dendritic field diameter was calculated from the field area, how was this done? Please re-write and clarify.

11. Line 639: The reference to “Nath and Schwartz, 2016” is ref. 88. Please correct.

12. Line 649: Please provide full details for the acquisition of “two-photon volume images”, i.e. interval between slices, number of slices, how many used for illustrations, illustrations as single slices or maximum intensity projections, etc.

13. Line 701: The word “its” does not refer back to the correct phrase.

14. Line 703: I assume that “...to -45 mV the location of...” should be “...to -45 mV at the location of...”.

15. Line 705: Perhaps it would be clearer to write “...was measured at the location of all 91 ribbon output synapses...”.

16. Line 716: I assume that the equation here should be corrected to:

$$(1 - (Q1/Q4)) * 100$$

17. Table 3:

- HCN2 channel: ref. 121 is to ModelDB and "Firing neocortical layer V pyramidal neuron". Why is this selected as a reference here? Ref. 121 is not included as a reference for the model parameters in the rightmost column.

- L-type Ca²⁺ channels: ref. 126 is to ModelDB and "Simulated light responses in rod photoreceptors". Why is this selected as a reference here? Ref. 126 is not included as a reference for the model parameters in the rightmost column.

There are a number of grammatical mistakes in the manuscript, but I assume that the authors will catch them with a careful reading. For example:

- Introduction, line 26: grammatical error, either "representation...is" or "representations...are"

- Results, line 203: "Well-described" should be "Well described"

- Results, line 224: "which BCs types" should be "which BC types"

- Results, line 226: "dendritic arbor of" should be "dendritic arbors of"

Reviewer #2 (Remarks to the Author):

Surround inhibition is a canonical computation observed at almost every stage of sensory processing. Delineating the neural substrates that mediate surround inhibition is central to understanding how neural circuits process information. In this study, Swygart and colleagues investigate different levels of surround inhibition in two distinct types of retinal ganglion cells, the PixON and ON alpha RGCs, that have similar dendritic morphologies and stratification patterns, yet different inhibitory surrounds: PixONs have stronger surrounds, compared to the ON alpha RGCs, consistent with recent reports.

The authors suggest that the patterns of excitation rather than inhibition or cell-intrinsic factors (including morphology, receptor saturation/desensitization, channel, etc.) likely account for differences in surround inhibition between the two RGC types. This conclusion is based on the cumulative results from voltage-clamp (EPSC/IPSC analysis), dynamic clamp, pharmacology, and mathematical modeling experiments. Furthermore, the weak excitatory inputs to PixONs evoked with large spots are greatly enhanced by the application TTX and/or TPMPA (GABA_AR antagonist), indicating that spiking wide-field amacrine cells provide surround suppression at the level of presynaptic bipolar cell terminals.

Usually, different patterns of excitation would be taken to imply that different bipolar cells receiving different levels of surround inhibition drive these distinct RGCs. Surprisingly, however, this does not

appear to be the case. Using elegant SBEM and confocal analysis these authors demonstrate that both the PixON and ON alpha receive connections from the same set of bipolar cells. In some instances, single bipolar terminals were shown to connect to both RGC types.

Together, the physiological and anatomical results seed the hypothesis that different synapses within a single bipolar cell axonal terminal can be differentially inhibited and thus convey distinct streams of information. This is paradigm-shifting, as bipolar cell terminals are small and generally believed to be electronically compact. Using a mathematical model, they show that in the presence of active conductances, inhibitory inputs can remain compartmentalized enough within the axonal arbor to allow differential glutamate release from different synapses. Thus, this hypothesis seems biophysically feasible.

Results from this study using cutting technologies provide exciting and provocative new ideas. However, the foundational results on which the central hypothesis rests require to be solidified.

Major Comments:

1) Common input (functional)? The major conclusions of this study hinge on the idea that PixON and On alphas receive input from the same set of bipolar cells, but there is no functional evidence to support this idea. In fact, the bits of circumstantial evidence gleaned from the example traces refute this idea. Specifically, in response to small spots, the kinetics of the EPSCs would be expected to be similar in the two cell types, since there is little widefield inhibition recruited. However, EPSCs in ON alphas appear more sustained than those measured in PixONs. Comparing the properties of AMPA receptor-mediated EPSCs in the two types of RGCs (in the presence of TTX, TPMPA and NMDA receptor blockers; and cholinergic blockers, since work from the Demb lab shows ON alphas receive cholinergic starburst input?), across spot size and over a range of voltages (to ensure linearity), would more conclusively reveal whether or not the glutamate input properties are similar.

That the bipolar input is shared between these two RGC types would require a 'noise' analysis. For example, spontaneous EPSCs recorded simultaneously from pairs of PixON and ON alpha RGCs with overlapping receptive fields could be used to directly demonstrate common input (e.g. Trong and Rieke, 2008). Alternatively, direct stimulation of bipolar cells with Chr2 (using the in-house CCK-cre line that labels T6s) could demonstrate that the same T6s drive these RGCs (the trial-to-trial amplitude fluctuations of the optogenetically-evoked responses would be expected to co-vary). Albeit technically challenging, these experiments are well within the expertise of this lab and would go a long way in solidifying the base on which the primary hypothesis rests on.

2) Common input (anatomical)? In the same vein as point 2, the complementary anatomical analysis needs to be conducted more rigorously. For instance, the authors found the same type 6 bipolar cell could synapse onto both PixON and ON alpha ganglion cells. What fraction of the total synapses to each ganglion cell came from such common bipolar cells? The authors count a total of 8 bipolar cells that synapse with both cells, but the reader has no sense of whether this is a high/relevant number

compared to the total number of synapses. In general, more description of the EM data would be valuable, especially in the regions of dendritic overlap between the ganglion cells.

In the cases where a bipolar cell provided output to both ganglion cells, how far away were those synapses from each other? The model relies on synapses in the first and last quartile of voltage decay, which presumably correspond to synapses that are furthest away from each other or have the most branched path between each other. Is this realistic based on the synapses identified in the EM data set? Since we have ground truth data in the EM, the distances between PixON and ON alpha synapses used in the model should correspond to the distances found in the EM data set.

Other Comments/suggestions:

1) Ganglion cell type identification: It is not entirely clear what morphological functional criteria were used to ascertain recordings were made specifically from PixON and ON alpha RGCs, since these parameters are known to vary greatly across the retina. There is a particular concern for the PixON type, as many of their functional properties reported here do not appear to match previous studies in which these cells have been characterized in genetically tagged RGCs. For example, the profound surrounds of the PixONs are not apparent in work from Kerschensteiner's group (In fact, the response profiles to increasing spots size appear more like the ON alphas shown here). In addition, the contrast response functions (CRFs) were also previously reported to be linear, in contrast to the rectifying CRFs shown in this study. The authors should discuss these differences and provide a more in-depth description of how they identified RGC type and perhaps consider using other functional properties to confirm their identities (such as blue/green color opponency).

2) Dynamic clamp: E/I interactions occur on the dendrites and it is not clear to what extent the somatic dynamic clamp experiments capture this behavior. The results would be more compelling if this issue was tested experimentally. For example, how does stimulating surround inhibition with an annulus affect the responses generated by a somatic 'dynamic' excitatory conductance? Or conversely, how is the center spiking responses evoked by a small spot affected by artificially applying inhibitory conductances at the soma?

3) Computational model: In the model, the dendritic excitation appears to be a step change in conductance, which is then inhibited by a step change in inhibitory conductance in the axon terminals. The level of suppression at each output synapse is estimated by its steady-state membrane potential after inhibition is activated. If the amplitude of the hyperpolarization is decaying by as much as 50% in the active model (Fig. 6b), I wonder about the lengthening of the kinetics of the hyperpolarization that will also occur as voltage spreads down the cable to the other synapse. By using step changes in activity, it's difficult to assess how the kinetics and timing of inputs will affect glutamate release. Is it possible that by the time the hyperpolarization has reached the distant synapse, it has a slow enough rising phase that the depolarization from the excitatory input is unable to be blocked, especially in the presence of fast active conductances? The authors could use a more stochastic, event-based model to test how inhibitory events spread to other parts of the axonal arbor. It is expected that steady, tonic voltage

changes will permeate the arbor more effectively than brief, event-based activity. It's also possible that the excitatory drive from photoreceptors is more tonic, while the NaV-dependent inhibition is briefer and event-based. I think using more realistic synaptic activity patterns is important; it could end up supporting the authors' model even further.

4) Alternative models: The authors are careful not to overstate their results and do consider the possibility of chemical compartmentalization. Bipolar cells may express L and T-type Ca channels with distinct voltage-dependent properties (and both channel types support release; Pan et al., 2001; Neuron). In theory, a differential expression of these channels could trigger heterogeneous release during a global hyperpolarization of the terminal (owing to the higher threshold required for L-Type Ca channel activation). Alternatively, different Ca²⁺ buffering could lead to similar effects. Such mechanisms might be faster and more plausible than the second messenger-dependent mechanisms suggested in the discussion.

5) Inhibition at on alpha synapses: In the case that the author's model is accurate—where PixON synapses and ON alpha synapses are electrotonically separated enough to drive different levels of glutamate release—it is still the case that in the EM data, there was always an inhibitory synapse within a micron of each ON alpha synapse. Since these are all wide-field amacrine inputs, wouldn't these also be active during a surround-encompassing visual stimulus? By what mechanism would only the PixON amacrine inputs be active?

6) Nano-scale compartmentalization: The idea that the compartmentalization might be caused by the fact that inhibition is closer to the PixON synapses (line 317) seems a bit far-fetched since we're on the scale of 0.75 μm to 1 μm .

7) Inhibitory shunt? It would be good to discuss how one expects the inhibitory conductance itself to act at the site of glutamate release. Is there any significant shunting effect? The length constant of an open channel shunt is expected to be half that of voltage spread, which would be well-suited to help compartmentalize inhibitory activity.

8) Figure 3: Figure 3 is hard to interpret. Blocking glutamate receptors is likely to remove inhibition and therefore could increase glutamate levels. Is there any positive control to be done to confirm that the NBQX/Kyn experiment is capable of revealing whether saturation or desensitization is responsible for the weak surround suppression?

9) Contribution of postsynaptic inhibition: The authors claim that it is the patterns of excitation that dominate surround inhibition, rather than postsynaptic inhibition. This could easily be tested by using

GABA_A to block inhibition (since presynaptic inhibition is mediated by GABA_C, excitation would not be disturbed as shown in Fig. 5a).

10) Typos: Figure 2 and 3 legends have some mislabeling with the lettering.

Reviewer #3 (Remarks to the Author):

Swygart and colleagues investigate the underlying mechanisms for the difference in surround suppression between the PixON and ON alpha RGC types in mouse. Both receive similar synaptic inputs, yet surround suppression is much stronger in the PixON RGC. The authors find the PixON's surround suppression is driven by excitatory synaptic conductances (bipolar cell input) and cannot be explained by intrinsic properties, post-synaptic desensitization or saturation. Modeling suggests the difference in surround suppression is caused by differences in bipolar cell input, but serial EM reconstructions show no significant difference in presynaptic bipolar cell types to the PixON vs. ON alpha. Pharmacology indicates the cause of the PixON's surround suppression is narrowed down to spiking wide-field amacrine cells input to GABA-C receptors on the presynaptic Type 6 bipolar cells. The serial EM shows wide-field amacrine cell input located closer the Type 6 bipolar cell's synapses onto the PixON than the ON alpha. While the difference in distance is quite small (~0.25 microns), it is sufficient to produce differences in surround suppression with an active model of the bipolar cell axonal arbor. The model contains many assumptions but these limitations are sufficiently addressed and other possibilities such as chemical compartmentalization are also possible.

The demonstration of subcellular functional specialization in bipolar cells is an exciting result demonstrating that, yet again, the mammalian retina is far more complex than classically thought. While several groups have investigated functional divergence at the output of a bipolar cell, this paper is the most comprehensive to date and is novel in addressing both the mechanism within the bipolar cell and the impact on two distinct post-synaptic neurons.

Overall, this is an excellent manuscript and impressively comprehensive. The results are backed by multiple experiments employing a wide range of techniques (electrophysiology, serial electron microscopy, immunohistochemistry and multiple types of modeling) and alternative hypotheses were exhaustively investigated. Throughout the paper, questions or concerns about a result would arise and then the next paragraph would acknowledge and address them with a separate experiment. Accordingly, I only have a few relatively minor comments and questions.

I was a little unclear on the precise goal and limitations of the bipolar cell model predicting the differences in surround suppression between the PixON and ON alpha (page 9). The model suggests a difference in bipolar cell input generates the distinct responses of the two cells. This difference could be due to differences in the types of BC input or differences in the nature of a single BC type's input to each

RGC. The model fits excitatory responses produced by all presynaptic bipolar cell types, but the bipolar cell subunits are constrained to have the center receptive field size of a Type 6 bipolar cell.

- If the goal was to look specifically at Type 6 bipolar cells, how much of a limitation is it that the receptive fields are being fit to a response generated by all presynaptic bipolar cell types? Could a difference in the relative amount or strength of the different presynaptic bipolar cell types to the PixON vs. ON alpha RGCs produce a difference in the surround size and CSR simply because the model is doing its best to predict a response that includes non-Type 6 BCs?

- If the goal was to look at differences in bipolar cell types, how much of a limitation is it that the center receptive field size is fixed to that of a Type 6 BC?

Stabio et al (2017) reported cone opponency in the M5/PixON RGCs. Is it possible that this opponency could account for any of the surround suppression measured physiologically rather than Type 6 BC input? Would any of the results be expected to change with the spectral composition of the stimulus? Other readers might share my confusion in navigating cone opponency results from a species with an opsin gradient and appreciate explanation from the authors (no additional experiments needed).

I particularly enjoyed the series of dynamic clamp experiments and found them to be very compelling. I was curious about the dip in spiking after the transient response at stimulus onset with the ON alpha RGC conductances (Fig 2b and d) which was not present in Fig 1c. Why might this occur?

Page 14 – “We traced the presynaptic ACs at each type 6 BC ribbon” reads as if this all the presynaptic ACs near a ribbon were reconstructed but the Fig 5c and 5e legends specify only the nearest amacrine cell was reconstructed. Regardless, the possibility that amacrine cells target ribbon synapses presynaptic to a specific RGC type is very intriguing.

We sincerely thank the reviewers for the work they did scrutinizing our manuscript. This was an especially detailed and thoughtful set of reviews, and we believe that in answering each point, we have substantially improved the work. This included new experiments, new modeling, and rewrites of large portions of the manuscript. The reviews are copied below with our responses in blue text.

To aid reviewers, we would like to point out major figure changes that have occurred since the previous manuscript submission.

- **Supplementary Figure 6** was moved to the main document to become **Figure 4**. This shifts **Figures 4-7** from the previous document forward to become **Figures 5-8** in the current manuscript.
- **Figure 7** (now **Figure 8**) underwent major conceptual and visual changes, in which we more directly test our hypothesis by using center and surround ratio measurements instead of inferring them by inhibitory voltage decay.
- **Supplementary Figure 6** was added to estimate the preference of T6 and T7 BCs for $Pi_{X_{ON}}$ and ON alpha RGCs when taking into account the full RGC coverage factor. This analysis was prompted by reviewer #2, major criticism #2.
- **Supplementary Figure 10** (length constant of inhibitory synapse) was eliminated and replaced with **Supplementary Figure 11**, which calculates length constant across frequencies by sinusoidal current injection as suggested by reviewer #1, major criticism #1, and reviewer #2, minor criticism #3.
- **Supplementary Figure 10** was added to investigate dynamic properties vs. membrane conductance for active model results per Reviewer #1, major criticism #1.
- **Supplementary Figure 12** was added to compare model results to SBFSEM analysis in response to reviewer #1, major criticism #7, and reviewer #2, major criticism #3.
- **Supplementary Figure 13** was added to show the identification of synapses across SBFSEM z-sections in response to Reviewer #1, major criticism #8.

Reviewer #1

Major Criticisms

1. In their discussions of “electrotonic compactness”, the authors need to disentangle several related issues. For a passive compartmental model, the degree of electrotonic compactness can be precisely defined, preferably for a series of frequencies, including the DC level which is what the authors seem to address in their illustrated examples. In the Introduction, line 38-39, they refer to the supposedly commonly held view of bipolar cells as “electrically compact neurons”. The implicit message here is that perhaps bipolar cells are not electrotonically compact, or at the very least that this (commonly held) idea somehow has to be modified. In the remaining part of the paper, the authors do not bring up this issue again in a clear way, but their “solution” to the problem is that a different behavior (with respect to synaptic output) is found in their active models of bipolar cells. The authors do indeed acknowledge this (Results, line 327), but as a whole, the treatment is a bit unsatisfactory, as it does not make clear whether this change in behavior is a result of the dynamic properties of the added ion conductances or a change that for all practical purposes appears as a change in passive membrane properties. This needs to be addressed, potentially by additional simulations or analysis. More specifically, on page 21 (line 440->), the authors state explicitly that the results observed for the simulation with the active compartmental model might be explained by the influence of the added (active) ion conductances on the (effective) membrane resistance. To clarify their arguments, the authors need to state clearly how the conductances were added to the model, i.e., was the density the same in all compartments? Did the introduced channels change the effective resistance “at rest”? Have the authors investigated if similar results can be obtained simply by reducing the membrane resistance (to values that are still realistic)?

This is a very good point, and the answers will hopefully allow the reader some additional insights into the details of possible electrical isolation in BCs. We performed an additional set of simulations based on these suggestions, and we have included those results in the updated manuscript.

- a. The reviewer is correct that our previous modeling simulations were only testing electrical compactness for the DC level. We recognize that this does not reflect the true biology. Thus, we have implemented a new version of the model with stochastic activation of synapses with realistic kinetics (**Fig. 7b,c** and line 821 for implementation details).
- b. We also performed additional simulations, as suggested by the reviewer, to directly investigate electrical compactness across a range of frequencies (**Supplementary Fig. 11**). We injected sine wave currents in the soma at different frequencies and measured attenuation at the ribbons, allowing us to calculate length constants along these paths. These results showed that in the active model, substantial attenuation only occurs at very high frequencies (>100 Hz) and yielded relatively long length constants (>400 μm). However, when adding stimulation of the excitatory and inhibitory synapses, attenuation became largely frequency independent and yielded much shorter length constants (~70 μm).
- c. We performed further simulations to investigate the degree to which active channels increase membrane conductances and decrease effective membrane resistance (**Supplementary Fig. 10a-e**). We found that the addition of active channels did increase total membrane conductance by a factor of 5.4 ± 0.15 (mean \pm std) during exc./inh. stimulation.
- d. To determine if the dynamic properties of the active channels played a role beyond increasing total membrane conductance, we eliminated the active membrane conductances and increased the passive leak channel conductance so that the total membrane conductance remained the same (used average total conductance during stimulation in the active model). We first did this uniformly across the whole membrane so that all portions of the membrane had the same leak channel density. This passive model with increased leak conductance did exhibit about a 4x wider range of CSR values than the original passive model. However, the active model did maintain a 2x greater range of CSR values over the uniformly increased leak conductance passive model (**Supplementary Fig. 10f**). Since conductances in the active model varied by location, we performed one additional simulation in which the passive leak conductance was increased locally at every location on the membrane to match the sum of all conductances at that location in the active model. The results were nearly identical to that of the uniformly increased leak experiment.
- e. We have made sure that details about the locations of the ion channels and their biophysical properties are listed in **Table 3 | Key parameters of T6 BC NEURON model**. K_v^+ channels were equally dense across the entire cell membrane. L-type Ca^{2+} channels and HCN2 channels were restricted to the axonal arbor of the cell.

These results suggest the voltage-gated channels greatly increase total membrane conductance and thus decrease the effective membrane resistance. This increase in membrane conductance increases the range of CSR values measured at the ribbons, but the dynamic properties appear to increase the range of ribbon CSR values even further. We also see that the shunting effect of inhibition and excitation play a major role in decreasing the electrical compactness across a wide range of frequencies.

2. In Figure 5f (and the accompanying) text in the Results section, the authors illustrate the distance from ribbon synapses to the nearest inhibitory synapse.
 - a. - first, why do the authors believe that measuring the Euclidean distance is the correct / relevant measure here? Cf. Methods (line 671-672). Would not path distance be more relevant in the context of functional interaction?

At scales < 2 μm (distance between a ribbon and the nearest inhibitory synapse), path distance is not well defined in the SBFSEM reconstruction of the BCs. These synapses are always on the same compartment in the model and a Euclidan path does not cross the cell membrane. Whenever measuring distances at larger scales, such as the distance *between* ribbons, we measured path distance (**Supplementary Fig. 11 & 12**).

- b. - second, the authors make it a point that the difference between Pix_ON and ON alpha was statistically significant. However, although statistically significant, the difference seems very small and there is considerable overlap. Please discuss the functional importance of the difference (potentially for path distance, see above).

We absolutely agree that a distance difference at this tiny scale is only suggestive of some functional difference that likely has nothing to do with voltage spread. We have added the following text to this section.

Although this data was statistically significant, the difference in distance was very small, and its functional implications are unclear. Voltage signals are expected to be nearly identical at this scale (Fig. 7), but these differing distances could suggest the presence of other functionally important differences, such as input from differing amacrine cell types or differing synapse structures.

We speculate in the **Discussion** that distances in this range could play a role in the compartmentalization of a chemical signal (e.g., a neuromodulator from an AC with direct, voltage-independent effects on Ca²⁺ channels).

3. The explanation of the analysis performed of the extent of hyperpolarization induced by inhibitory inputs and the extraction of top and bottom quartiles (Q1 and Q4) is very confusing. It seems to me that part of the confusion is due to inconsistent description of what “top” and bottom” quartiles refer to. Is “top” Q1 and “bottom” Q4? Or is it the reverse? The legend of Figure 6 calls Q1 top and Q4 bottom, but the legend of Figure 7 calls Q4 top and Q1 bottom. Figure 6c-e illustrates that the values for Q1 are smaller than the values for Q4. The legend for Figure 7f refers to the decrement of average hyperpolarization from Q2 to Q1, but the Y axis label indicates “Q4 to Q1”. See also the text on page 16, Figure 6 (compare panels c-e with the text in the legend), and text on page 28 (lines 713-716).

We thank the reviewer for their critique of our presentation of this part of the data. Indeed, we found it challenging to present these modeling results with clarity, and that was made even worse by a couple of typos in the figure legends that we have now corrected. We have modified the text in several areas to more explicitly describe Q1 and Q4. We also changed the Y-axis label in Fig. 6g to be more descriptive.

4. The details of the compartmental simulations of the interaction between excitatory and inhibitory inputs in the bipolar cell (axon terminals) are a bit confusing:
 - a. - it should be clearly stated if all measurements of inhibitory hyperpolarizations were made with reference to (relative to) a prior excitatory input (incl. Methods, line 711)

Agreed. We rewrote the model considerably, so now it uses measurements of center-surround ratio (CSR) instead of hyperpolarization. This is more directly related to our scientific question.

- b. - the way I understand it, all inhibitory synapses had their strength adjusted to induce a local hyperpolarization to -45 mV. How can the authors justify this and what are the consequences for the rest of the analysis? Does this not amount to a “correction” of the associated conductance such that synaptic input at a location with low(er) input resistance is automatically increased to compensate for the higher dissipation of the evoked voltage? Perhaps the simulations should instead vary the inhibitory conductances randomly for the different locations?

Based on this comment, we rewrote the model with inhibitory strength based on the voltage recorded at the ribbon output synapses. We set inhibition so that the ribbon synapses experience a CSR equal to that best fit by the BC subunit model for Pix_ON RGCs. This allows us to analyze the degree to which the inhibitory surround decays, allowing some ribbons to experience decreased surround suppression. We also tested the model across a range of inhibitory strengths (**Supplementary Fig. 9f**).

- c. - for Figure 6b-e: why does the inhibitory hyperpolarization never go more negative than -45 mV, given the E_Cl was set to -50.5 mV (according to Table 3)? Is this a result of the scaling of the inhibitory conductances?

Yes, inhibitory conductances were not large enough to drive the membrane all the way to the Cl^- equilibrium potential. We tested a range of scaling factors for the inhibitory and excitatory conductances in **Supplementary Figure 9f** and find that the range of CSR values measured at the ribbon synapses increases with increased excitatory and inhibitory conductances.

- d. -how was the excitatory input modeled? As a fixed current or as a fixed conductance? As the current/conductance required to depolarize to -35 mV irrespective of other changes to the cell? What are the consequences of the strategy chosen for the results obtained?

Excitatory synapses were originally modeled as a fixed conductance with a strength that would drive the cell to -35 mV. However, based on this reviewer's suggestions, we now model both excitatory and inhibitory synapses with two-state kinetic schemes that more closely reflect the rise and decay times of real synapses. The event rate and maximum synapse conductance are still set to achieve specific membrane voltages at the ribbon synapses (-30 mV for excitation and CSR value of 1.1 for the first quartile of ribbon synapses during simultaneous simulation of excitation and inhibition).

5. Discussion, line 495-499: Here the authors discuss the morphology of the reconstructed type 6 bipolar cell and the possibility of measuring local Ca^{2+} at the scale of individual synapses. It is not my intention to argue that the authors should attempt measurement of local Ca^{2+} for the current paper, but I think their discussion should be modified by taking the following into account. They argue (line 496) that the reconstructed axon terminals constitute "a compact 3D structure mostly lacking anatomical compartmentalization". There are several problems with this statement. First, it seems inconsistent with other reports of the morphology of cone bipolar cell axon terminals that have described discrete varicosities (Golgi, fluorescence, EM). Does this mean that the reconstructed cell is an atypical cone bipolar cell? Second, in a different place in the paper (Methods, line 670), the authors describe the reconstructed axon terminals in different terms, using the words "...all the presynaptic (amacrine) inhibitory sites on the bipolar cell bouton...", suggesting that anatomical compartmentalization in the form of "boutons" was indeed present. Perhaps the authors need to clarify their use of the term "anatomical compartmentalization"?

In the discussion, we were comparing cone bipolar cells to A17 amacrine cells. We meant to convey that the cone bipolar cells have relatively less isolated varicosities and a much smaller structure. The cone bipolar cells do indeed have a complex 3D axonal structure with enlarged sections of their axonal tree connected by relatively thinner sections (see **Reviewer Fig. 1**). However, we do see ribbon output synapses on both the wide "boutons" and the narrower parts of the axon. Additionally, we see multiple ribbon output synapses on a single "bouton". We have rewritten the relevant portions of the manuscript for clarity [line 591].

Reviewer Fig. 1 | Bouton structures on the T6 BC axonal arbor.

a, SBFSEM reconstruction of T6 BC axonal arbor (same as Fig. 7a). Numbers 1-4 indicate zoomed-in views of axon terminals with bouton structures. Note: View in 1-4 has been rotated to best illustrate the bouton structures.

- When an increasing number of inhibitory synapses were activated, the authors stated that they added a number N of nearest inhibitory synapses (by path distance). I am not sure I understand the motivation or necessity of this approach. What would the results be if one instead added a selection of N [1 - 119] extra inhibitory synapses randomly selected from the total population?

Indeed, there are many possible combinations of inhibitory synapses that could be activated. Part of the justification for using the N nearest is that there is precedent for individual cells forming multiple synapses near each other purely because of contact probability. More importantly, however, we were trying to push the model to the limit where it could achieve maximal electrotonic isolation to see if it was

theoretically possible to achieve an amount of isolation consistent with our results. Indeed, there are many configurations of synaptic input that would lead to decreased isolation. We have added text to **Results** that we hope more clearly spells out this justification and its limitations [line 396].

7. The authors suggest a differential targeting of ribbon synapses within single cone bipolar axon terminals onto the retinal ganglion cells, depending on their modulation (weak vs/ strong) by inhibitory input from amacrine cells. Can the authors use their SBFSEM data to examine whether the ribbons with the largest / smallest hyperpolarizations differentially (preferentially) target the different types of retinal ganglion cells?
 - a. We don't know, experimentally, which ribbons have strong vs. weak input from ACs. We measured a set of anatomical parameters of the synapses that were available in the SBFSEM data to see if any of those parameters varied systematically with the RGC type target. The only anatomical parameter for which we found a significant difference was the distance to the nearest inhibitory synapse between T6BC synapses to PixON and T6BC synapses to ON alpha RGCs, as shown in **Figure 5f**.
 - b. We did examine anatomical features that influenced ribbon CSR values in our model and found that ribbons nearer to the soma tended to have higher CSR values (**Supplementary Fig. 12d-f**). This was due to these ribbons being closer to the dendritic excitatory synapses and further from the axonal inhibitory synapses. However, when we analyzed the EM reconstruction, we did not find that the T6 BC - ON alpha synapses occurred closer to the BC primary axonal stalk (**Supplementary Fig. 12g**). It should be noted that the correlation between soma distance and CSR value was most prominent when activating all 120 inhibitory synapses, but this set of activations was found to be insufficient for predicting Pix_{ON} and ON alpha surround suppression (**Fig. 8f**).
8. From the figures included in the paper, it seems to me that some synapses (ribbon, amacrine) were securely identified, but some of the examples included in the figures are not very convincing. I acknowledge that this can be related to the low resolution in the figures provided for review, but it can also be a problem with the overall lower resolution of SBFSEM images and potentially raises a question of the reliability of the quantitative data generated from these experiments. See specific comments below (FIGURES).

The resolution of our EM images was degraded in converting to PDF. Additionally, we rely on serial sections to identify a synapse, so it can be hard to see in a single section. We added an additional SBFSEM slice in **Figure 5e** to better show ribbon identification. We also created **Supplementary Figure 13** which shows multiple SBFSEM slices used to identify both excitatory and inhibitory synapses.

9. In the leftmost part of Figure 5a, we can see that the experiment with combined application of TPMPA and TTX was only performed for two cells. Although the suppression is very strong, it would definitely be preferable to see the result reproduced for a few more cells.

More cells were recorded (N=5), and the responses remained consistent.

10. It is easy to fix, but there are major problems with lack of consistency between figure panels and legends / text (see below under "FIGURES").

Thank you for catching these errors. They have been fixed.

Minor Criticisms

1. For the morphological results, it is a problem that several figures / panels are very small. For some of these illustrations, larger figures with much higher resolution would have made the evaluation easier. As is, some figures / panels with morphological data (e.g. immunolabeling, reconstructions) mostly have a decorative value. To the extent permitted by the journal, this should be improved.

Thank you. Our figures appear to have lost quality during conversion to PDF. We have ensured higher quality images with this resubmission.

2. Abstract, lines 17-18: The sentence “Our results show that divergence in the level of surround suppression occurs subcellularly, at bipolar cells synapses” is difficult to understand. Perhaps it would be clearer if “divergence” was replaced with e.g. “variability” or “heterogeneity”?

Done. Replaced with “heterogeneity.”

3. Introduction, line 30-31: The sentence “Nonetheless, the number of functionally distinct RGC types exceeds their stratification diversity” is misleading. Nobody believes that it is the stratification per se, which essentially is just an epiphenomenon, that is the crucial mechanism for differential signal processing. The stratification level provides an opportunity for engaging processes of other cells in synaptic circuits, which directly links to the authors’ subsequent arguments. The main result of the current study is that the potential for differential processing goes even further, with mechanism(s) that might permit differential output from within a single bipolar axon terminal.

We definitely agree. In our view, the degree to which one stresses the importance of RGC stratification in its functional properties still varies amongst retinal neurobiologists, but clearly, we are preaching to the choir with this reviewer. We have tried to emphasize in the text that selective wiring within a stratum is known to exist, but indeed our results go a step further in demonstrating sub-cellular specificity.

4. Introduction, line 35: Here and elsewhere (e.g. line 62), the authors need to tighten up their terminology. The terms “sublamina” / “sublaminae” refer to sublamina a (sclerad 2/5 of the inner plexiform layer) and sublamina b (vitread 3/5 of the inner plexiform layer). The mammalian inner plexiform layer is generally divided into five equally thick strata, termed stratum 1 - stratum 5 (S1 - S5). For the example in line 35, “sublamina” should be replaced with “stratum”.

We have changed the text throughout, eliminating the term “sublamina” and replacing it with the more specific “stratum.”

5. In some places, e.g. Introduction (lines 38-39, line 51), Discussion (lines 411), the authors contrast their own idea / speculations with what they term e.g. “a commonly held view”. This can be a useful “rhetorical strategy”, but for it to be more than a straw man argument, the authors need to provide examples of references such that we (as readers) are convinced that the views indeed are (fairly) commonly held. Similarly, if the authors want to argue that “bipolar cells are assumed to be isopotential” (line 411), they should provide a couple of references where this is stated explicitly.

We have added references to each of these locations in the paper to show that these views are commonly expressed in the literature.

6. The authors use both the term “electrical” and “electrotonic” in relation to compactness. I would suggest that they stick to one expression.

Agreed. We switched two instances of “electrotonic” to “electrical”. “Electrotonic” suggests passive current flow only, so we use this term accordingly.

7. Introduction, line 42: To state that “surround suppression” is one of the “oldest” visual computations is unclear. Do the authors mean “oldest” in the sense of “phylogenetically oldest” or in the sense of “classical(ly studied)”.

We meant classically studied, not phylogenetically old, but indeed it was ambiguous. We took out the word “oldest” to avoid this confusion.

8. Results, line 61-63: The text refers to visual responses (“ON-sustained”), but Fig. 1a,b does not show visual responses.

Fixed.

9. Results, line 75: Should “Fig. 1d,e” be “Fig. 1c-e”?

Fixed. We went through all references to figures in legends and the main text and corrected several typos.

10. Results, line 84: Here and elsewhere (e.g. line 674, 680), clarity could be improved when the authors use the word “across”, e.g. as in the phrase “across stimulus size”, by re-writing e.g. as “across a range of stimulus sizes”.

Fixed.

11. Results, line 84-87: The authors refer to published results (ref. 12) concerning spatially distinct regions where PixON retinal ganglion cells receive excitation and inhibition, but do they argue that this is a precondition for the ability of voltage-clamp recording to separately record excitatory and inhibitory inputs arriving in the dendritic tree? Please clarify.

This was not a precondition for achieving voltage-clamp. We simply utilized this feature of the Pix_{ON} receptive field to verify the effectiveness of our voltage clamp. Since specific visual stimuli effectively isolated excitation (small spot) and inhibition (annulus), we were able to verify that our voltage-clamp holding potentials were effective in isolating excitatory currents (-60 mV) and inhibitory (20 mV) currents. We have tried to make this more explicit in the text [line 86].

12. Results, line 102: Here and in several other places (e.g. line 266), the authors need to increase the level of precision when they use the phrase “surround suppression”. Please be consistent and always make it clear if the phrase refers to spiking or conductance (excitatory / inhibitory).

Agreed. Precision in this point is important to the results. We have gone through and changed the term throughout to make it explicit whether we are referring to spikes, excitation, or inhibition.

13. Results, line 110: I do not think that the sentence that starts “To independently test the ability of...” is correctly structured. I suppose that what the authors intend to say is that they wanted to test the ability of an injected excitatory conductance to drive surround suppression of spiking, independently of an inhibitory conductance. Please clarify.

The sentence has been rewritten.

14. Results, line 131: I think it would be clearer to rearrange the sentence to read “Differing levels of surround suppression of excitatory conductances between the...”.

We felt that our wording was slightly more clear, but the difference is minor.

15. Results, line 133-135: Here the authors speculate how different size stimuli might impact the state of saturation / desensitization of glutamate receptors on retinal ganglion cells. To make the analysis / arguments easier to follow, please make it clear if the authors believe that the excitatory drive (conductance) would be reduced for larger spot sizes. What about invoking a receptor-dependent mechanism for the stronger surround suppression seen for the Pix_{ON} retinal ganglion cells? Could that work?

We rewrote this section to make the logic more clear. We are unaware of a receptor-dependent mechanism that would make surround suppression stronger in Pix_{ON} RGCs than the expectation from the BC glutamate release profile.

16. Results, line 142: please restate argument / logic briefly, instead of just providing a reference to earlier work.

We added a paragraph here to restate the logic of this argument.

17. Results, line 145: The way the authors have constructed the text within the parenthesis, “Kyn” and “NBQX” appear as subjects with “suppressed” as verb. I do not think this is the intention. Please re-write / clarify.

Fixed. We rewrote this sentence.

18. Results, line 164: Here and elsewhere, the authors use the term “subunit”. As far as I know, this term has its origin as an abstract entity in relation to studies of receptive field properties, but in this paper the authors use it sometimes to refer to an abstract entity and sometimes to a bipolar cell (as a functional term?). As a whole, the text can be a bit confusing and I would encourage the authors to rewrite to make it more stringent and easier to understand.

Indeed, the word “subunit” was first used by Enroth-Cugell and colleagues to refer to an abstract, non-linear region of a receptive field (RF). Later work confirmed that BCs can often account for the RF subunits^{1,2}, but that is not necessarily the case, and nonlinear spatial integration is not the focus of our study. We needed a term to refer to the contribution of a single BC to the excitatory drive to an RGC. We landed on “BC subunit” as a compromise because it retains the notion of a sub-section of the RF but also explicitly includes “BC.” We have been careful in the revised manuscript not to give the impression that we are explaining nonlinear RF subunits in this work.

19. Results, line 169-170: Please re-write to make it clearer what “differences” in “...differences in their dendritic arbor...” might refer to, i.e., what kind of potential differences do the authors have in mind?

We rewrote this sentence for clarity.

20. Results, line 172-173: Please clarify the argument involving the phrase “...which could cancel out BC subunit inhibitory surrounds over a larger area”. Perhaps re-writing with a proposed example could improve the understanding.

We rewrote this sentence for clarity.

21. Results, line 176: It is not (entirely) clear what “These two variables” refer back to. Please clarify / re-write.

We rewrote this sentence for clarity.

22. Results, line 177: I think it would be good if the authors could make it clear when writing that they “modeled ... conductances” if this involved a compartmental model or not. Please think of a way to make this clearer in the main text, i.e., which of the several modeling techniques a specific instance of “modeling” refers to.

We have clarified in the text that this is a different model, now referred to as the “BC receptive field model” each time it appears.

23. Results, line 204: The phrase “...did not result in converging...” is unclear. I guess that what the authors intend to say is that the results from fitting remained different, but the use of “converged” is suggestive of some kind of reiterative fitting procedure. Please clarify.

We rewrote this sentence for clarity.

24. Results, line 230 and rest of paragraph: Please clarify what the reported percentages are a percentage Fixed.

25. Results, line 265: I think it is better to write “GABA or glycine receptor antagonists” as the different types of receptors were not blocked simultaneously.

Fixed.

26. Results, line 274-276: Here and elsewhere (Discussion), how can the authors explain that the GABA_C receptor antagonist TPMPA alone was not as efficient as when combined with TTX? Can it be explained by incomplete antagonism with the concentration of TPMPA used?

GABA_C receptors were mostly blocked with our concentration of TPMPA (50 μM), and we couldn't go much higher, or we would block a lot of GABA_A.

GABA_C K_D = 2.1 uM; 96% bound

GABA_A K_D = 320 uM; 14% bound

GABA_B K_D = 500 uM; 9% bound

As for the non-additivity of TPMPA and TTX, we have now addressed this point in the text but only with speculation [line 559]. Indeed, this is part of the problem with bath-applied pharmacology; there may be a host of interactions from unknown circuits. TTX alone could reduce the GABA transmission from wide-field (spiking) amacrine cells but not totally eliminate it because of passive voltage spread in the neurites. The lack of a complete block of surround suppression with TPMPA might suggest that another

receptor, perhaps for a different molecule, is involved in the regulation of glutamate release from the BC ribbon synapses.

27. Results, line 279-283: This section is very clearly written. I suggest using some of these phrases to improve the corresponding part of the Abstract, which I found much less clear.

Thank you. While we agree that the precision of the writing is better here than in the abstract, there is simply not enough room in the Abstract to provide the appropriate context for these details.

28. Results, line 292-293: Please state the (numerical) ranges for each statistic reported here (synapse distances).

We now include the range in the figure legend and text

29. Results, line 306-307: Something is wrong with the grammar in “alpha synapses” in line 307. Perhaps it is better to re-write in less “interpretative” terms, e.g. by something like “A synapse made by the amacrine cell constituted the nearest neighbor for ribbon synapses onto both types of RGCs”?

Fixed.

30. Results, line 311: I think it might be better to write “...within the same BC axon terminal...”.

Changed to “axonal arbor.”

31. Results, line 329: Here and elsewhere, the authors often use the notation “HCN₂” for HCN2 channels. As used in some places in the paper, HCN2 is the standard way to refer to this channel type. Please modify accordingly, here and elsewhere.

Fixed.

32. Results, line 325-328: It is a bit misleading when the authors write that “multiple studies have measured voltage-gated ion conductances from bipolar cells” but provide a series of references to support this statement (ref. 32-35), only one which (ref. 32) is an experimental study. Please re-write and clarify, ideally by citing studies where bipolar cells conductances were measured experimentally.

Agreed. We added references with explicit measurements of each of the voltage-gated channels in BCs that we added to the model.

33. Results, line 348: When referring to recruitment of inhibitory synapses, the authors sometimes refer to the “n-nearest” and sometimes to the “N-nearest”. Please be consistent.

Fixed.

34. Results, line 364: The phrase “...voltage decay of the BC’s inhibitory surround...” seems like lab jargon. Please re-write / clarify.

We rewrote this entire paragraph to improve clarity.

35. Results, line 393: I guess “independently” should be “independently of it”?

Fixed.

36. Discussion, line 413-414: I am unsure how relevant it is for the context of bipolar cell axon terminals to refer to the paper by Linden (ref. 43) as his discussion referred to much larger axonal trees where the crucial idea is conduction block of action potentials, e.g. at branch points.

Fair enough. Indeed, Linden was making a point about much larger axonal trees, but the implications for neural computation apply to functional divergence at any scale. It’s perhaps even more remarkable that a neuron this small can deliver functionally distinct outputs. We have added a sentence to make this point.

37. Discussion, line 430: I suggest changing “electrotonic isolation” to “electrotonic compartmentalization” to make the point more general.

Changed to “electrical compartmentalization” because we are avoiding “electrotonic” based on the comment above (electrotonic infers a passive model).

38. Discussion, line 437: It seems misleading to write "...a cell type typically modeled as passive and isopotential". To model as passive is an explicit choice, but to model as isopotential is a conclusion from modeling, not a choice that can be made before modeling starts.

Agreed. We removed this sentence since it did not really fit in this part of the Discussion anyway.

39. Discussion, line 439: In the context of the terminology appropriate for the compartmental modeling, it is not correct to write that the inclusion of active conductances decreased the membrane resistance. The membrane resistance specifies a passive membrane property. Perhaps it is better to refer to the "input resistance"? Or the "apparent membrane resistance" that would be estimated if an active conductance was active at the membrane voltage at which membrane resistance would be estimated?

Changed to "effective membrane resistance."

40. Discussion, line 458: In "these barriers to Ca²⁺ diffusion", the phrase "these barriers" refers back to words that are not included in the sentence. Please re-write / clarify.

We rewrote this sentence for clarity.

41. Discussion, line 501: "the molecule driving postsynaptic current" should be changed to "the molecule driving postsynaptic conductance".

Fixed.

42. Discussion, line 518: The statement that the authors "showed that different glutamate release profiles are experienced by RGCs..." seems a bit too strong. It is an interpretation, not a direct demonstration, as no direct measurements of glutamate release profiles were made. Please improve.

Agreed. The sentence now reads as follows:

"Thus, while we did not directly measure different functional signals within the terminals of a single BC, we showed that the same population of BCs can drive differing excitatory conductances in downstream RGCs."

43. Discussion, line 534: It is unclear to me if the citation of the paper in ref. 30 is made to support the statement in the corresponding sentence or not? Please clarify the intended meaning here.

We removed this reference. It appears in the preceding sentence where it refers to the actual findings of that paper and not their interpretation.

44. When discussing dendritic branches, the authors need to clarify if they mean e.g. "branch segments" (between nodes) or the whole branch from origin at soma to the dendritic tip(s).

We added text that clarifies that the average branch length was "distance between branching nodes".

FIGURES

Figure 1:

- - the annotation for panels c, f and i could preferably be moved to make it immediately clear that it pertains to both the left and the right graphs in each panel (row)

We added more letters and updated the figure legend.

- - in panels g and j, the Y axis labels could be made easier to understand by changing the unit from "delta nS" (ΔnS) to "nS" and by changing the text from "Excitatory / Inhibitory conductance" to "Delta Excitatory / Inhibitory conductance" (" Δ Excitatory / Inhibitory conductance")

Done.

- - in the legend for panel c, change "Example peristimulus time histograms to..." to e.g. "Example peristimulus time histograms in response to..."

Done.

Figure 2:

- the panel lettering does not match the lettering in the legend and in the main text
The panel mismatch was because of a version issue with the figure. We apologize for the mistake. It is corrected in the updated version.
- the legend for panels b,c states "...histogram recorded..." but it should be "...histograms recorded..."
done
- to make panel "h" (erroneously labeled "f" in the figure itself) easier to grasp quickly, consider changing the annotations "Exc_ff" and "Inh_ff" to "Exc_full" and "Inh_full". This style also better parallels that used for "Exc_pref" and "Inh_pref"
done
- for panels e-g the legend reads "e-g Same as b-c". This is awkward, as a consecutive series of three panels does not match two consecutive panels. Consider to re-write as "e-g Same as b-d"
done
- the legend for panel h states "...conductances as to isolate..." but I believe it should read "...conductances to isolate..."
done
- the legend for panel i states "...when switching to full-field excitation..." but it is less clear what the switching was from, please clarify
Rewrote sentence for clarity.

Figure 3:

- panel a as labeled in the figure is not included or commented on in the legend or the main text. panels b-g as labeled in the figure are wrongly labeled in relation to the legend and the main text
Fixed. We had included the wrong figure version for this legend.
- please clarify how the conductances illustrated in panel c (erroneously labeled d in the figure) were measured (peak?)
done

Figure 4:

- panel c: is the region displayed in this panel taken from the region displayed in b? Please clarify and modify panel b accordingly
We added a bounding box for clarity.
- panel c: the annotation indicates that ribbons are supposed to be labeled red, but apart from the labels "1" and "2", it is really hard to see any clearly labeled (red) ribbons
The resolution drop when converting to PDF did not help, but indeed they are extremely small, so we added arrows.
- panel c: what is the significance of the "grayish" and "whitish" regions of the reconstructed bipolar cell axon terminal?
We added the words "semi-transparent gray mesh" to the legend.
- panel e: is this really a ribbon?
Fixed
- panel f: is the region displayed in this panel rotated or flipped relative to the corresponding region displayed in panel c? Please clarify
Yes. clarified in legend

- panel g: where is the ribbon?

Clarified in legend

Figure 5:

- the legend for panel a states “antagonists of” but includes “TTX” at the end of the sentence. However, TTX is a blocker, not an antagonist, of Nav channels (there is no receptor and no [endogenous] agonist). Please re-write

Fixed.

- the legend for panel a states that the conductance trace in TPMPA+TTX was shifted down 2 nS. I assume this only refers to the trace for the full-field stimulus? Please clarify

Yes. Clarified in legend.

- panel c: here the indicated ribbon is very convincing, but the (inhibitory) synapse supposedly made by the amacrine cell onto the bipolar axon terminal is much less convincing. Please clarify / comment

We rely on serial sections to identify a synapse, so it can be hard to see in a single section. We added an additional z-slice in **Figure 5e** and also added **Supplementary Figure 13**, which shows multiple SBFSEM slices used to identify excitatory and inhibitory synapses.

Figure 6:

- the legend for panel b (and elsewhere) uses the phrase “voltage trace of ribbons”. This is lab jargon, please re-write more precisely, perhaps “voltage response recorded at location of synaptic ribbon” is better?

Fixed.

- in panel b, why does the membrane potential before onset of the excitatory conductance reside at -45 mV, given that the reversal potential of the leak conductance was set to -60 mV (Table 3)?

We updated the model and clarified this point in the text [line 825].

- in panel c, does “1 inh. activated” refer to the same inhibitory synapse for all simulations, i.e., all ribbons, or does the specific inhibitory synapse activated vary according to the ribbon? Please clarify in figure itself and/or in legend and text

Panel c shows the resulting CSR measurements when activating 1 example inhibitory synapse for a single simulation. We try to make this clear in the text [line 383].

Figure 7:

- in the legend title, should “inhibition on” be “inhibition of”?

We rewrote the legend for this figure.

METHODS

1. Line 574: the sign of the liquid junction potential does (presumably) not follow the convention of E. Neher. It should probably be +8.6 mV, not -8.6 mV. This is important if the correction involves subtraction or addition of the liquid potential.

Fixed.

2. Line 575: “reversal potential of chloride” should be “equilibrium potential of chloride”.

Fixed.

3. Line 579: “77 mOsm” should presumably be “277 mOsm”, or perhaps another value (a bit odd if exactly identical to the value for the other intracellular solution)?

Fixed.

4. Line 582: Desai et al. is not in the list of references. The hardware described in Desai et al. is by now a bit old and, if relevant, it would be useful if the authors in addition briefly described newer hardware used in their own study.

Fixed the missing reference. This is the hardware we used.

5. Line 588: What does “best reproduced” mean in terms of the method used? As determined by eye or was a re-iterative fitting procedure used?

More information has been added to this section to clarify.

6. Line 592: Does “maximal spiking response” refer to the peak firing rate? How was it determined and how many repetitions were averaged?

Maximal spiking response refers to the spot size that caused maximal spiking. Text was edited for clarity.

7. Line 602: I suggest reminding the readers here, using a few words, that the ganglion cells studied are intrinsically photosensitive.

Added.

8. Line 618: What was the paraformaldehyde dissolved in?

Added. 0.1M phosphate buffer.

9. Line 626: It is a bit difficult to figure out what the fills with Neurobiotin were used for. Please also provide a brief description of how Neurobiotin was visualized. Streptavidin is only mentioned as such in the list of compounds.

We added references to figures that utilize confocal imaging and immunohistochemistry to image Neurobiotin labeled with Alexa-Fluor-conjugated Streptavidin.

10. Line 633: The sentence “Similarly, the dendritic field diameter ... around the tips of the dendrites” seems misleading. At first, I thought this had to do with measuring dendritic diameters, but realized that it most likely refers to measuring dendritic field areas, i.e., 2D convex hulls. If dendritic field diameter was calculated from the field area, how was this done? Please re-write and clarify.

Clarified in the text. We computed the “effective diameter” based on the convex hull area ($\text{area} = \pi * r^2$).

11. Line 639: The reference to “Nath and Schwartz, 2016” is ref. 88. Please correct.

Fixed.

12. Line 649: Please provide full details for the acquisition of “two-photon volume images”, i.e. interval between slices, number of slices, how many used for illustrations, illustrations as single slices or maximum intensity projections, etc.

Clarified in text.

13. Line 701: The word “its” does not refer back to the correct phrase.

Sentence is no longer there in rewritten model methods.

14. Line 703: I assume that “...to -45 mV the location of...” should be “...to -45 mV at the location of...”.

Sentence is no longer there in rewritten model methods.

15. Line 705: Perhaps it would be clearer to write “...was measured at the location of all 91 ribbon output synapses...”.

Sentence is no longer there in rewritten model methods.

16. Line 716: I assume that the equation here should be corrected to: $(1 - (Q1/Q4))*100$

Sentence is no longer there in rewritten model methods.

17. Table 3: HCN2 channel: ref. 121 is to ModelDB and “Firing neocortical layer V pyramidal neuron”. Why is this selected as a reference here? Ref. 121 is not included as a reference for the model parameters in the rightmost column.

Fixed.

18. Table 3: L-type Ca²⁺ channels: ref. 126 is to ModelDB and “Simulated light responses in rod photoreceptors”. Why is this selected as a reference here? Ref. 126 is not included as a reference for the model parameters in the rightmost column.

Fixed.

19. There are a number of grammatical mistakes in the manuscript, but I assume that the authors will catch them with a careful reading. For example:

- a. -Introduction, line 26: grammatical error, either “representation...is” or “representations...are”

Fixed.

- b. - Results, line 203: “Well-described” should be “Well described”

Fixed.

- c. - Results, line 224: “which BCs types” should be “which BC types”

Fixed.

- d. - Results , line 226: “dendritic arbor of” should be “dendritic arbors of”

Fixed.

Reviewer #2

Major Criticisms:

1. Common input (functional)? The major conclusions of this study hinge on the idea that PixON and On alphas receive input from the same set of bipolar cells, but there is no functional evidence to support this idea. In fact, the bits of circumstantial evidence gleaned from the example traces refute this idea. Specifically, in response to small spots, the kinetics of the EPSCs would be expected to be similar in the two cell types, since there is little widefield inhibition recruited. However, EPSCs in ON alphas appear more sustained than those measured in PixONs. Comparing the properties of AMPA receptor-mediated EPSCs in the two types of RGCs (in the presence of TTX, TPMPA, and NMDA receptor blockers; and cholinergic blockers, since work from the Demb lab shows ON alphas receive cholinergic starburst input?), across spot size and over a range of voltages (to ensure linearity), would more conclusively reveal whether or not the glutamate input properties are similar. That the bipolar input is shared between these two RGC types would require a ‘noise’ analysis. For example, spontaneous EPSCs recorded simultaneously from pairs of PixON and ON alpha RGCs with overlapping receptive fields could be used to directly demonstrate common input (e.g. Trong and Rieke, 2008). Alternatively, direct stimulation of bipolar cells with ChR2 (using the in-house CCK-cre line that labels T6s) could demonstrate that the same T6s drive these RGCs (the trial-to-trial amplitude fluctuations of the optogenetically-evoked responses would be expected to co-vary). Albeit technically challenging, these experiments are well within the expertise of this lab and would go a long way in solidifying the base on which the primary hypothesis rests on.

This is an important point. We have several responses.

- a. Interpretation of experiments probing for functionally common input is complicated as we are suggesting that common bipolar cells could output functionally distinct signals (although we performed these experiments, as outlined in **point d** below). Do we expect the synaptic noise to be correlated between these functionally distinct outputs from the same BC? Will it depend on

stimulus conditions? We do not know whether signal and noise have the same functional divergence properties.

- b. Stimulation of T6 BCs with ChR2 would indeed be an experiment to provide some functional evidence of shared BC input. We attempted many different viral constructs for expressing ChR2 (and iGluSnFR and GCaMP6f) in BCs, and all failed. We could breed a CCK-Cre x ChR2-flox transgenic line, but the CCK-promoter is quite promiscuous. A huge number of ACs and some RGCs are also labeled in the reporter line, including almost all the ON starburst ACs. Thus, activation with blue light would engage a heterogeneous circuit. Furthermore, since nearly all ON BCs are coupled through All ACs, strong activation of any single population of them would likely elicit spreading activation to other BC types, complicating the interpretation of such a result. In short, while this experiment could provide a bit of evidence of shared input that complements our EM data, we feel that the EM is easier to interpret since it is more direct evidence of synaptic connectivity.
- c. We investigated differing EPSC kinetics (**Reviewer Fig. 2**). Indeed, Pix_{ON} EPSCs do tend to be more transient than ON alpha EPSCs, although it should be noted that these populations are highly overlapping. We repeated measurements of Pix_{ON} EPSCs with bath-applied drugs to block inhibition, nACh receptors, and NMDA receptors (strychnine, gabazine, saclofen, TPMPA, Hexamethonium, TTX, D-AP5) to prevent these possible mechanisms of functional divergence; however, the Pix_{ON} EPSCs kinetics remained similar to control conditions. We thus do not have an explanation for why the EPSC kinetics differ, although it should be noted that we were unable to control for the many other possible mechanisms that could lead to functional divergence at the BC-RGC synapse (see **Discussion**; voltage-gated Ca²⁺ channel subtypes, peptide transmitters, calcium or glutamate clearance, etc.).

Reviewer Fig. 2 | Kinetics of excitatory conductances differ between Pix_{ON} and ON alpha RGCs.

a, Average excitatory conductance of Pix_{ON} RGCs (purple, n=37) and ON alpha RGCs (brown, n=21). Gray bar indicates 1s stimulation with a 220 μm diameter spot of light. Conductances from each cell were normalized by the maximum conductance value recorded for that cell. Shaded region indicates the standard error of the mean. **b**, The magnitude of the excitatory conductance 1s post-stimulus onset normalized as in **a**. Dots indicate individual Pix_{ON} (purple, n=37) and ON alpha (brown, n=21) RGCs. The black bar indicates the average and gray bars indicate the standard error of the mean. **c,d**, Same as **a,b**, but comparing Pix_{ON} excitatory

conductances in control conditions (purple) to Pix_{ON} excitatory conductances with bath application of a drug cocktail meant to block feedback from amacrine cells, nACh receptors, and NMDA receptors (strychnine, gabazine, saclofen, TPMPA, Hexamethonium, TTX, D-AP5; red, n=3).

- d. We additionally performed the suggested noise analysis between Pix_{ON} and ON alpha RGCs with overlapping dendrites (**Reviewer Fig. 3**). We found that most Pix_{ON} and ON alpha RGCs had little or no noise correlation, although one Pix_{ON} - ON alpha pair with a high degree of dendritic overlap did have a high noise correlation. Additionally, we tested other ON RGC control pairs that shared photoreceptors, but are not expected to share bipolar cells (Pix_{ON} - ON DS, ON alpha - ON DS). These control pairs also had very little noise correlation. These results were somewhat surprising as Trong and Rieke (2008) observed noise correlations due to shared photoreceptors, even among RGCs of opposite polarity which did not share the same bipolar cell input. It should be noted, however, that this study was performed in primate retina, which might have different levels of localized regulation of its bipolar cells. Ultimately, interpretation continues to be difficult for these experiments when viewed in the context of functionally divergent signals from common bipolar cells.

Reviewer Fig. 3 | Low noise correlation between Pix_{ON} and ON alpha excitatory conductances.

a, Example paired voltage clamp recordings of excitatory conductances of a Pix_{ON} RGC (purple) and an ON alpha RGC (brown) which have overlapping dendritic arbors. **b**, Autocorrelation and cross-correlation of cells in **a**. Autocorrelation of the Pix_{ON} (purple) and ON alpha (brown) were corrected for common noise by subtracting the cross-correlation (black). Dotted line indicates shuffled control of the cross-correlation. **c,d**, same as **a,b**, but for a Pix_{ON} - ON alpha pair without correlated noise. **e,f**, Same as **a,b**, but for a control pair of cells (Pix_{ON} and ON DS), which are expected to share photoreceptors, but not bipolar cells. **g**, Excitatory noise cross-correlation for Pix_{ON}-ON alpha RGC pairs (circles), Pix_{ON} - ON DS RGC pairs (plus sign), and ON alpha - ON DS RGC pairs (triangles). Dendritic overlap indicates the area of the cells' dendritic arbor which overlapped.

- Common input (anatomical)? In the same vein as point 2, the complementary anatomical analysis needs to be conducted more rigorously. For instance, the authors found the same type 6 bipolar cell could synapse onto both PixON and ON alpha ganglion cells. What fraction of the total synapses to each ganglion cell came from such common bipolar cells? The authors count a total of 8 bipolar cells that synapse with both cells, but the reader has no sense of whether this is a high/relevant number compared to the total number of synapses. In general, more description of the EM data would be valuable, especially in the regions of dendritic overlap between the ganglion cells.

We have thought quite a bit about how to add more quantification of the BCs synapsing onto both RGCs. Unfortunately, we are limited in some fundamental ways by the EM volume and the fact that we only traced one of each RGC type.

What fraction of the total synapses to each ganglion cell came from such common bipolar cells?

We analyzed BCs within the overlapping area of Pix_{ON} and ON alpha dendrites, for which we identified multiple ribbon synapses. We found that 5 of the 10 T6s and 5 of the 7 T7s synapsed with both the Pix_{ON} and the ON alpha. However, our reconstruction included only a single Pix_{ON} and a single ON alpha, and the coverage factor of each of these RGC types is between 3 and 4 (ref. ³). Thus, this is surely an underestimate of the fraction of BCs that synapse onto both types of RGCs. Additionally, our reconstruction only contained a small area of dendritic overlap between the RGCs, thus the power of this analysis is limited.

To investigate BC preference for Pix_{ON} and ON alpha RGCs while taking into account the real RGC coverage factor, we analyzed the EyeWire SBFSEM reconstruction (**Supplementary Fig. 6**). While we could not identify synapses in this dataset, previous studies have found a strong relationship between BC axon - RGC dendrite overlap and the number of BC - RGC synapses². We found that all of the T6 BCs (n=47) and 97% of the T7 BCs (n=29) had axons that overlapped with both Pix_{ON} and ON alpha RGCs. We additionally analyzed the relative preference each BC had for Pix_{ON} vs. ON alpha RGCs and found that their preferences were not different from a shuffled control.

3. In the cases where a bipolar cell provided output to both ganglion cells, how far away were those synapses from each other? The model relies on synapses in the first and last quartile of voltage decay, which presumably correspond to synapses that are furthest away from each other or have the most branched path between each other. Is this realistic based on the synapses identified in the EM data set? Since we have ground truth data in the EM, the distances between PixON and ON alpha synapses used in the model should correspond to the distances found in the EM data set.

We measured the path distance between the ribbons of the first and last quartile (**Supplemental Fig. 12h**). Indeed, these distances tended to be larger than the pairwise distance between all ribbon synapses in the model. These values were also greater than what we measured in the SBFSEM reconstruction. We do agree that this calls into doubt whether the kind of functionally specific synapse formation necessary for a voltage-mediated mechanism actually occurs in this circuit. We try to make this clear in the text and discuss other mechanisms (chemical) that would not require this level of spatially segregated synapses [line 551].

Minor Criticisms

1. Ganglion cell type identification: It is not entirely clear what morphological functional criteria were used to ascertain recordings were made specifically from PixON and ON alpha RGCs since these parameters are known to vary greatly across the retina. There is a particular concern for the PixON type, as many of their functional properties reported here do not appear to match previous studies in which these cells have been characterized in genetically tagged RGCs. For example, the profound surrounds of the PixONs are not apparent in work from Kerschensteiner's group (In fact, the response profiles to increasing spots size appear more like the ON alphas shown here). In addition, the contrast response functions (CRFs) were also previously reported to be linear, in contrast to the rectifying CRFs shown in this study. The authors should discuss these differences and provide a more in-depth description of how they identified RGC type and perhaps consider using other functional properties to confirm their identities (such as blue/green color opponency).

Independent of surround suppression, we distinguished Pix_{ON} RGCs from ON alpha RGCs by their characteristic spike amplitude adaptation for spot sizes near the peak firing rate (**Reviewer Fig. 4**). We also used soma size as an indicator of cell type; Pix_{ON} somas are smaller than those of ON alphas (**Supplementary Fig. 1c**).

Indeed, the Pix_{ON} responses we report here and in our previous paper⁴ are different in some respects from those published by the Kerschensteiner lab in the Grik4 line⁵, and by the Berson lab for RGCs they called M5⁶. We suspect that experimental differences in the preparation between labs account for much of this difference, but this would have to be verified with additional experiments across labs. Noted differences in methods from the paper by the Kerschensteiner lab⁵ include: (1) We used Ames medium, and they used ACSF. (2) Our cells were mostly recorded from darkness (though some were in photopic conditions in **Supplementary Fig. 2**), while theirs were all on a background illumination of 1500 R*/rod/s. (3) We used WT animals while they used the Grik4 line. While we have no specific reason to expect the Cre insertion in this line to alter light responses, our experience with this line was that it labeled several RGC types in addition to Pix_{ON} and that the light responses were inconsistent. (4) We targeted the cells under 950 nm LED illumination in WT animals while they used 2P targeting, potentially causing some degree of light adaptation prior to recording.

The claim of color opponency in M5 RGCs remains controversial. We did not find evidence for color opponency in Pix_{ON} RGCs recorded in WT or in Grik4-Cre. Tiffany Schmidt's lab has shown that the classification of M5 RGCs in the initial report from David Berson's lab may have included M4 RGCs in the temporal retina and that color opponency may be an artifact of unphysiological bleach, especially in cells targeted under 1P illumination⁷.

Reviewer Fig. 4 | Strong spike amplitude adaptation in Pix_{ON} RGC responses but not ON alpha RGC responses.

a, Cell attached recordings of three different Pix_{ON} RGCs in response to a preferred size spot stimulus. Gray bar indicates 1s stimulus presentation. **b**, Same as **a**, but spiking response recorded from three different ON alpha RGCs.

- Dynamic clamp: E/I interactions occur on the dendrites and it is not clear to what extent the somatic dynamic clamp experiments capture this behavior. The results would be more compelling if this issue was tested experimentally. For example, how does stimulating surround inhibition with an annulus affect the responses generated by a somatic 'dynamic' excitatory conductance? Or conversely, how is the center spiking responses evoked by a small spot affected by artificially applying inhibitory conductances at the soma?

These are interesting ideas, and indeed dynamic clamp is an imperfect tool for understanding E/I interactions due to the conductances being supplied to the soma. While it can be tweaked by the methods suggested, it is fundamentally limited in providing a strong quantitative bound on the contribution of E vs. I to a spiking response. Thus, we have been careful in the updated manuscript to point out its limitations [line 126].

However, the dynamic clamp experiments are not foundational to the key findings of the paper. Our voltage clamp measurement of surround suppression of excitatory conductances in conjunction with EM reconstructions of BC input provides evidence that subcellular regulation is occurring presynaptic to the RGCs. The dynamic clamp experiments attempted to estimate the relative impact that this might have on spiking output but are not needed for the key conclusion of subcellular regulation.

3. Computational model: In the model, the dendritic excitation appears to be a step change in conductance, which is then inhibited by a step change in inhibitory conductance in the axon terminals. The level of suppression at each output synapse is estimated by its steady-state membrane potential after inhibition is activated. If the amplitude of the hyperpolarization is decaying by as much as 50% in the active model (Fig. 6b), I wonder about the lengthening of the kinetics of the hyperpolarization that will also occur as voltage spreads down the cable to the other synapse. By using step changes in activity, it's difficult to assess how the kinetics and timing of inputs will affect glutamate release. Is it possible that by the time the hyperpolarization has reached the distant synapse, it has a slow enough rising phase that the depolarization from the excitatory input is unable to be blocked, especially in the presence of fast active conductances? The authors could use a more stochastic, event-based model to test how inhibitory events spread to other parts of the axonal arbor. It is expected that steady, tonic voltage changes will permeate the arbor more effectively than brief, event-based activity. It's also possible that the excitatory drive from photoreceptors is more tonic, while the NaV-dependent inhibition is briefer and event-based. I think using more realistic synaptic activity patterns is important; it could end up supporting the authors' model even further.

Based on this suggestion, we implemented a new version of the model with stochastic activation of synapses with realistic kinetics (**Fig. 7b,c** and line 821 for implementation details). We also explicitly examined the frequency dependence of voltage attenuation in our model by injecting sine waves at different frequencies and measuring the length constant (**Supplementary Fig. 11**). These results showed that substantial attenuation only occurs at very high frequencies and that adding excitation and inhibition decreases the overall resistance substantially, making the BC largely frequency independent in terms of voltage attenuation.

4. Alternative models: The authors are careful not to overstate their results and do consider the possibility of chemical compartmentalization. Bipolar cells may express L and T-type Ca channels with distinct voltage-dependent properties (and both channel types support release; Pan et al., 2001; Neuron). In theory, a differential expression of these channels could trigger heterogeneous release during a global hyperpolarization of the terminal (owing to the higher threshold required for L-Type Ca channel activation). Alternatively, different Ca²⁺ buffering could lead to similar effects. Such mechanisms might be faster and more plausible than the second messenger-dependent mechanisms suggested in the discussion.

Thank you. We have augmented this part of the **Discussion** with speculation about different Ca²⁺ channel types or different local buffering.

5. Inhibition at on alpha synapses: In the case that the author's model is accurate—where PixON synapses and ON alpha synapses are electrotonically separated enough to drive different levels of glutamate release—it is still the case that in the EM data, there was always an inhibitory synapse within a micron of each ON alpha synapse. Since these are all wide-field amacrine inputs, wouldn't these also be active during a surround-encompassing visual stimulus? By what mechanism would only the PixON amacrine inputs be active?

We were very limited in our ability to determine the typology of the ACs due to the limited size of our SBFSEM volume (80 x 150 μm). We categorized ACs as medium to wide-field based on neurite spans of >40 μm. However, 40% of the ACs could not be classified in this manner because less than 40 μm of their neurite was present in the volume. Even if all ACs are wide-field, it remains possible that the ACs present at BC synapses to Pix_{ON} vs. ON alpha RGCs differ both in functional properties (whether they have surround suppression) and in the transmission of additional peptides beyond their assumed GABA release. Alternatively, as Reviewer 2 points out in the previous suggestion, there could be

presynaptic differences in either Ca^{2+} channel types or local Ca^{2+} buffering. We have tried to make these points more clearly in the updated text.

6. Nano-scale compartmentalization: The idea that the compartmentalization might be caused by the fact that inhibition is closer to the PixON synapses (line 317) seems a bit far-fetched since we're on the scale of $0.75\ \mu\text{m}$ to $1\ \mu\text{m}$.

We agree that the compartmentalization of voltage at this scale is unfeasible and have attempted to make this clear in the text [line 338]. We discuss the possibility that differing distances to inhibitory synapses could indicate some other functionally relevant difference (e.g. different AC types, neurotransmitter, or receptor type). However, we do believe that chemical compartmentalization could occur on the $1\ \mu\text{m}$ scale. We discuss the possibility of Ca^{2+} compartmentalization as $20\ \text{nm}$ Ca^{2+} nanodomains have been suggested to occur at BC ribbon output synapses [ref. ⁸ and line 563].

7. Inhibitory shunt? It would be good to discuss how one expects the inhibitory conductance itself to act at the site of glutamate release. Is there any significant shunting effect? The length constant of an open channel shunt is expected to be half that of voltage spread, which would be well-suited to help compartmentalize inhibitory activity.

We tested explicitly the degree to which the shunt from the synaptic conductances alters voltage transmission within the model BC (**Supplementary Fig. 11**). Indeed, this is a large effect, shortening the effective length constant from over $500\ \mu\text{m}$ to $\sim 70\ \mu\text{m}$. This point is now emphasized in the text [line 430].

8. Figure 3: Figure 3 is hard to interpret. Blocking glutamate receptors is likely to remove inhibition and, therefore, could increase glutamate levels. Is there any positive control to be done to confirm that the NBQX/Kyn experiment is capable of revealing whether saturation or desensitization is responsible for the weak surround suppression?

We have not been able to come up with an experiment here that would be substantially easier to interpret. We agree that NBQX/Kyn will substantially reduce inhibition, but both treatments were effective in decreasing the excitatory current substantially in ON alpha RGCs despite a possible release from inhibition in the BC terminals (**Fig. 3b-d**). The proposed (null hypothesis) model by which saturation or desensitization could account for the apparent difference in surround suppression is essentially a ceiling effect where excitatory current saturates in ON alpha RGCs. We find no evidence for such a saturation effect since surround suppression was similar with a much smaller current. We have included some text about the limitations of this experiment in **Results** [line 171].

9. Contribution of postsynaptic inhibition: The authors claim that it is the patterns of excitation that dominate surround inhibition rather than postsynaptic inhibition. This could easily be tested by using GABA_A to block inhibition (since presynaptic inhibition is mediated by GABA_B, excitation would not be disturbed as shown in Fig. 5a).

We performed the suggested experiment, and the results are shown in **Reviewer Figure 5**. We observed a partial effect. Gabazine decreases surround suppression in Pix_{ON} RGCs (likely from its effects on postsynaptic inhibition), but it does not eliminate it entirely, probably because the GABA_C-mediated surround suppression in excitation remains. We hesitate to interpret the size of this change more quantitatively because of the inherent limitations of bath-applied pharmacology (off-target effects on unknown circuits), but this qualitative result agrees with our conclusion that both presynaptic and postsynaptic sources contribute to surround suppression in the spike responses of Pix_{ON} RGCs. As noted in point 2 above, the extent to which surround suppression in excitation contributes to surround suppression in the spike responses of Pix_{ON} RGCs is not central to our main conclusions.

Reviewer Fig. 5 | Gabazine reduces surround suppression of Pix_{ON} spiking response.

a, Example peristimulus time histograms recorded from a Pix_{ON} RGC in response to preferred size (dark line) and full-field (lighter line) light spot stimuli in control conditions (bottom) and during bath application of Gabazine (10 μ M). The gray horizontal bar indicates the 1-second presentation of the 250 R*/rod/s spot stimulus. **b**, Example spike rates recorded from a Pix_{ON} RGC in response to a range of spot sizes in control conditions (black) and during bath application of Gabazine (red). **c**, Surround suppression of spiking responses in control and Gabazine conditions. Dots indicate data from individual Pix_{ON} RGCs. Bar plots indicate average \pm s.e.m.

10. Typos: Figure 2 and 3 legends have some mislabeling with the lettering.

Fixed.

Reviewer #3

Minor Criticisms

1. I was a little unclear on the precise goal and limitations of the bipolar cell model predicting the differences in surround suppression between the PixON and ON alpha (page 9). The model suggests a difference in bipolar cell input generates the distinct responses of the two cells. This difference could be due to differences in the types of BC input or differences in the nature of a single BC type's input to each RGC. The model fits excitatory responses produced by all presynaptic bipolar cell types, but the bipolar cell subunits are constrained to have the center receptive field size of a Type 6 bipolar cell. - If the goal was to look specifically at Type 6 bipolar cells, how much of a limitation is it that the receptive fields are being fit to a response generated by all presynaptic bipolar cell types? Could a difference in the relative amount or strength of the different presynaptic bipolar cell types to the PixON vs. ON alpha RGCs produce a difference in the surround size and CSR simply because the model is doing its best to predict a response that includes non-Type 6 BCs? - If the goal was to look at differences in bipolar cell types, how much of a limitation is it that the center receptive field size is fixed to that of a Type 6 BC?

Indeed, while our EM data allowed us to count the number of synapses from each BC type, it did not offer information about synaptic strength. Previous work has suggested a strong correspondence between anatomical synapse counts and the amplitude of excitatory currents², but it remains a possibility that an imbalance between the T6 BC and T7 BC input to these two RGCs contributes to some degree to the different surround suppression in their excitation. We think it is highly unlikely that an imbalance large enough to produce the difference we measured in the excitation to the two RGC types has no anatomical signature we could measure at the EM level.

As for the RF size of the BCs themselves, we expect T6 and T7 to be quite similar due to their similar dendritic area⁹, and they have been measured to be similar in glutamate imaging studies^{10,11}. Indeed, T8 and T9 BCs are substantially larger, but since they make up such a small fraction of the input, they are unlikely to make a strong contribution to the overall excitatory drive. It is worth noting here as well that iGluSnFR measurements of both the receptive field properties (RF center size, RF surround size,

and CSR) and surround suppression of mouse BCs show a rather narrow range of values between types^{10,11}.

2. Stabio et al (2017) reported cone opponency in the M5/PixON RGCs. Is it possible that this opponency could account for any of the surround suppression measured physiologically rather than Type 6 BC input? Would any of the results be expected to change with the spectral composition of the stimulus? Other readers might share my confusion in navigating cone opponency results from a species with an opsin gradient and appreciate explanations from the authors (no additional experiments needed).

The claim of color opponency in M5 RGCs remains controversial. We did not find evidence for color opponency in Pix_{ON} RGCs recorded in WT or in Grik4-Cre (the line that Daniel Kerschensteiner's lab used to target them⁵). Tiffany Schmidt's lab has shown that the classification of M5 RGCs in the initial report from David Berson's lab⁶ may have included M4 RGCs in the temporal retina and that color opponency may be an artifact of unphysiological bleach, especially in cells targeted under 1P illumination⁷.

3. I particularly enjoyed the series of dynamic clamp experiments and found them to be very compelling. I was curious about the dip in spiking after the transient response at stimulus onset with the ON alpha RGC conductances (Fig 2b and d) which was not present in Fig 1c. Why might this occur?

We did see a dip in spiking rate when simulating ON alpha conductances with dynamic clamp, but we do not see this when recording spiking responses to visual stimuli. We fully admit that dynamic clamp simulations at the soma are not the same as current flow through synapses in the dendrites. Excitation/inhibition interactions can occur in dendrites, and it seems that the steep onset of ON alpha inhibition to full-field stimuli is filtered out during dendritic integration. Regardless, we do think that the dynamic clamp experiments show that surround suppression of excitatory conductances is likely to play a role in generating surround suppression of the spiking response. Additionally, our key conclusion of subcellular regulation does not depend upon the dynamic clamp experiments.

4. Page 14 – “We traced the presynaptic ACs at each type 6 BC ribbon” reads as if all the presynaptic ACs near a ribbon were reconstructed, but the Fig 5c and 5e legends specify only the nearest amacrine cell was reconstructed. Regardless, the possibility that amacrine cells target ribbon synapses presynaptic to a specific RGC type is very intriguing.

Fixed.

References

1. Demb, J.B., Zaghoul, K.A., Haarsma, L., and Sterling, P. (2001). Bipolar cells contribute to nonlinear spatial summation in the brisk-transient (Y) ganglion cell in mammalian retina. *J. Neurosci.* 21, 7447–7454.
2. Schwartz, G.W., Okawa, H., Dunn, F.A., Morgan, J.L., Kerschensteiner, D., Wong, R.O., and Rieke, F. (2012). The spatial structure of a nonlinear receptive field. *Nature Neuroscience* 15, 1572–1580. 10.1038/nn.3225.
3. Bae, J.A., Mu, S., Kim, J.S., Turner, N.L., Tartavull, I., Kemnitz, N., Jordan, C.S., Norton, A.D., Silversmith, W.M., Prentki, R., et al. (2018). Digital Museum of Retinal Ganglion Cells with Dense Anatomy and Physiology. *Cell* 173, 1293–1306.e19.
4. Goetz, J., Jessen, Z.F., Jacobi, A., Mani, A., Cooler, S., Greer, D., Kadri, S., Segal, J., Shekhar, K., Sanes, J.R., et al. (2022). Unified classification of mouse retinal ganglion cells using function, morphology, and gene expression. *Cell Rep.* 40, 111040.
5. Johnson, K.P., Zhao, L., and Kerschensteiner, D. (2018). A Pixel-Encoder Retinal Ganglion Cell with Spatially Offset Excitatory and Inhibitory Receptive Fields. *Cell Rep.* 22, 1462–1472.

6. Stabio, M.E., Sabbah, S., Quattrochi, L.E., Ilardi, M.C., Fogerson, P.M., Leyrer, M.L., Kim, M.T., Kim, I., Schiel, M., Renna, J.M., et al. (2018). The M5 Cell: A Color-Opponent Intrinsically Photosensitive Retinal Ganglion Cell. *Neuron* 97, 251.
7. Sonoda, T., Okabe, Y., and Schmidt, T.M. (2020). Overlapping morphological and functional properties between M4 and M5 intrinsically photosensitive retinal ganglion cells. *J. Comp. Neurol.* 528, 1028–1040.
8. Jarsky, T., Tian, M., and Singer, J.H. (2010). Nanodomain control of exocytosis is responsible for the signaling capability of a retinal ribbon synapse. *J. Neurosci.* 30, 11885–11895.
9. Wassle, H., Puller, C., Muller, F., and Haverkamp, S. (2009). Cone Contacts, Mosaics, and Territories of Bipolar Cells in the Mouse Retina. *Journal of Neuroscience* 29, 106–117.
10. Franke, K., Berens, P., Schubert, T., Bethge, M., Euler, T., and Baden, T. (2017). Inhibition decorrelates visual feature representations in the inner retina. *Nature* 542, 439–444.
11. Strauss, S., Korympidou, M.M., Ran, Y., Franke, K., Schubert, T., Baden, T., Berens, P., Euler, T., and Vlasits, A.L. (2022). Center-surround interactions underlie bipolar cell motion sensitivity in the mouse retina. *Nat. Commun.* 13, 5574.

REVIEWER COMMENTS

Reviewer #1 (Remarks to the Author):

D Swygart, W-Q Yu, S Takeuchi, RROL Wong & GW Schwartz.

“A presynaptic source drives differing levels of surround suppression in two mouse retinal ganglion cell types”. Nature Communications (NCOMMS-22-46416-T, revision 1).

In the revised version of their paper, Swygart et al. have adequately addressed most of the points of my original review.

The main challenge with their study, as presented in the revised MS, is that the confidence in the major conclusion is weakened. After carefully reading the revised MS, the supplementary information and the reviews together with the authors' response / rebuttal, I get the distinct impression that the authors themselves are ready to drop the core hypothesis that electrical compartmentalization in cone bipolar axon terminals can explain the observed differential surround suppression. The authors have done a large amount of very impressive work, but at the end they are ready to suggest that chemical compartmentalization is, after all, a more likely mechanism. Unfortunately, whereas their discussion of possible mechanisms of chemical compartmentalization is quite reasonable, such mechanisms were not explored in the current study.

Another way of thinking about this is that the authors have potentially found a mechanism of electrical compartmentalization that could work, given certain specific circumstances, but for the particular case they have studied, it seems unlikely to be the responsible mechanism. Whereas this view comes clearly across in the authors' response to the original reviews, and to some extent also in the Discussion of the MS, it does not come across at all in the Abstract and other places in the MS. This is unfortunate.

Below I have added some specific comments that hopefully can help the authors further improve their MS.

MAJOR COMMENTS / CRITICISMS:

1. I find it hard to understand how the authors interpret the results for the sensitivity to inhibition of different ribbons in the bipolar cell axon terminal. In lines 436-439, the authors state that they examined the anatomical features that influenced this sensitivity, expressed as the CSR value. Their conclusion is that ribbons nearer to the activated inhibitory synapses tended to have lower CSR values (“stronger effective inhibition”) than those further from the activated inhibitory synapses (which had higher CSR values, “weaker effective inhibition”). Given that the set of activated inhibitory synapses can vary, unless all are activated simultaneously, this must mean that individual ribbons cannot be assigned to either low or high CSR values, i.e., a low or high CSR value cannot be a non-variant property of any given ribbon. If inhibitory synapses close to a given ribbon are activated, a low CSR value will result and if inhibitory synapses further away are activated, a larger CSR value will result. In contrast to this dynamically varying situation, the wiring must be considered fixed. If the authors believe that this interpretation is incorrect,

they should attempt to describe their results / analysis in a way that avoids any possibility of such misunderstanding.

MINOR CRITICISMS:

1. In my original review, I challenged the authors to provide concrete support for their statement "...the commonly held view of bipolar cells as electrically compact neurons that constitute single information channels". In the revised MS (see lines 37-38, lines 507-508, line 536), the authors provide four references (4, 10-12). The first, ref. 4 is a review by Masland (2012). In his review, Masland clearly makes a statement that bipolar cells are electrically compact, but, interestingly, does not add any references to support this statement. The three papers of ref. 10-12 are all from the same laboratory (Euler). In none of these papers could I find phrases like "compact", "electrotonic" or "isopotential". One of papers, a review from 2014 (Euler et al., Nat. Rev. Neurosci.) actually devotes a section to a discussion of ideas that are fully in line with the core hypothesis of the current study (and could more appropriately be added to the references for their statement later in the Discussion, line 621). In one place in their paper, Euler et al. actually writes "Far from being isopotential neurons that simply relay the photoreceptor signal to the inner retina,...". This hardly justifies the way the authors have phrased their argument and my conclusion is that unless they can come up with better support, it must be seen as a straw man argument. Perhaps a more reasonable assessment of the scientific literature would be that there are reports of electrical filtering, potentially of functional relevance, but that nobody had so far studied electrical compartmentalization in the axon terminals of cone bipolar cells.

2. The authors consistently refer to their bipolar cell model as a cable model. Strictly speaking it is a compartmental cable model.

3. Lines 145-149: Instead of using terms like "weak" and "strong" with respect to the glutamate receptor antagonists, it might be preferable to use terms like low- and high-affinity antagonists, as it is precisely the difference in on and off rates that is exploited in the experiments.

4. Lines 151-155:

a) When the authors write that "Kyn is expected to reduce glutamate receptor desensitization and saturation", please clarify if this is just a theoretical speculation or whether it can be backed up by direct experimental evidence (based on earlier studies).

b) Whereas reduced desensitization can be clearly expressed in terms of kinetic mechanisms (reduced probability of entering and/or increased probability of leaving desensitized states), I wonder if the authors can make the phrase "reduced saturation" more precise?

c) Whereas it makes sense to state that binding of Kyn to a glutamate receptor molecule can protect it from desensitization, it is unclear to me that it makes sense to state that binding of Kyn to a glutamate receptor molecule can protect it from saturation. Can a single receptor molecule be saturated? Can the authors clarify this and increase the level of precision?

5. Lines 190-192: “their dendritic arbors” refers back to “excitatory conductances”, which does not make sense.

6. Line 193: Here and elsewhere, I would suggest that when the authors use phrases like “larger dendritic arbors”, they should make it clear that they refer to the size of the dendritic field.

7. Lines 194-196: Please explain more clearly how larger dendritic arbors / trees would cancel out bipolar cell subunit inhibitory surrounds.

8. Line 209: “accurately” seems overly optimistic. Perhaps “roughly” is more appropriate.

9. Lines 297-300: The authors discuss “GABAC receptors clustered at cone bipolar cell output synapses”, but ref. 33 refers to a study of rod bipolar cells (Chavez et al. 2010).

10. Lines 368-370: The authors refer to “multiple studies that have measured voltage-gated ion conductances from bipolar cells”, but several of the cited studies are modeling studies which did not measure any currents.

11. Lines 414-416: How can the authors explain that increasing the leak conductance (in the passive model), to match the increased conductance evoked by voltage-gated channels, did not increase the “center-surround ratio” range to the same extent? Instead of matching by conductance, they could perhaps use the attenuation or length constant as a matching criterion when increasing the leak conductance of the passive model, to see if this increased the “center-surround ratio” range further?

12. Line 474: Here and elsewhere, the authors use the phrase “inhibitory decay”. If I understand it correctly, in the context of their presentation this expression refers to reduced depolarization. However, “decay” suggests a time-varying reduction, in this case of inhibition. As time is not relevant here, however, I suggest that they write “reduction” instead of decay and state clearly what was reduced.

13. Line 523: Leach should be leech.

14. Lines 574-576: The authors write as if the finding of PIP2-dependent Ca²⁺ release from internal stores was obtained for bipolar cells, but the reference provided (81) is for hippocampal neurons.

15. Line 579: “reverse agonists” shall presumably be “inverse agonists”.

16. Lines 633-636: The acknowledgement that the more refined analysis in ref. 93 found evidence for functional divergence in bipolar cells seems to contradict the authors’ rejection earlier in the Discussion that investigating potential functional divergence with imaging using e.g. iGluSnFR or Ca²⁺ has any chance of being successful.

17. Line 655: Which buffer was used for the oxygenated Ames medium?

18. Line 719: What was the concentration of AlexaFluor 488 in the patch pipette solution?

19. Line 742: Clarify if “dendritic diameter” is “dendritic field diameter”.

20. Lines 759-760: Which excitation wavelength and which filters were used for the simultaneous two-photon imaging of Alexa 488 (in RGCs) and labeled T6 bipolar cells?

21. Lines 821-822: Should “...at the RGC dendrites” be “at the BC dendrites”?

22. Lines 830-831: “its” does not refer back to a specific noun.

23. Table 3:

a) For the row “Inhibition at axon”, it could be a good idea to state explicitly that the model parameters were selected to mimic GABAC inhibition.

b) For the row “Kv+ channel”, it is unclear what ref. 132 is used for and why Kv1.2 is stated in the reference. The web address as stated in ref. 132 is a non-existent page.

FIGURES:

1. Fig. 1a: The legend states that each illustrated cell is a “maximum intensity z-projection”. Is this correct? To me they look more like shape plots from reconstructions.

2. Fig. 3a: The color codes for the square waveforms in the upper right panel (“Exc”) are missing.

3. Fig. 4d: The legend states that the “Model suppression” is the average predicted surround suppression. Given that each point in the graph represents a single cell, it is not clear to me what “average” refers to.

4. Figure 5, Figure 6:

Fig. 5d, e: The legend refers to both “ribbon synapse” and “ribbon synapses”, making it unclear how many ribbon synapses are illustrated in these panels.

Fig. 5e, g: The red arrow head presumably points to a ribbon. For panel e, I notice that the arrow head does not point to the same structure that it did in the original MS. It would be helpful if the authors outlined the structures they believe correspond to a ribbon (perhaps in a supplementary figure).

Fig. 5h: The legend says “cross-sectional view” for the bottom subpanel, but to me it looks more like a projection, orthogonal to the orientation in the top panel.

See also Fig. 6d for the same problem.

Fig. 5j: It could be helpful to add “synapses” after “n=86” and “n=50”.

5. Figure 8:

Fig. 8a: The third line of the legend refers to Q2 instead of Q4.

Fig. 8d: The legend refers to a value of 1.7 for the average of Q4 from panel a. To me, the average of Q4 in panel a looks more like 2.25 or so.

SUPPLEMENTARY FIGURES

Suppl. Fig. 1:

a) For panel d and legend, please clarify if “dendritic diameter” is “dendritic field diameter”. Please state explicitly how “dendritic field diameter” was calculated.

b) Please clarify if “dendritic area” is “dendritic field area”.

c) Please define “branch length”.

d) Please define “branch”.

Suppl. Fig. 3: Please consider if a grayscale is better suited for adequate discrimination of the different dots.

Suppl. Fig. 5: Please define “slope” and “contrast”.

Suppl. Fig. 6: For panel d, I could not find an explanation of how “overlap” was calculated”.

Suppl. Fig. 7: How were PSD95 puncta belonging to the filled RGC identified?

Suppl. Fig. 9: The legend says “see Fig. 6”, but this should presumably be “see Fig. 7”.

Suppl. Fig. 10: The legend says “as shown in Fig. 6”, but this should presumably be “as shown in Fig. 7”.

Suppl. Fig. 11: The legend says “To maintain a voltage equivalent to previous simulations (-38 mV, Fig. 6)”. This is confusing. First, “Fig. 6” should presumably be “Fig. 7”. Second, in Fig. 7, it is unclear for what “-38 mV” is a relevant number.

Suppl. Fig. 12: The legend says “see Fig. 6”, but this should presumably be “see Fig. 7”. The legend says “see Fig. 4”, but this should presumably be “see Fig. 5”.

Suppl. Fig. 13: “yellow array” should be “yellow arrow”.

Reviewer #2 (Remarks to the Author):

These authors have done a commendable job of addressing the reviewer’s comments in general. However, interpreting the EPSC data remains difficult, for this reviewer. In particular, the new data presented R2 figure, shows that kinetic differences in EPSCs in Pix ON and ON alphas are removed when a cocktail of drugs is applied, raising the possibility that cholinergic input or differences in AMPA/NMDA contribute to the EPSCs. If this is true, the observed differences in surround inhibition could arise through postsynaptic processes rather than presynaptically at the BC terminal as the authors suggest.

Since interpreting EPSCs correctly is key to the central claim of this paper, my suggestions would be to 1) demonstrate that the differences in surround inhibition between the two ON cells exist under conditions in which nAChRs and NMDA receptors are blocked. And 2) show that the EPSCs reverse at 0 mV (in the presence of inhibitory receptor blockers) to rule out the possibility that gap junctions contribute to the EPSCs in ON RGCs (note gap junctions are implicated in shaping the responses of ON RGCs in primate and mouse retina).

Minor comments:

Suggested citations:

Line 38: BC as electrically compact units: Please cite Olstedal et al., 2009 (JPhysiol; model BCs) and

Poleg-Polsky and Diamond 2016 (JNSc; correlated Ca signals from BC terminals);

Grimes et al., 2014 (Elife): This paper shows that ribbon synapses across single RBPCs may have distinct properties and the level of ‘cross synaptic’ synchrony may vary depending on light levels, which supports the central tenet of the current study.

Reviewer #3 (Remarks to the Author):

The authors have fully addressed my original points. Their investigation of functional divergence within a single bipolar cell type is exceptionally thorough and will be of broad interest.

We once again thank the reviewers for their responses and suggestions. We have attempted to address concerns with new experiments and text clarifications. The reviews are copied below with our responses in blue text.

To aid reviewers, we would like to point out the addition of **Supplementary Figure 5** (excitatory I-V curve of $P_{iX_{ON}}$ and ON alpha), which shifts the numbering of all subsequent Supplementary figures by +1.

Reviewer #1

Remarks to the Author

In the revised version of their paper, Swygart et al. have adequately addressed most of the points of my original review.

The main challenge with their study, as presented in the revised MS, is that the confidence in the major conclusion is weakened. After carefully reading the revised MS, the supplementary information, and the reviews together with the authors' response/rebuttal, I get the distinct impression that the authors themselves are ready to drop the core hypothesis that electrical compartmentalization in cone bipolar axon terminals can explain the observed differential surround suppression. The authors have done a large amount of very impressive work, but at the end they are ready to suggest that chemical compartmentalization is, after all, a more likely mechanism. Unfortunately, whereas their discussion of possible mechanisms of chemical compartmentalization is quite reasonable, such mechanisms were not explored in the current study.

Another way of thinking about this is that the authors have potentially found a mechanism of electrical compartmentalization that could work, given certain specific circumstances, but for the particular case they have studied, it seems unlikely to be the responsible mechanism. Whereas this view comes clearly across in the authors' response to the original reviews, and to some extent also in the Discussion of the MS, it does not come across at all in the Abstract and other places in the MS. This is unfortunate.

We agree with how the reviewer has summarized our interpretation of the results. In the revised manuscript, we have been careful not to lead the reader to the assumption that bipolar synapses are necessarily electrically isolated. Electrical isolation is not mentioned in the abstract.

Below I have added some specific comments that hopefully can help the authors further improve their MS.

MAJOR COMMENTS / CRITICISMS:

1. I find it hard to understand how the authors interpret the results for the sensitivity to

inhibition of different ribbons in the bipolar cell axon terminal. In lines 436-439, the authors state that they examined the anatomical features that influenced this sensitivity, expressed as the CSR value. Their conclusion is that ribbons nearer to the activated inhibitory synapses tended to have lower CSR values (“stronger effective inhibition”) than those further from the activated inhibitory synapses (which had higher CSR values, “weaker effective inhibition”). Given that the set of activated inhibitory synapses can vary, unless all are activated simultaneously, this must mean that individual ribbons cannot be assigned to either low or high CSR values, i.e., a low or high CSR value cannot be a non-variant property of any given ribbon. If inhibitory synapses close to a given ribbon are activated, a low CSR value will result and if inhibitory synapses further away are activated, a larger CSR value will result. In contrast to this dynamically varying situation, the wiring must be considered fixed. If the authors believe that this interpretation is incorrect, they should attempt to describe their results / analysis in a way that avoids any possibility of such misunderstanding.

We do consider the wiring to be fixed. However, we are limited in our knowledge of the functional signals carried by this fixed wiring (e.g. which of these inhibitory synapses convey surround suppression). Thus, we attempt to model as wide a range of functional connectivity as possible. We have tried to clarify this point in the text (line 449).

MINOR CRITICISMS:

1. In my original review, I challenged the authors to provide concrete support for their statement “...the commonly held view of bipolar cells as electrically compact neurons that constitute single information channels”. In the revised MS (see lines 37-38, lines 507-508, line 536), the authors provide four references (4, 10-12). The first, ref. 4 is a review by Masland (2012). In his review, Masland clearly makes a statement that bipolar cells are electrically compact, but, interestingly, does not add any references to support this statement. The three papers of ref. 10-12 are all from the same laboratory (Euler). In none of these papers could I find phrases like “compact”, “electrotonic” or “isopotential”. One of papers, a review from 2014 (Euler et al., Nat. Rev. Neurosci.) actually devotes a section to a discussion of ideas that are fully in line with the core hypothesis of the current study (and could more appropriately be added to the references for their statement later in the Discussion, line 621). In one place in their paper, Euler et al. actually writes “Far from being isopotential neurons that simply relay the photoreceptor signal to the inner retina,...”. This hardly justifies the way the authors have phrased their argument and my conclusion is that unless they can come up with better support, it must be seen as a straw man argument. Perhaps a more reasonable assessment of the scientific literature would be that there are reports of electrical filtering, potentially of functional relevance, but that nobody had so far studied electrical compartmentalization in the axon terminals of cone bipolar cells.

2. The authors consistently refer to their bipolar cell model as a cable model. Strictly speaking it is a compartmental cable model.

Clarified in text.

3. Lines 145-149: Instead of using terms like “weak” and “strong” with respect to the glutamate receptor antagonists, it might be preferable to use terms like low- and high-affinity antagonists, as it is precisely the difference in on and off rates that is exploited in the experiments.

Updated terms.

4. Lines 151-155:

a) When the authors write that “Kyn is expected to reduce glutamate receptor desensitization and saturation”, please clarify if this is just a theoretical speculation or whether it can be backed up by direct experimental evidence (based on earlier studies).

This effect is based on an earlier study in the retina which we reference. We also provide a brief discussion of its caveats and limitations in our study (line 177).

b) Whereas reduced desensitization can be clearly expressed in terms of kinetic mechanisms (reduced probability of entering and/or increased probability of leaving desensitized states), I wonder if the authors can make the phrase “reduced saturation” more precise?

Changed text to “prevent saturation”.

c) Whereas it makes sense to state that binding of Kyn to a glutamate receptor molecule can protect it from desensitization, it is unclear to me that it makes sense to state that binding of Kyn to a glutamate receptor molecule can protect it from saturation. Can a single receptor molecule be saturated? Can the authors clarify this and increase the level of precision?

Clarified in the text that competition with Kyn prevents glutamate receptor saturation (line 162)

5. Lines 190-192: “their dendritic arbors” refers back to “excitatory conductances”, which does not make sense.

fixed

6. Line 193: Here and elsewhere, I would suggest that when the authors use phrases like “larger dendritic arbors”, they should make it clear that they refer to the size of the dendritic field.

clarified.

7. Lines 194-196: Please explain more clearly how larger dendritic arbors / trees would cancel out bipolar cell subunit inhibitory surrounds.

We removed reference to this alternative hypothesis. The following modeling does not find evidence for such a mechanism, so further discussion may be misleading.

8. Line 209: “accurately” seems overly optimistic. Perhaps “roughly” is more appropriate.
changed.

9. Lines 297-300: The authors discuss “GABAC receptors clustered at cone bipolar cell output synapses”, but ref. 33 refers to a study of rod bipolar cells (Chavez et al. 2010).
We removed this reference. The other 3 references do discuss cone bipolar cells.

10. Lines 368-370: The authors refer to “multiple studies that have measured voltage-gated ion conductances from bipolar cells”, but several of the cited studies are modeling studies which did not measure any currents.
Removed references to modeling studies.

11. Lines 414-416: How can the authors explain that increasing the leak conductance (in the passive model), to match the increased conductance evoked by voltage-gated channels, did not increase the “center-surround ratio” range to the same extent? Instead of matching by conductance, they could perhaps use the attenuation or length constant as a matching criterion when increasing the leak conductance of the passive model, to see if this increased the “center-surround ratio” range further?
This experiment was specifically designed “To determine if the dynamic properties of the active channels played a role beyond increasing total membrane conductance”. To do this we need to match total membrane conductance.

While we could also match to attenuation or length constant, this is typically our dependent variable and effectively an alternative measure to the center-surround ratio range.

12. Line 474: Here and elsewhere, the authors use the phrase “inhibitory decay”. If I understand it correctly, in the context of their presentation this expression refers to reduced depolarization. However, “decay” suggests a time-varying reduction, in this case of inhibition. As time is not relevant here, however, I suggest that they write “reduction” instead of decay and state clearly what was reduced.
In this context, the reduction of depolarization varies across neuron length. We have added text to clarify that “decay” refers to the exponential decay of voltage along the neuron length (see **Supplementary Fig. 12** and methods for calculation of length constant).

13. Line 523: Leach should be leech.
fixed.

14. Lines 574-576: The authors write as if the finding of PIP2-dependent Ca²⁺ release from internal stores was obtained for bipolar cells, but the reference provided (81) is for hippocampal neurons.
clarified in text.

15. Line 579: “reverse agonists” shall presumably be “inverse agonists”.
fixed.

16. Lines 633-636: The acknowledgement that the more refined analysis in ref. 93 found evidence for functional divergence in bipolar cells seems to contradict the authors’ rejection earlier in the Discussion that investigating potential functional divergence with imaging using e.g. iGluSnFR or Ca²⁺ has any chance of being successful.

Ref. 93 was analyzing iGluSnFR signals. In our discussion of iGluSnFR, we highlight that uncertainty about the origin of the glutamate makes interpretation difficult. This complication to interpretation still holds true for the findings in ref. 93.

17. Line 655: Which buffer was used for the oxygenated Ames medium?
Ames was buffered with sodium bicarbonate. We have clarified in the text and added specific vendor and reagent number.

18. Line 719: What was the concentration of AlexaFluor 488 in the patch pipette solution?
0.2 mM. Clarified in text.

19. Line 742: Clarify if “dendritic diameter” is “dendritic field diameter”.
clarified.

20. Lines 759-760: Which excitation wavelength and which filters were used for the simultaneous two-photon imaging of Alexa 488 (in RGCs) and labeled T6 bipolar cells?
clarified.

21. Lines 821-822: Should “...at the RGC dendrites” be “at the BC dendrites”?
fixed.

22. Lines 830-831: “its” does not refer back to a specific noun.
clarified.

23. Table 3:

a) For the row “Inhibition at axon”, it could be a good idea to state explicitly that the model parameters were selected to mimic GABAC inhibition.

clarified

b) For the row “Kv⁺ channel”, it is unclear what ref. 132 is used for and why Kv1.2 is stated in the reference. The web address as stated in ref. 132 is a non-existent page.

fixed.

FIGURES:

1. Fig. 1a: The legend states that each illustrated cell is a “maximum intensity z-projection”. Is this correct? To me they look more like shape plots from reconstructions.

Clarified. They are z-projections following manual tracing.

2. Fig. 3a: The color codes for the square waveforms in the upper right panel (“Exc”) are missing.

fixed.

3. Fig. 4d: The legend states that the “Model suppression” is the average predicted surround suppression. Given that each point in the graph represents a single cell, it is not clear to me what “average” refers to.

Details are given in “Methods: BC receptive field model”. Four hundred random fitting combinations of the 14 Pix_{ON} and 8 ON alpha RGCs were performed to obtain average cross-validation values. We have attempted to make this clear in the legend.

4. Figure 5, Figure 6:

Fig. 5d, e: The legend refers to both “ribbon synapse” and “ribbon synapses”, making it unclear how many ribbon synapses are illustrated in these panels.

fixed.

Fig. 5e, g: The red arrow head presumably points to a ribbon. For panel e, I notice that the arrow head does not point to the same structure that it did in the original MS. It would be helpful if the authors outlined the structures they believe correspond to a ribbon (perhaps in a supplementary figure).

We have included Supplementary Fig. 14 illustrating the identification of ribbon synapses across SBFSEM z-sections.

Fig. 5h: The legend says “cross-sectional view” for the bottom subpanel, but to me it looks more like a projection, orthogonal to the orientation in the top panel.

See also Fig. 6d for the same problem.

fixed

Fig. 5j: It could be helpful to add “synapses” after “n=86” and “n=50”.

clarified.

5. Figure 8:

Fig. 8a: The third line of the legend refers to Q2 instead of Q4.

fixed.

Fig. 8d: The legend refers to a value of 1.7 for the average of Q4 from panel a. To me, the average of Q4 in panel a looks more like 2.25 or so.

fixed.

SUPPLEMENTARY FIGURES

Suppl. Fig. 1:

a) For panel d and legend, please clarify if “dendritic diameter” is “dendritic field diameter”. Please state explicitly how “dendritic field diameter” was calculated.

clarified.

b) Please clarify if “dendritic area” is “dendritic field area”.

clarified.

c) Please define “branch length”.

clarified in methods.

d) Please define “branch”.

clarified in methods.

Suppl. Fig. 3: Please consider if a grayscale is better suited for adequate discrimination of the different dots.

We tried, but the white faded into the background, so we left as is.

Suppl. Fig. 5: Please define “slope” and “contrast”.

clarified.

Suppl. Fig. 6: For panel d, I could not find an explanation of how “overlap” was calculated”.

clarified.

Suppl. Fig. 7: How were PSD95 puncta belonging to the filled RGC identified?

Added section to Methods: PSD95 puncta analysis.

Suppl. Fig. 9: The legend says “see Fig. 6”, but this should presumably be “see Fig. 7”.

Fixed.

Suppl. Fig. 10: The legend says “as shown in Fig. 6”, but this should presumably be “as shown in Fig. 7”.

Fixed.

Suppl. Fig. 11: The legend says “To maintain a voltage equivalent to previous simulations (-38 mV, Fig. 6)”. This is confusing. First, “Fig. 6” should presumably be “Fig. 7”. Second, in Fig. 7, it is unclear for what “-38 mV” is a relevant number.

Clarified.

Suppl. Fig. 12: The legend says “see Fig. 6”, but this should presumably be “see Fig. 7”.

The legend says “see Fig. 4”, but this should presumably be “see Fig. 5”.

Fixed.

Suppl. Fig. 13: “yellow array” should be “yellow arrow”.

Fixed.

Reviewer #2

Remarks to the Author

These authors have done a commendable job of addressing the reviewer's comments in general. However, interpreting the EPSC data remains difficult, for this reviewer. In particular, the new data presented R2 figure, shows that kinetic differences in EPSCs in Pix ON and ON alphas are removed when a cocktail of drugs is applied, raising the possibility that cholinergic input or differences in AMPA/NMDA contribute to the EPSCs. If this is true, the observed differences in surround inhibition could arise through postsynaptic processes rather than presynaptically at the BC terminal as the authors suggest.

Since interpreting EPSCs correctly is key to the central claim of this paper, my suggestions would be:

1) demonstrate that the differences in surround inhibition between the two ON cells exist under conditions in which nAChRs and NMDA receptors are blocked.

We have completed the requested experiment (**Reviewer Fig. 1**). We did not find that D-AP5 and Hexamethonium significantly altered surround suppression in our sample of 3 Pix_{ON} RGCs and 3 ON alpha RGCs (**Reviewer Fig. 1c,g**). While 1 of the 3 Pix_{ON} RGCs did show a decrease in surround suppression after drug application, it is unclear if this is a direct effect or the result of network alterations caused by bath application of the drugs. We suspect the latter since D-AP5 + Hexamethonium did not systematically decrease the amplitude of the excitatory conductance to the preferred spot size for either Pix_{ON} or ON alpha (**Reviewer Fig. 1d,h**). This suggests NMDA and acetylcholine receptors are not major contributors to the excitatory conductance in Pix_{ON} and ON alpha RGCs. On the other hand, sub-saturating concentrations of Kynurenic acid or NBQX did significantly decrease the excitatory conductances for the preferred spot sizes (**Fig. 3b-d**), suggesting AMPA receptors drive the Pix_{ON} and ON alpha excitatory conductances.

2) show that the EPSCs reverse at 0 mV (in the presence of inhibitory receptor blockers) to rule out the possibility that gap junctions contribute to the EPSCs in ON RGCs (note gap junctions are implicated in shaping the responses of ON RGCs in primate and mouse retina).

We have completed the requested experiment (**Supplementary Fig. 5**) and do find the response to be typical of AMPA receptors (reverses near 0 mV and largely linear).

Reviewer Fig. 1 | Surround suppression of excitatory conductances during bath application of NMDA and nicotinic receptor antagonists.

a, Excitatory conductances recorded from a Pix_{ON} RGC during 1s stimulation of a preferred size or full-field stimulus. **b**, Same as **a**, but after bath application of D-AP5 (100 μM) and Hexamethonium (100 μM). **c**, Surround suppression of excitatory conductances recorded in control conditions and during bath application of D-AP5 + Hexamethonium for Pix_{ON} RGCs (n=3). Bar plots indicate average ± s.e.m., dots indicate individual RGCs, NS p>0.05, paired two-sample Student's t-test. **d**, Average excitatory conductance elicited by the preferred size stimuli in control conditions and during bath

application D-AP5 + Hexamethonium for the Pix_{ON} RGCs from **c. e-h**, Same as **a-d**, but recorded from ON alpha RGCs.

Minor comments:

Suggested citations:

Line 38: BC as electrically compact units: Please cite Olstedal et al., 2009 (JPhysiol; model BCs) and Poleg-Polsky and Diamond 2016 (JNSc; correlated Ca signals from BC terminals); Grimes et al., 2014 (Elife): This paper shows that ribbon synapses across single RBPCs may have distinct properties and the level of 'cross synaptic' synchrony may vary depending on light levels, which supports the central tenet of the current study.

We have included these citations.

Reviewer #3

Remarks to the Author

The authors have fully addressed my original points. Their investigation of functional divergence within a single bipolar cell type is exceptionally thorough and will be of broad interest.

Thank you!

REVIEWERS' COMMENTS

Reviewer #1 (Remarks to the Author):

D Swygart, W-Q Yu, S Takeuchi, RROL Wong & GW Schwartz.

“A presynaptic source drives differing levels of surround suppression in two mouse retinal ganglion cell types”. Nature Communications (NCOMMS-22-46416-T, revision 2).

In the second revision of their paper, Swygart et al. have adequately addressed most, but not all, of the points in my second review.

With the additional corrections, I believe the manuscript has been further improved and the text now corresponds more adequately and honestly to the evidence obtained from physiological experiments, ultrastructural imaging, and computational modeling. Taken together, this study adds strong evidence to support the idea of differential output from the multiple ribbons of a cone bipolar cell axon terminal, but the mechanism(s) remain(s) to be adequately elucidated.

I have a few additional comments to the authors:

1) The way the paper cites Euler et al. 2014 (lines 37-38, lines 515-516) is in my opinion a misrepresentation. The authors have not responded to the criticism in the previous review.

2) In some places, the writing is still a bit awkward. For example, in lines 554-55, they write:

“However, other inaccuracies, such as a supralinear relation between voltage and glutamate release, ...”

This literally means that “a supralinear relation between voltage and glutamate release” amounts to an “inaccuracy”, but I doubt that this is the intended meaning. The “inaccuracy” that the authors probably have in mind, is their model’s simplified assumption of a linear relation between voltage and glutamate release.

3) As far as I could see, the new figure included in the authors’ response / rebuttal (“Reviewer Fig. 1”) is not included in the revised MS. Given the high relevance of Reviewer 2’s comments, I believe that it would strengthen the paper if this figure is included in the supplementary material.

Reviewer #2 (Remarks to the Author):

These authors have addressed all my concerns. However, I would strongly encourage them to include the new data shown to reviewers (characterization of pharmacologically isolated AMPA EPSCs) in the supplemental section.

We once again thank the reviewers for their responses and suggestions. The reviews are copied below with our responses in blue text. Per reviewer suggestion, we have included **Reviewer Figure 1** from our last response to reviewers as **Supplementary Figure 6** in the current manuscript, which shifts the numbering of all subsequent Supplementary figures by +1.

Reviewer #1 (Remarks to the Author):

In the second revision of their paper, Swygart et al. have adequately addressed most, but not all, of the points in my second review.

With the additional corrections, I believe the manuscript has been further improved and the text now corresponds more adequately and honestly to the evidence obtained from physiological experiments, ultrastructural imaging, and computational modeling. Taken together, this study adds strong evidence to support the idea of differential output from the multiple ribbons of a cone bipolar cell axon terminal, but the mechanism(s) remain(s) to be adequately elucidated.

I have a few additional comments to the authors:

1) The way the paper cites Euler et al. 2014 (lines 37-38, lines 515-516) is in my opinion a misrepresentation. The authors have not responded to the criticism in the previous review. We apologize for accidentally failing to respond to the reviewer's earlier criticism. We have revised these portions of the manuscript.

2) In some places, the writing is still a bit awkward. For example, in lines 554-55, they write: "However, other inaccuracies, such as a supralinear relation between voltage and glutamate release, ..."

This literally means that "a supralinear relation between voltage and glutamate release" amounts to an "inaccuracy", but I doubt that this is the intended meaning. The "inaccuracy" that the authors probably have in mind, is their model's simplified assumption of a linear relation between voltage and glutamate release.

Revised to clearly state the assumption vs. the possible reality.

3) As far as I could see, the new figure included in the authors' response / rebuttal ("Reviewer Fig. 1") is not included in the revised MS. Given the high relevance of Reviewer 2's comments, I believe that it would strengthen the paper if this figure is included in the supplementary material. We have now included this figure as **Supplementary Figure 6**.

Reviewer #2 (Remarks to the Author):

These authors have addressed all my concerns. However, I would strongly encourage them to include the new data shown to reviewers (characterization of pharmacologically isolated AMPA EPSCs) in the supplemental section.

We have now included this figure as **Supplementary Figure 6**.